# Management induced changes of soil organic carbon on global croplands

Kristine Karstens[1, 3], Benjamin Leon Bodirsky[1], Jan Philipp Dietrich[1], Marta Dondini[2], Jens Heinke[1], Matthias Kuhnert[2], Christoph Müller[1], Susanne Rolinski[1], Pete Smith[2], Isabelle Weindl[1], Hermann Lotze-Campen[1, 3], and Alexander Popp[1]

[1]Potsdam Institute for Climate Impact Research (PIK), Member of the Leibniz Association, P.O. Box 60 12 03, 14412 Potsdam, Germany
[2]Institute of Biological & Environmental Sciences, University of Aberdeen, Aberdeen, UK
[3]Humboldt-Universität zu Berlin, Department of Agricultural Economics, Unter den Linden 6, 10099 Berlin, Germany

**Correspondence:** Kristine Karstens (kristine.karstens@pik-potsdam.de)

**Abstract.** Soil organic carbon (SOC), one of the largest terrestrial carbon (C) stocks on Earth, has been depleted by anthropogenic land-cover change and agricultural management. However, the latter has so far not been well represented in global C stock assessments. While SOC models often simulate detailed biochemical processes that lead to the accumulation and decay of SOC, the management decisions driving these biophysical processes are still little investigated at the global scale. Here we develop a spatially explicit data set for agricultural management on cropland, considering crop production levels, residue returning rates, manure application, and the adoption of irrigation and tillage practices. We combine it with a reduced-complexity model based on the IPCC Tier 2 method to create a half-degree resolution data set of SOC stocks and SOC stock changes for the first 30 cm of mineral soils. We estimate that due to arable farming, soils have lost around $34.6\,\mathrm{GtC}$ relative to a counterfactual hypothetical natural state in 1975. Within the period 1975–2010 this SOC debt continued to expand by $5\,\mathrm{GtC}$ $(0.14\,\mathrm{GtC\,yr^{-1}})$ to around 39.6 GtC. However, accounting for historical management led to $2.1\,\mathrm{GtC}$ less $(0.06\,\mathrm{GtC\,yr^{-1}})$ emissions than under the assumption of constant management. We also find that management decisions have influenced the historical SOC trajectory most strongly by residue returning, indicating that SOC enhancement by biomass retention may be a promising negative emissions technique. The reduced-complexity SOC model may allow to simulate management-induced SOC enhancement also within computationally demanding integrated (land-use) assessment modeling.

  **1  Introduction**

Soil organic carbon (SOC), the amount of organic carbon stored in the Earth's soil, exceeds the carbon in the atmospheric and vegetation pools multiple times (Batjes, 1996). Even small changes in processes affecting SOC lead therefore to substantial shifts in the terrestrial carbon cycle and influence the amount of $CO_2$ in the atmosphere (Friedlingstein et al., 2020; Minasny et al., 2017). The specific amount of carbon stored in soils globally is quantified with estimates ranging from 1500 to 2400 GtC
for the first meter of the soil profile (Batjes, 1996; Sanderman et al., 2017).

Natural properties like climatic, biophysical, and landscape characteristics clearly play the most important roles to determine SOC variations over space and time. Recent studies have focused on the evaluation of total SOC stocks of the world as well as on the spatial disaggregation of soil properties such as SOC content (Batjes, 2016; Hengl et al., 2017; FAO, 2018). However, these studies often do not include human interventions, like land cover change and agricultural management, in their analysis.
Compared to climatic and geological driving forces, human interventions alter terrestrial carbon pools over much shorter time scales and are currently one of the most dominant drivers of SOC changes on managed land (Hansis et al., 2015; Bastos et al., 2021).

The anthropogenic impact can be measured by the SOC debt (also referred to as SOC component of land-use change emissions, see Pongratz et al. (2014)), which is the amount of organic carbon soils have lost under cultivation compared to a
hypothetical potential natural vegetation state. Sanderman et al. (2017) identified the anthropogenic SOC debt for the first meter of the soil profile due to land cover change at around 116 GtC (37 GtC for the first 30 cm), compared to previous estimates of 60–130 GtC for the first meter (Lal, 2001).

Global assessments of the carbon cycle via dynamic global vegetation models (DGVMs), Earth System Models (ESMs) or bookkeeping models (BKMs) have analyzed SOC losses as part of a comprehensive evaluation of the global carbon budget and
land-use change (LUC) emissions (Friedlingstein et al., 2020). While providing estimates of the magnitude of SOC losses due to land-cover change, most models lack a detailed representation of agricultural management. Earlier DGVM- and ESM-based assessments only considered changes in land cover, but ignored the removal of biomass at harvest (Strassmann et al., 2008; Betts et al., 2015). BKMs are designed to estimate LUC related emissions and have largely improved in estimating additional emissions from wood harvest and shifting cultivation. However, state of the art models do not consider impacts of varying
agricultural management (Friedlingstein et al., 2020; Houghton et al., 2012; Hansis et al., 2015; Bastos et al., 2021).

Managed agricultural systems were introduced in greater detail to DGVMs and ESMs to improve the assessment of the terrestrial carbon balance (e.g. Bondeau et al., 2007; Lindeskog et al., 2013). Pugh et al. (2015) explicitly consider agricultural management in the form of tillage, irrigation and biomass extraction at harvest, but worked with uniform scenario assumptions on management rather than with historical management data. They also showed the importance of accounting for the land-use
history, as many carbon emissions from agricultural soils are caused by historical LUC and the slow decline of SOC under cropland before a new equilibrium is reached.

In global-scale carbon cycle assessments, management systems are typically represented as spatially explicit patterns that are static in time (e.g. for growing seasons in Portmann et al., 2010; multiple cropping systems in Waha et al., 2020; irrigation

systems in Jägermeyr et al., 2015) or as stylized (in the sense of uniform management assumptions) scenarios (e.g. Pugh et al., 2015; Lutz et al., 2019). Herzfeld et al. (2021) account for historical changes in fertilizer and manure inputs, residue removal rates and tillage systems and report SOC losses from cropland expansion over the period from 1700–2018 of 215 GtC. Within their stylized future management scenarios under future climate change they find that none of the management aspects considered (residue management, no-tillage) can create a positive SOC stock change on current cropland areas that counteracts the still continuing legacy flux from the initial land-use change.

More data sets on spatially explicit agricultural management time series with global coverage are becoming available (e.g. on tillage systems, see (Porwollik et al., 2019; Prestele et al., 2018)) and modeling approaches are increasingly being developed to project the dynamics of management systems into the future (e.g. (Iizumi et al., 2019; Minoli et al., 2019)), but have — to our knowledge — not yet found their way into comprehensive assessments of the terrestrial carbon cycle in DGVMs and BKMs.

Field-scale models (Del Grosso et al., 2001; Coleman et al., 1997; Smith et al., 2010; Taghizadeh-Toosi et al., 2014) are able to better account for historical agricultural management if detailed information on crop yield levels, fertilizer inputs and various other on-farm measures is available for the studied sites. However, due to the lack of comprehensive global management data as input to these models, scaling up to the global domain remains a complex challenge (Morais et al., 2019).

Managed soils have been increasingly studied not only for their carbon emitting behavior, but also because of their capacity to re-store carbon (soil carbon sequestration (SCS) techniques). However, assessing SCS dynamically considering the inter-dependency with environmental, social and economic sustainability targets has been difficult so far, as integrated assessment models (IAMs) (Popp et al., 2016; Rogelj et al., 2018; Forster et al., 2018) have not integrated soil management into their mitigation pathways. More detailed process-based models are typically computationally too demanding to be integrated into optimization-based IAMs. Better accounting for soil carbon management in IAMs thus requires a light-weight model suitable for iterative modeling with detailed options to represent agricultural soil management.

The objectives of our study are (1) to develop a reduced-complexity SOC model able to account for SCS in IAM frameworks; (2) to create a comprehensive data set of the global gridded management time series, including crop production levels, residue input rates, manure amendments, and the adoption of irrigation and tillage practices; and (3) to provide global as well as spatially explicit SOC and SOC debt estimates that consider spatially explicit and time-variant agricultural management. We decompose the contribution of different management activities through a scenario analysis, identifying the most impacting management decisions for SOC development. Moreover, we compare our model performance against other SOC stock and SOC emission estimates, to evaluate the suitability of this reduced-complexity approach for integration into IAM modeling.

## 2  Methods

In Sect. 2.1 we introduce the basic concept of SOC dynamics as applied in this study and explained in more detail in the refinement of the IPCC guidelines vol. 4 Chapter 5 on "Cropland" (Ogle et al., 2019b). We additionally describe how we configured and extended the Tier 2 modeling approach (for model code see Karstens and Dietrich, 2020). In Sect. 2.2 we shortly refer to the concept of stock change factors as outlined in the Tier 1 approach of the IPCC guidelines (Eggleston et al., 2006; Calvo Buendia et al., 2019). Section 2.3 provides a detailed description of the global, gridded management data used to drive the model, including crop production levels, residue input rates, manure amendments, and the adoption of irrigation and tillage practices (for model code see Bodirsky et al., 2020a). In Sect. 2.4 we define the management scenarios used to analyze the role of different management aspects in historical cropland SOC dynamics.

### 2.1  SOC stocks and stock changes following the Tier 2 modeling approach

Following the Tier 2 modeling approach of the refinement of the IPCC guidelines vol. 4 Chapter 5 on "Cropland" (Ogle et al., 2019b); referred to as *Tier 2 modeling approach* in the following), we estimate $SOC$ stocks and their change over time for cropland at half-degree resolution from 1975 to 2010. We restrict our analysis to the first 0-30 cm of the soil profile. Moreover, we assume the current $SOC$ state converges towards a steady state, which itself depends on biophysical, climatic and agronomic conditions. Therefore, we take the following three steps for each year of our simulation period: (1) We calculate annual land-use type-specific steady states and decay rates for $SOC$ stocks (Sect. 2.1.1); (2) we account for land conversion by transferring $SOC$ from and to natural vegetation (Sect. 2.1.2), (3) we estimate $SOC$ stocks and changes based on the stocks of the previous time step, the steady state stocks and the decay rate (Sect. 2.1.3). To initialize the first year of our simulation period we use a spin-up period of 74 years (Sect. 2.1.4).

### 2.1.1  Steady-state SOC stocks and decay rates

In a simple first order kinetic approach the steady-state soil organic carbon stocks $SOC^{\mathrm{eq}}$ are given by

$$SOC^{\mathrm{eq}}_{i,t,sub,lu} = \frac{C^{\mathrm{in}}_{i,t,sub,lu}}{k_{i,t,sub,lu}} \tag{1}$$

with $C^{\mathrm{in}}$ being the carbon inputs to the soil, $k$ denotes the soil organic carbon decay rate. This equation is valid for all grid cells $i$ and all years $t$. We use the Tier 2 modeling approach for our calculations, which assumes three soil carbon sub-pools $sub$ (active, slow and passive) and interactions between them, following the approach in the Century model (Parton et al., 1987). Annual carbon inflow to each sub-pool and annual decay rates of each sub-pool are land-use type $lu$ specific. We distinguish two land-use types: cropland and uncropped land under potential natural vegetation as representative for all other land-use types including forestry and grasslands (referred to as natural vegetation in the following). Forage crops are included within cropland, whereas pastures (including mowed meadows (perennials) and rangelands) are assigned to natural vegetation. Carbon flows connected to livestock are only considered in this study when they originate from cropland feed sources, while

the manure originating from pasture biomass is disregarded, implicitly assuming that this manure is excreted or applied to pastures.

Carbon inputs for cropland are below- and above-ground crop residues left or returned to the field (see Sect. 2.3.2) and manure inputs (see Sect. 2.3.3); for natural vegetation, litterfall including fine root turnover (Schaphoff et al., 2018b) is the only source of carbon inflow to the soil. Following the IPCC guidelines (Ogle et al., 2019b), carbon inputs are disaggregated into metabolic and structural components depending on their lignin and nitrogen content. For each component the sum of all carbon input sources is allocated to the respective $SOC$ sub-pools via transfer coefficients. This implies that both the amount of carbon and its structural composition determine the effective inflow into the different pools.

Whereas residue and manure default lignin and nitrogen fractions are given by the IPCC guidelines (Ogle et al., 2019b), we use plant-functional type and plant-organ specific parameterization for the natural litterfall. Global distribution of plant-functional types is given by (Schaphoff et al., 2018b) as well as separation of litter into leaf, fine root and wood litter compartments excluding litter biomass burnt in wild fires. Leaf litter parameters are given by Brovkin et al. (2012), fine root to leaf litter lignin ratio by Guo et al. (2021), lignin content of wood litter by Rahman et al. (2013) and nitrogen content scaling factors for leaf to fine roots and leaf to wood litter by von Bloh et al. (2018). Data sources for all considered carbon inputs as well as for lignin and nitrogen parameterization are listed in Table 1.

**Table 1.** type and data sources for carbon inputs and parameterization to different land-use types

| land-use types | source of carbon inputs | data source | nitrogen and lignin content |
|---|---|---|---|
| cropland | above-ground residues, below-ground residues, manure | FAOSTAT (2016), Schaphoff et al. (2018b), Weindl et al. (2017) | LG:C generic values according to Table 5.5B, 5.5C from IPCC (Ogle et al., 2019b), crop-specific N:C from Bodirsky et al. (2012) |
| natural vegetation | annual litterfall | Schaphoff et al. (2018b) | leaf N and LG concentration from Brovkin et al. (2012), root to leaf litter LG ratio Guo et al. (2021), lignin content of wood litter Rahman et al. (2013) and nitrogen scaling factors for leaf to root and wood litter from von Bloh et al. (2018) |

The sub-pool specific decay rates $k_{sub}$ are influenced by climatic conditions, biophysical and biochemical soil properties as well as management factors that all vary over space $i$ and time $t$. Following the Tier 2 modeling approach (Ogle et al., 2019b),

we consider temperature $temp$, water $wat$, sand-fraction $sf$, and tillage $till$ effects to account for spatial and temporal variation of decay rates. Thus, $k_{sub}$ rates are given by

$$
\begin{aligned}
k_{i,t,\text{active},lu} &= k_{\text{active}} \quad \cdot temp_{i,t} \cdot wat_{i,t,lu} \quad \cdot till_{i,t,lu} \cdot sf_i \\
k_{i,t,\text{slow},lu} &= k_{\text{slow}} \quad \cdot temp_{i,t} \cdot wat_{i,t,lu} \quad \cdot till_{i,t,lu} \quad . \\
k_{i,t,\text{passive},lu} &= k_{\text{passive}} \quad \cdot temp_{i,t} \cdot wat_{i,t,lu}
\end{aligned}
\tag{2}
$$

For natural vegetation, we assume rainfed and non-tilled conditions, whereas for cropland, we distinguish the effect of different tillage (see Sect. 2.3.5) and irrigation (see Sect. 2.3.4) practices on decay rates. We calculate area-weighted means for $till$ and $wat$ on cropland for each grid cell, using area shares for the different tillage and irrigation practices. Data sources as well as used parameters for the different decay drivers for all land-use types are listed in Table 2; equations are displayed by equations 5.0B–5.0F in Ogle et al. (2019b).

**Table 2.** type and data sources for carbon inputs to different land-use types

| land-use types | type of decay driver | parameter use to represent driver | data source |
|---|---|---|---|
| all | soil quality | sand fraction of the first 0-30 cm | Hengl et al. (2017) |
| | mircobial activity | air temperature | Harris et al. (2020) |
| | soil moisture | precipitation & potential evapotranspiration | Harris et al. (2020) |
| cropland (additionally) | soil moisture* | irrigation | Sect. 2.3.4 |
| | soil disturbance | tillage | Sect. 2.3.5 |

### 2.1.2 SOC transfer between land-use types

We calculate $SOC$ stocks based on the area shares of land-use types $lu$ within our grid cells $i$. If land is converted from one land-use type $lu = \{crop, natveg\}$ into the other $!lu = \{natveg, crop\}$, a respective share of the $SOC$ is reallocated within our budget. We do not distinguish between newly converted and existing cropland, but can work with the average carbon content as the relative decay of $SOC$ is proportional to the $SOC$ stock (see 1). We account for land conversion at the beginning of each time step $t$ by calculating a preliminary stock $SOC_{t^*}$ via

$$
SOC_{i,t^*,sub,lu} = SOC_{i,t-1,sub,lu} - \frac{SOC_{i,t-1,sub,lu}}{A_{i,t-1,lu}} \cdot AR_{i,t,lu} + \frac{SOC_{i,t-1,sub,!lu}}{A_{i,t-1,!lu}} \cdot AE_{i,t,lu}
\tag{3}
$$

with $A_{lu}$ being the land-use type specific areas, $AR_{lu}$ the area reduction and $AE_{lu}$ the area expansion of the two land-use types. Data sources and methodology on land-use states and changes are described in Sect. 2.3.1.

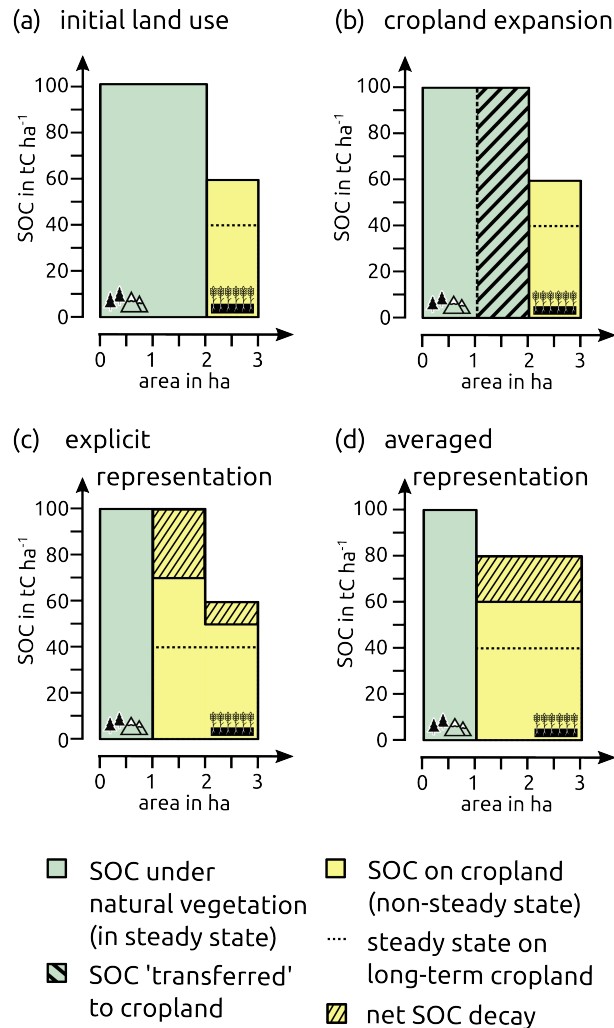

**Figure 1.** Scheme of land-use transition representation. Given an initial land-use pattern (as in this example 2 ha land under natural vegetation and 1 ha of cropland), there are separate $SOC$ stocks for natural vegetation and cropland. While in this example we assume $SOC$ under natural vegetation to be in steady state, the cropland $SOC$ stock approaches its steady state without having reached it yet (a). Upon cropland expansion (in this example half of the natural vegetation is cleared to be used as cropland), $SOC$ stocks on cropland increase due to a transfer of land from natural vegetation (b). Explicitly representing newly converted cropland and existing cropland to account for SOC dynamics (c) leads to the same weighted mean value as averaging $SOC$ stocks (d), due to the linearity of Eq. 4 and cropland-age independent decay rates (see Eq. 2).

### 2.1.3 Total SOC stocks and stock changes

$SOC$ converges towards the calculated steady-state stock $SOC^{\mathrm{eq}}$ for each grid cell $i$, each annual time step $t$, each land-use type $lu$ and each sub-pool $sub$ like

$$SOC_{i,t,sub,lu} = SOC_{i,t^*,sub,lu} + (SOC^{\mathrm{eq}}_{i,t,sub,lu} - SOC_{i,t^*,sub,lu}) \cdot k_{i,t,sub,lu} \cdot 1\mathrm{a}. \tag{4}$$

Note that the decay rates have to be multiplied by one year $(1a)$ to form a dimensionless factor. Reformulating this equation, we obtain a mass balance equation as follows

$$SOC_{i,t,sub,lu} = SOC_{i,t^*,sub,lu} - \underbrace{SOC_{i,t^*,sub,lu} \cdot k_{i,t,sub,lu} \cdot 1\mathrm{a}}_{\text{outflow}} + \overbrace{SOC^{\mathrm{eq}}_{i,t,sub,lu} \cdot k_{i,t,sub,lu} \cdot 1\mathrm{a}}^{\text{input (using equation (1))}}. \tag{5}$$

The global $SOC$ stock for each time step $t$ can then be calculated via

$$SOC_t = \sum_i \sum_{lu} \underbrace{\overbrace{\sum_{sub} SOC_{i,t,sub,lu}}^{SOC_{i,t,lu} - \text{land-use type specific } SOC \text{ stock within cell}}}_{SOC_{i,t} - \text{total } SOC \text{ stock within cell}}. \tag{6}$$

According to the IPCC guidelines $SOC$ changes can be expressed as the difference of two consecutive years (see Eq. 5.0A in Ogle et al., 2019b). This, however, will also include naturally occurring changes due to climatic variation over time. For our study, we define the absolute and relative $SOC$ changes in relation to a potential natural state $SOC^{\mathrm{pnv}}$ under the same climatic conditions in grid cell $i$ at time $t$ that is based on the natural vegetation $SOC$ calculations as defined above without accounting for land conversion from cropland at any time. The absolute changes $\Delta SOC$ and relative changes $F^{\mathrm{SCF}}$ are thus given by

$$\Delta SOC_{i,t} = SOC_{i,t} - SOC^{\mathrm{pnv}}_{i,t} \qquad \text{and} \qquad F^{\mathrm{SCF}}_{i,t} = \frac{SOC_{i,t}}{SOC^{\mathrm{pnv}}_{i,t}}. \tag{7}$$

Note that the absolute changes $\Delta SOC$ can be also interpreted as the SOC debt (Sanderman et al., 2017) due to human cropping activities; whereas relative changes $F^{\mathrm{SCF}}$ can be considered stock change factors as defined within the IPCC guidelines of 2006 (Eggleston et al., 2006). Moreover, $\Delta SOC$ is equivalent to the negated cumulative SOC component of human land-use change emissions (Pugh et al., 2015).

**2.1.4   Initialization of SOC pools**

The initialization of $SOC$ pools is very important and has to include the proper accounting for the land-use history, as many $CO_2$ emissions from agricultural soils are caused by historical land-use change (LUC) and the slow decline of $SOC$ under

crop cultivation, before it reaches a new equilibrium. We initialize our $SOC$ sub-pools using a three-step approach, since input data availability is limited for climate and litter estimates (starting only in 1901) as well as for agricultural management data (starting only in 1965):

Firstly, in order to account for the impacts of legacy fluxes from land-use changes long before the time horizon of interest, we consider land-use change from 1510 onwards. In 1510, we assume all $SOC$ pools to be in natural steady-state, implying that all land-use change prior to that time occurs in 1510. We assume that by 1901, all cropland converted in 1510 has reached its new steady state, so that it is not necessary to explicitly account for even older land conversion. Model inputs for 1901–1930 for climate and natural vegetation litterfall are repeated for 1510–1900 to mimic constant climate conditions for this first initializing period. Similarly, agricultural management data are held constant at the level of 1965 until 1965. We acknowledge that this introduces a bias as agricultural management has changed prior to 1965, but this approach follows others studies on effects of land-use change and management (e.g. Schaphoff et al., 2018a; Herzfeld et al., 2021) and is limited by data availability on harvest statistics (and other management effects).

Secondly, with the availability of transient climate data after 1901, we account not only for land-use change, but also for historical climate change and consequently natural litter inputs to the soil from 1901 to 1965 still considering constant agricultural input data, which are not available prior to 1965.

Thirdly, we run the model for 10 years from 1965 to 1975 with historical dynamic data on agricultural management and start analyzing results from 1975 onward. This is in line with the IPCC guidelines vol. 4 method suggestion to have a 5-20 year spin-up period (Ogle et al., 2019b).

With transient climate considered after 1901, decay rates $k_{sub}$ become dynamic in time. As the decay rates are also affected by irrigation and tillage (see Sect. 2.1.1), we also account for transient changes in irrigated areas after 1901. Data on no-tillage practices are only available after 1974 and we assume conventional tillage on all cropland prior to 1975.

## 2.2 SOC stocks and stock changes following Tier 1

Additionally to the Tier 2 modeling approach (Ogle et al., 2019b) and the detailed analysis of management data coming with it, $SOC$ changes can be estimated using the IPCC Tier 1 approach of IPCC guidelines (Eggleston et al., 2006; Calvo Buendia et al., 2019). Here, stocks are calculated via stock change factors ($F^{\text{SCF}}$) given by the IPCC for the topsoil (0-30 cm) and based on observational data. Note that IPCC factors are derived under the assumption that there is a linear change between steady states over 20 years. Estimates of $F^{\text{SCF}}$ are differentiated by crops, management and input systems (here summarized under $m$) reflecting different dynamics under changed in- and outflows without explicitly tracking these flows. Moreover, estimates of $F^{\text{SCF}}$ vary for different climatic zones ($c$) specified by the IPCC (see Fig. A1). The actual $SOC$ stocks are thus calculated based on a given reference stock $SOC^{\text{ref}}$ by

$$SOC_{i,t} = \sum_{c,m} T_{c,i} \cdot SOC_{i,t}^{\text{ref}} \cdot F_{c,m}^{\text{SCF}} \tag{8}$$

with $T_{c,i}$ being the translation matrix for grid cells $i$ into corresponding climate zones $c$. For this analysis, we use the default $F^{\mathrm{SCF}}$ from the Tier 1 method of Eggleston et al. (2006) and Calvo Buendia et al. (2019) as a comparison and consistency check for our more detailed Tier 2 steady-state approach.

## 2.3 Agricultural management data at 0.5 degree resolution

We compile country-specific FAO production and cropland statistics (FAOSTAT, 2016) to a harmonized and consistent data set. The data is prepared in 5-year time steps from 1965 to 2010, which restricts our analysis to the time span from 1975 to 2010 (after a spin-up phase from 1510–1974). For all the following data, if not declared differently, we interpolate values linearly between the time steps and keep them constant before 1965.

### 2.3.1 Land use and land-use change

Land-use patterns are based on the Land-Use Harmonization 2 (Hurtt et al., 2020) data set (short LUH2), which we sum up from quarter-degree to half-degree resolution. We disaggregate the physical area (given as total land area in million ha) of the five different cropland subcategories (c3ann: C3 annual crops, c3per: C3 perennial crops, c4ann: C4 annual crops, c4per: C4 perennial crops, c3nfx: C3 nitrogen-fixing crops) of LUH2 into our 17 crop groups (see Table "FAO2LUH2MAG_croptypes.csv" in Karstens, 2020a), applying the relative shares for each grid cell based on the country- and year-specific area harvested shares of FAOSTAT data (FAOSTAT, 2016). By calculating country-specific multicropping factors $MCF$ using FAOSTAT data, we are able to compute crop-group specific area harvested on grid cell level. Land-use transitions are calculated as net area differences of the land-use data at half-degree resolution, considering no split up into crop-group specific areas but only total cropland and natural vegetation areas.

### 2.3.2 Crop and crop residues production

Crop production patterns are compiled crop group specific using half-degree yield data from LPJmL (Schaphoff et al., 2018b) as well as half-degree cropland patterns (see Sect. 2.3.1). We calibrate cellular yields with a country-level calibration factor for each crop group to meet historical FAOSTAT production (FAOSTAT, 2016). Note

Crop residue production and management is based on a revised methodology of Bodirsky et al. (2012) and key aspects are explained as they play a central role in soil carbon modeling. Starting from crop yield estimates $Y$ and respective physical crop area $CA$, we estimate total above-ground $AGR$ and below-ground $BGR$ residue biomass (in tonnes) using crop group $cg$ specific ratios for above-ground residues to harvested biomass $r_{cg}^{\mathrm{ag,prod}}$ in $(tDM\,ha^{-1})(tDM\,ha^{-1})^{-1}$ above-ground residues to harvested area $r_{cg}^{\mathrm{ag,area}}$ in $\mathrm{tDM\,ha^{-1}}$ and below-ground residues to above-ground biomass $r_{cg}^{\mathrm{bg}}$ in $\mathrm{tDM\,tDM^{-1}}$ as follows

$$
\begin{aligned}
AGR_{i,t,cg} &= CA_{i,t,cg} \cdot \left( Y_{i,t,cg} \cdot r_{cg}^{\mathrm{ag,prod}} + MCF_{i,t} \cdot r_{cg}^{\mathrm{ag,area}} \right) \qquad \text{and} \\
BGR_{i,t,cg} &= \left( CA_{i,t,cg} \cdot Y_{i,t,cg} + AGR_{i,t,cg} \right) \cdot r_{cg}^{\mathrm{bg}} \quad .
\end{aligned}
\tag{9}
$$

Following the IPCC guidelines, we split the above-ground residue calculations into a yield-dependent slope ($r^{\mathrm{ag,prod}}$) and a positive intercept ($r^{\mathrm{ag,area}}$) fraction (Hergoualc'h, Kristell et al., 2019). Residues biomass therefore increases under-proportionally with rising yields, reflecting a shifting harvest index of higher-yielding breeds. Deviating from Bodirsky et al. (2012) we use harvested instead of physical crop area (denoted in Eq. (9) by MCF described in Sect. 2.3.1) to account for increased residue biomass due to multiple cropping (multiple harvests with each lower yields) and decreased residue amounts due to fallow land. We assume that all $BGR$ are left in the soil, whereas $AGR$ can be burned or harvested for other purposes such as feeding animals (Weindl et al., 2017), fuel or for material use.

A country-specific fixed share of the $AGR$ is assumed to be burned on field depending on the per-capita income of the country. Following Smil (1999b) we assume a burn share of 25% for low-income countries according to World Bank definitions ($< 10000\,\mathrm{USD\,yr^{-1}\,cap^{-1}}$), 15% for high-income ($> 10000\,\mathrm{USD\,yr^{-1}\,cap^{-1}}$) and linearly interpolate shares for all middle-income countries depending on their per-capita income for the periods before 1995. After 1995 we estimate a linear decline of burn shares to 10% for low-income countries and 0% for high-income countries till 2025 to account for recent increases in air pollution regulation. The estimated trends show good agreement with fire-satellite-image derived estimates by the Global Fire Database (van der Werf et al., 2017). Depending on the crop group, 80–90% of the carbon in the crop residues burned in the fields is lost within the combustion process (Eggleston et al., 2006).

From our 17 crop groups, we compile four residue groups (straw, high- and low-lignin residues, residues without dual use), of which the first three are taken away from the field for other purposes (see mappingCrop2Residue.csv in Bodirsky et al. (2020a)). Residue feed demand for five different livestock groups is based on country-specific feed baskets (see Weindl et al., 2017) that differentiate between the residue groups and take available $AGR$ biomass as well as livestock productivity into account. We estimate a material-use share for the straw-residue group of 5% and a fuel-share of 10% for all used residue groups in low-income countries. For high-income countries, no withdrawal for material or fuel use is assumed, and use shares of middle-income countries are linearly interpolated based on per-capita income, following the same rationale as for the share of burnt residues described above. The remaining $AGR$ as well as all $BGR$ are expected to be left on the field. We limit high residue return rates to at most $10\,\mathrm{t\,C\,ha^{-1}}$ in order to correct for outliers.

To transform dry matter estimates into carbon and nitrogen, we compiled crop-group and plant-part specific carbon and nitrogen to dry matter ratios (see Table A1).

### 2.3.3 Livestock distribution and manure excretion

Manure especially from ruminants is often excreted at pastures and rangelands, but due to the intensification of livestock systems a lot of the manure has to be stored and can be applied on cropland. We assume that manure is applied in close proximity to its excretion, so that the distribution of livestock is the driving factor for the spatial pattern of manure application.

To disaggregate country level FAOSTAT livestock production data to half-degree resolution, we use the following rule-based assumptions, drawing from the approach of Robinson et al. (2014) and applying feed basket assumptions based on a revised methodology from Weindl et al. (2017). We differentiate between ruminant and monogastric systems, as well as extensive and intensive systems. Due to high feed demand of ruminants, we assume that ruminant livestock is located where the production

of feed occurs to minimize transport of feed. We distinguish between grazed pasture, which is converted into livestock products in extensive systems, and primary-crop feed stuff, which we consider to be consumed in intensive systems. For poultry, egg and monogastric meat production we use the per-capita income of the country to distinguish between intensive and extensive production systems. For low-income countries, we assume only extensive production systems. We locate them according to the share of built-up areas based on the assumption that these animals are held in subsistence or small-holder farming systems with a high labor-per-animal ratio. Intensive production associated with high-income countries, is distributed within a country using the share of primary-crop production, assuming that feed availability is the most determining factor for livestock location. For middle-income countries we split the livestock production into extensive and intensive systems based on the per-capita income.

Manure production and management is based on a revised methodology of Bodirsky et al. (2012) and is presented here due to its central role in soil carbon modeling. Based on the gridded livestock distribution we calculate spatially explicit excretion by estimating the nitrogen balance of the livestock systems on the basis of comprehensive livestock feed baskets (Weindl et al., 2017), assuming that all nitrogen in protein feed intake, minus the nitrogen in the slaughter mass, is excreted. Carbon in excreted manure is estimated by applying fixed C:N ratios, which range from 10 for poultry up to 19 for beef cattle (for full detail see Calvo Buendia et al. (2019b). Depending on the feed system we assume manure to be handled in four different ways: All manure originated from pasture feed intake is excreted directly on pastures and rangelands (pasture grazing), deducting manure collected as fuel. Whereas for low-income countries, we adopt a share of 25% of crop residues in feed intake directly consumed and excreted on crop fields (stubble grazing), we do not consider any stubble grazing in high-income countries; middle-income countries see linearly interpolated shares depending on their per-capita income. For all other feed items, we assume the manure to be stored in animal waste management systems associated with livestock housing. To estimate the carbon actually returned to the soil, we account for carbon losses during storage, where return shares depend on different animal waste management and grazing systems. Whereas we assume no losses for pasture and stubble grazing, we consider that the manure collected as fuel is not returned to the fields. For manure stored in different animal waste management systems we compiled carbon loss rates (see calcClossConfinement.R in Bodirsky et al. (2020a) for more details) depending on the different systems and the associated nitrogen loss rates as specified in Bodirsky et al. (2012). We limit high application shares at $10\,\text{tC}\,\text{ha}^{-1}$ to correct for outliers that can occur due to inconsistencies between FAO production and 0.5 degree land-use data.

### 2.3.4  Irrigation

The LUH2v2 (Hurtt et al., 2020) data set provides irrigated fractions for its cropland subcategories. We sum up irrigation area shares for all crop groups within a grid cell, and calculate the water effect coefficient $wat$ on decay rates using these shares to compute the weighted mean between rainfed and irrigated $wat$ factors. As a result $wat$ is the same for all crop groups within a grid cell. Furthermore, we suppose the irrigation effect to be present for all 12 months of a year, since we do not have consistent crop group specific growing periods available. This will lead to an overestimation of the irrigation effect. We expect, however, water limitations to be a minor problem during the off-season in temperature limited cropping regions, causing our assumption to not dramatically overestimate the moisture effects. In tropical, water-limited cropping areas, irrigated growing periods might even span over the whole year.

### 2.3.5 Tillage

In order to derive a spatial distribution of the three different tillage types specified by the IPCC — full tillage, reduced tillage and no tillage —, we assume that all natural land and pastures are not tilled, whereas annual crops are under full and perennials under reduced tillage per default. Furthermore, we assume no tillage in cropland cells specified as no tillage cell based on the historical global gridded tillage data set from Porwollik et al. (2019). This data set is extended to the period of 1975–2010 by combining country-level data on areas under conservation agriculture from FAO (2016) and half-degree resolution physical crop areas from Hurtt et al. (2020), applying the methodology of Prowollik et al. (2019) to identify potential no-tillage grid cells.

## 2.4 Scenario definitions

To highlight the impact of changing management effects and to assess the sensitivity of the model towards different initialization choices, we perform a set of scenario runs. In the following section we outline name and idea of these scenarios (for technical implementation see Karstens and Dietrich, 2020).

To single out the impact of tillage practices, residue and manure inputs, we defined scenario with constant values for these three drivers: In the *constTillage* scenario the adoption of no-tillage practices are neglected (adoption starts in 1974 according to the available data set). The *constResidues* and the *constManure* scenario assume constant input rates from residues resp. manure (in $tha^{-1}$) at the level of 1975 onward. Within the *constResidue* scenario at different effects overlay each other: yields and with them residue biomass increase due to productivity gains; rates of residue left or returned to fields are raising; and shifts of cropping pattern change the amount of residue biomass due to crop-group specific harvest index values. The *constManagement* combines all three scenarios *constTillage*, *constResidues* and *constManure*.

As outlined in Sect. 2.1.4 we assume to start in a steady-state SOC stocks for the start year of the spin-up in 1510 followed by a long spin-up period of (*Initial-spinup1510*). As some SOC compartments decay over very long timescales, the initialization setting might strongly effect the overall outcome of SOC stocks and changes. Thus we conduct two counterfactual scenarios *Initial-natveg* and *Initial-eq*. Whereas in *Initial-natveg* we assume SOC stocks to be in a steady-state SOC under potential natural vegetation for all land-use types in 1901, SOC stocks start in their land-use type specific steady state in 1901 for the *Initial-eq* scenario. In that way, the two scenarios mark the two extreme cases: In *Initial-eq* all legacy fluxes already appeared in 1901, whereas in *Initial-natveg* all legacy fluxes before 1901 still have to appear. We additionally combined the counterfactual scenarios with the *constManagement* scenario.

## 3 Results

Detailed results for the spatially explicit global SOC budget including intermediate results on input data from manure and residues as well as SOC stock results for all scenario runs can be found in the data repository from Karstens (2020a). In the following, the most important results (see Karstens, 2020b, for post-processing scripts) are summarized.

### 3.1 SOC distribution and depletion

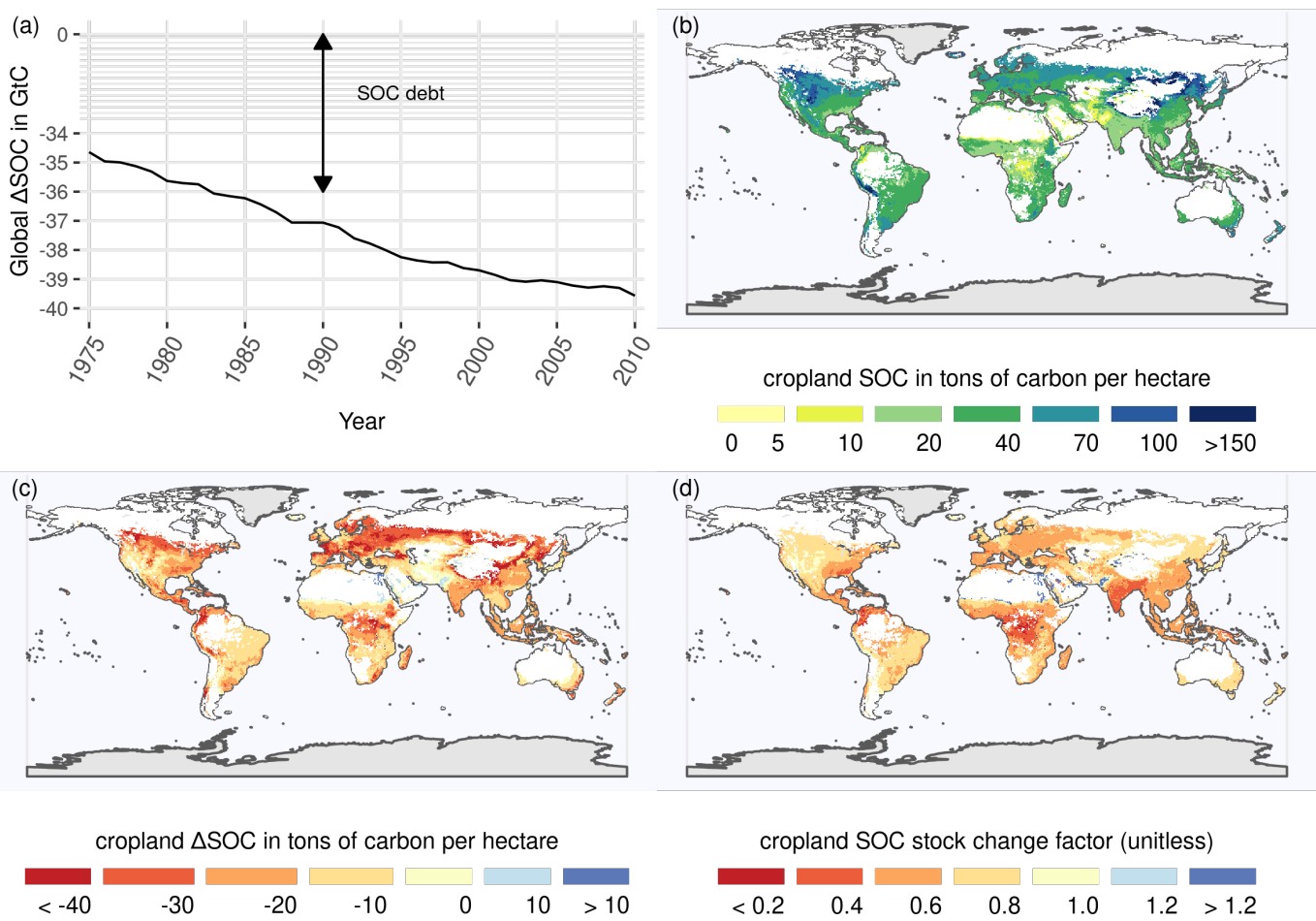

**Figure 2.** Global SOC stocks and SOC stock changes on cropland for the first 30 cm of the soil profile considering historical management data. Panel (a) shows global $\Delta SOC$ between historical land use and potential natural vegetation (PNV). The distribution of total global SOC stocks for the first 30 cm on cropland for the year 2010 is depcited in panel (b). Absolute (c) and relative (d) SOC stocks changes for the year 2010 are compared to a potential natural state identify different hotspots of SOC losses and gains.

The global SOC debt has increased by about 14% in the period between 1975 and 2010 from 34.6 to 39.6 GtC (Fig. 2(a)). This corresponds to an average loss rate of $0.14\,\mathrm{GtC\,yr^{-1}}$ in comparison to a hypothetical potential natural vegetation (PNV) state. Considering our estimate of the global SOC stock of around 705 GtC in the upper 30 cm in 1975, global SOC decreased by 0.2 per 1000 per year for the period between 1975 and 2010 in comparison to the PNV state. The speed of this SOC loss has decreased towards the end of the modeling period. Note that the SOC stock itself — without comparing it to a a PNV state — increases during the period between 1975 and 2010 from 705 GtC to 712 GtC, which corresponds to overall SOC stock increase of $0.2\,\mathrm{GtC\,yr^{-1}}$.

In Fig. 2(b) we provide a world map of SOC stocks estimates for the first 30 cm on cropland considering historical management data for the year 2010. Values range between over $100\,\mathrm{t\,ha^{-1}}$ in northern temperate cropland to less than $5\,\mathrm{t\,ha^{-1}}$ for arid and semiarid cropland. Our spatially explicit results show hotspots of SOC losses and gains compared to SOC under PNV in two complementary ways: 1. Absolute SOC changes $\Delta SOC$ (Fig. 2(c)) indicate areas with high importance for the global $SOC$ loss. They can be driven by large relative changes (e.g. in Central Africa) or by a high natural stock, from which even small relative deviations could lead to substantial absolute losses (e.g. North-East Asia). 2. Relative SOC changes measured as stock changes factors $F^{SCF}$ (Fig. 2(d)) are a helpful metric to analyze the impact of human cropping activities. They indicate areas with large differences in carbon inflows or SOC decay compared to natural vegetation that may hold potential to be overcome by improved agricultural practices. Large parts of tropical cropland seem to suffer from strongly reduced relative stocks, indicating SOC degradation. Conversely, irrigated cropland at the border to dry, unsuitable areas worldwide shows a strong relative increase in SOC stocks.

The spatial distribution of the total $\Delta SOC$ summed over all land-use types (Fig. 3(a)) and its change from 1975 to 2010 (Fig. 3(b1)) reveals areas of SOC debt decline and increase. Regions with large cropland expansion (e.g. Brazil, Southeast Asia, Canada) continue to lose SOC, whereas regions with cropland reduction (and thus SOC restoration) or with accumulating cropland SOC can be found e.g. in highly productive areas of Europe and Central USA.

## 3.2 Carbon flows in the cropland system

C is removed from the atmosphere via plant growth and allocated to different plant parts, which we aggregate to three pools (harvested organ, above- and below-ground residues). Whereas harvested organs as well as above ground-residues are taken (partially) from the field to be used for other purposes, below-ground residues (729 MtC in 2010) are directly returned to the field. We divide crop biomass usage into feed usage and aggregate all other usage types (e.g. food, bioenergy and material) into a human demand category. Livestock feed demand for crop organ harvest and above-ground residues of 1136 MtC is roughly equal to the human demand of 1129 MtC. Whereas large parts of feed intake are returned to the soils via manure (C input from manure at 384 MtC), we assume the carbon demanded from humans (ending up as e.g. compost, night soil and sewage) is not returned to soils. Besides manure C and below-ground residues, above-ground residues form the largest C input to the soil with 1350 MtC returned to the fields in the year 2010. However, around 60% of this organic C decomposes before it is integrated into soils. Due to the different composition of organic C, proportionally more C enters the slow pool from manure than from crop residue. According to our model results, land-use change dynamics led to a C transfer from natural vegetation

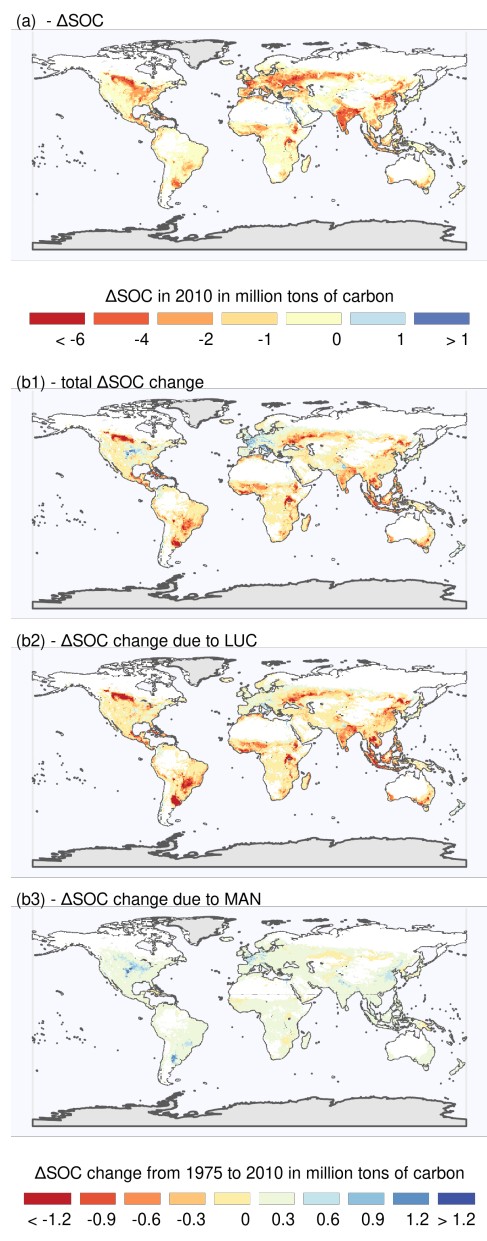

**Figure 3.** Global total $\Delta SOC$ and $\Delta SOC$ change for the first 30 cm of the soil profile. Panel (a) shows global $\Delta SOC$ as the difference between $SOC$ under historical land use and $SOC$ under potential natural vegetation (PNV) in the year 2010 summed over all land-use types. Computing the difference between the $\Delta SOC$ estimate for 2010 and for 1975 (b1) depicts areas of SOC depletion (SOC debt increase, red) and SOC accumulation (SOC debt decline, blue). Panel (b2) shows the of land-use change (LUC) induced change of $\Delta SOC$ between 1975 and 2010, whereas panel (b3) depicts the change due to changing agricultural management (MAN).

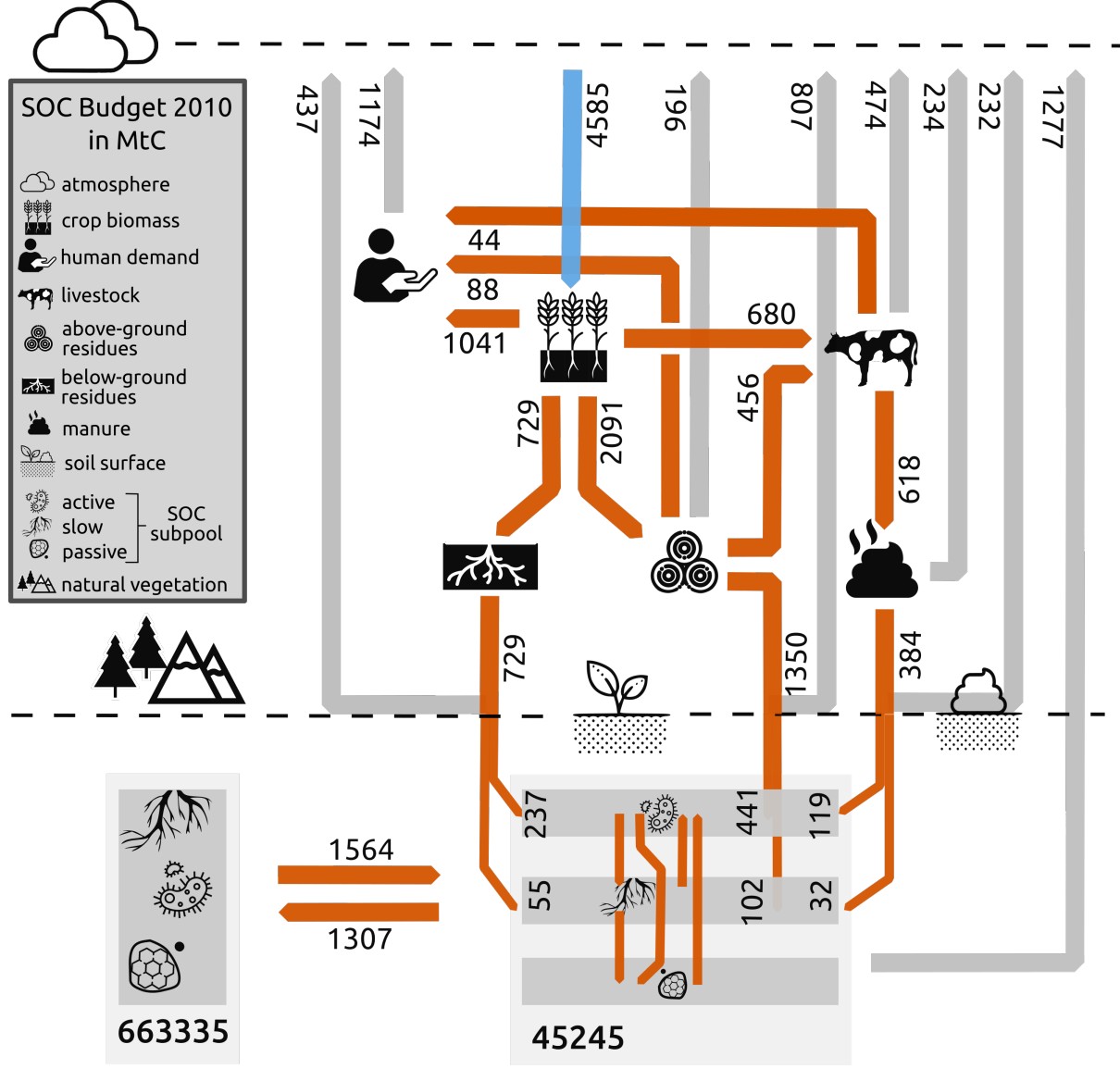

**Figure 4.** Global carbon flows within the cropland system for the year 2010 (in $\mathrm{MtC}$). Carbon is first photosynthesised by crop plants and then used for livestock feed and various other usages subsumed under human demand. After accounting for losses within the cropland system, there are three major C inputs to cropland SOC: manure, above- and below-ground residues. Large parts of C, however, are mineralized on the field before entering the soil. Additionally, C is transferred to and from global cropland soil stock via land-use change between cropland and natural vegetation. Finally, SOC is mineralized and flows back to the atmosphere.

to cropland of $257\,\mathrm{MtC}$ in 2010. The cropland system receives $4585\,\mathrm{MtC}$ assimilated by crop plants and releases $3554\,\mathrm{MtC}$

mostly through respiration. Accounting for SOC transfer and decomposition, the net SOC decrease of global cropland is around $33\,\mathrm{MtC}$ for the year 2010.

## 3.3  Agricultural management effects on SOC debt

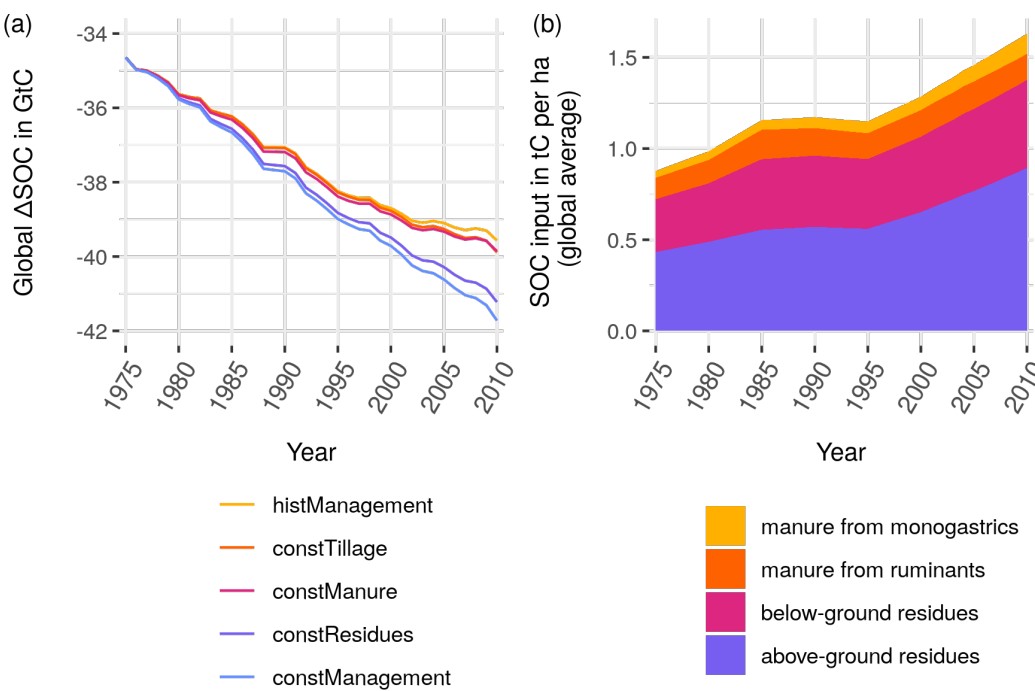

**Figure 5.** (a) Global $\Delta SOC$ in $\mathrm{GtC}$ for different management scenarios. The stylized scenarios deviate from historical agricultural management patterns (histManagement) by holding effects of carbon inflows from residues (constResidues) or manure (constManure) constant at the 1975 level, or neglecting adoption of no-tillage practices over time (constTillage). The scenario constManagement combines all three modifications. Note that $\Delta SOC$ is defined as the difference of SOC under land-use compared to a hypothetical natural vegetation state. Panel (b) shows the carbon inflows from crop residue and manure.

We analyze the relative impact of different management practices by comparing the actual historical management scenario with counterfactual scenarios, where individual management practices (residues in constResidues, manure in constManure, tillage practices in constTillage, all three in constManagement) are kept static at the 1975 values (Figure 5(a)). As shown by the difference between the constResidues scenario and the other counterfactuals, changes in residue return rates dominate the management effects. Without the historical increase in C inputs from residues to agricultural soils, the global $\Delta SOC$
would decrease to $41.7\,\mathrm{GtC}$ at a rate of $0.20\,\mathrm{GtC\,yr^{-1}}$ — a 35% increase compared to $0.14\,\mathrm{GtC\,yr^{-1}}$ for the histManagement estimates. Both the constManure and constTillage scenarios show only small deviations from the historical agricultural management values with $0.15\,\mathrm{GtC\,yr^{-1}}$. The effect of no-tillage only becomes discernible from 2000 onwards. The large con-

tribution of residues relative to manure also becomes visible when considering the annual C inputs of residues and manure to soils over a period of 35 years (Fig. 5(b)).

Using the constManagement results that only include land-use change (LUC) related changes of the SOC debt between 1975 and 2010, we are able to subtract the LUC effect from the overall SOC debt change within the histManagement results. The remaining effect can be attributed to the changing agricultural management (MAN) as other drivers such as climatic effects have been already canceled out by taking the difference to a PNV reference state when calculating $\Delta SOC$. The increasing SOC debt on global cropland are primarily caused by LUC (red areas in Fig. 3(b2)). Deteriorated management also contributed to increasing SOC debt in parts of Sub-saharan Africa and Central Asia. In contrast, agricultural management has led to an decrease in SOC debt in the USA, Europe, as well as in parts of China and India (blue areas in Fig. 3(b3)), which is not visible in the total $\Delta SOC$ change as LUC was happening at the same time.

Our sensitivity analysis shows that the management impact is robust to the initialization of SOC stocks (see Fig. A2 and Sect. 2.4) with around $2.15\,\mathrm{GtC}$ difference in SOC debt between the histManagement and the constManagement scenario. However, the SOC debt and SOC debt change varies with the different initialization choices. Whereas the default assumption (Initial-spinup1510) shows a $\Delta SOC$ of $39.6\,\mathrm{GtC}$ for the year 2010, the Initial-natveg scenario with high legacy fluxes to come only has a $\Delta SOC$ of $33.3\,\mathrm{GtC}$ and the Initial-eq scenarios with only little legacy fluxes left already a $\Delta SOC$ of $50.7\,\mathrm{GtC}$.

### 3.4 Model evaluation

To evaluate our model results against reference data in five steps: (1) we compare our stock change factors (see Sect. 2.2) to IPCC default assumptions (Lasco et al., 2006; Ogle et al., 2019b); (2) we compare our global (and climate-zone specific) total SOC stocks to other literature estimates; (3) we compare our results to point measurements. To evaluate the representation of our natural SOC stocks (4) we correlated LPJmL4 SOC stocks for PNV with our natural state SOC results on grid level; and (5) we do a similar correlation analysis for our modeled actual SOC stocks in comparison to the results of SoilGrids 2.0 (Poggio et al., 2021), which accounts for actual land use too.

### 3.4.1 Stock change factors compared to IPCC assumptions

To evaluate our modeled SOC stocks and stock changes under agricultural management, we compare our results to the default IPCC stock change factors $F^{\mathrm{SCF}}$ of 2006 (Lasco et al., 2006) and their refinements in 2019 (Ogle et al., 2019b). Both estimates are based on measurement data for cropland (see Fig. 6). To allow for comparison, we aggregate our stock change factors weighted by grid-level cropland area to derive median factors for the four IPCC climate zones (Fig. A1). Note that IPCC Tier 1 factors are derived under the assumption that there is a linear change between steady states over 20 years, whereas our aggregated factors just reflect the relative change compared to a given potential natural vegetation reference stock without specifically tracking age of the cropland.

Stock change factors for temperate climate zones of this study are lower than the default values of the IPCC. For the tropical regions the IPCC factors increased notably from the guidelines in 2006 (Lasco et al., 2006) to the update in 2019 (Ogle et al.,

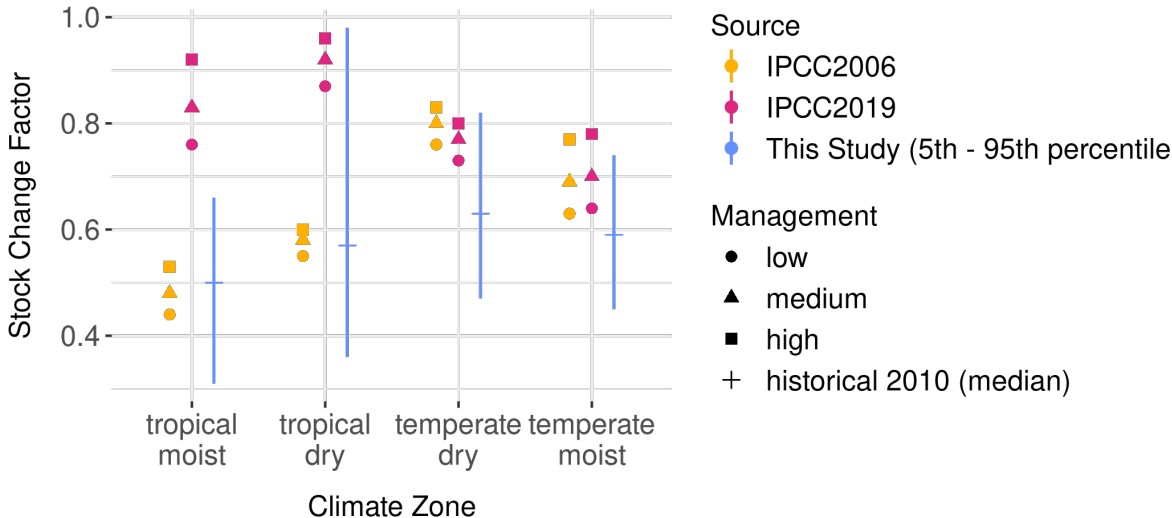

**Figure 6.** $F^{\mathrm{SCF}}$ in comparison to IPCC Tier 1 default factors from the guidelines in 2006 (Lasco et al., 2006) and the update in 2019 (Ogle et al., 2019b). IPCC factors vary a bit with different types of management (low, medium, high). The results of the study span over a wide range – indicated by the span between the 5th and 95th percentile – but not overlapping with the IPCCs 2019 factors for tropical moist. The median for most regions is considerably lower than the IPCC factors.

2019b) due to the inclusion of more data points. Our results span over a broad range due to the different ages of the cropland but also due to different agricultural management practices within climate regions.

### 3.4.2 Global SOC stocks comparison

We compare our global SOC stocks with a wide range of global SOC stock estimates for the first 30 cm of the soil profile, using data from WISE (Batjes, 2016), SoilGrids (Hengl et al., 2017), GSOC (FAO, 2018), LPJmL4 (Schaphoff et al., 2018a), SoilGrids 2.0 (Poggio et al., 2021), and SOCDebtPaper (Sanderman et al., 2017) in Fig. 7.

The global estimates of the total SOC stock of the upper 30 cm from this study are in the middle of the wide range of other modeled or observation-based estimates. Regional results (Fig. A3) show that our estimates are well within the range of other estimates for most regions, but at the lower end for tropical moist and tropical wet areas. SoilGrids (Hengl et al., 2017) especially stands out with its high estimate, since they include the litter horizon on top of the soil that might dominate especially polar and boreal soils. SoilGrids 2.0 (Poggio et al., 2021) however, excludes litter C and thus marks the lower end for particularly northern regions. For the same reason it is also more comparable to our results, which do not account for litter C as well.

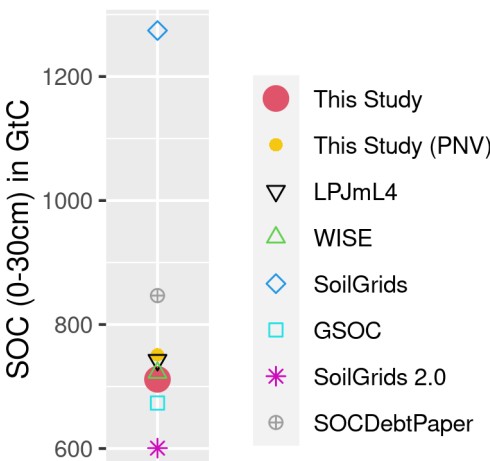

**Figure 7.** Modeled as well as observation-based estimates for global SOC stock in $\text{GtC}$ for the first 30 cm of soil aggregated over all land area. The comparison against observation-based data (SoilGrids, SoilGrids 2.0, GSOC and WISE) is supplemented by modeled data from LPJmL4 (Schaphoff et al., 2018a) and estimates from (Sanderman et al., 2017). We show values of this study for the year 2010 accounting for the historical land-use dynamics as well as for an hypothetical PNV.

### 3.4.3 Point-based evaluation

In Fig. A4 we correlate our SOC results for natural vegetation and cropland in 2010 with literature values from point measurements (for data base see appendix of Sanderman et al., 2017). The goodness of the fit is very low with an $R^2$ of $0.13$. Individually the correlations are even lower with a $R^2$ of $0.09$ for cropland and $0.08$ for areas of natural vegetation. This point to the fact that differences between land-use type SOC stocks could be better matched than the spatial pattern of the rather small point measurement data base. Due to the low number of small-scale measurements, statistical properties of the point data variability are not derived and thus, could not be used to improve the point-to-grid-cell comparison (see Rammig et al., 2018).

### 3.4.4 Natural SOC stock comparison with LPJmL4

Estimates of SOC stocks under natural vegetation influence our modeled results for cropland, which has been converted from natural vegetation at some point in time. As the Tier 2 modeling approach (Ogle et al., 2019b) is not specifically parameterized for natural vegetation it is important to evaluate its suitability to produce reasonable results in that domain as well at least comparable to other modeling approaches. We therefore also compare our modeled results for SOC under natural vegetation (derived using litterfall of LPJmL4) against estimates of SOC by LPJmL4 for a PNV simulation (see Fig. A5). Both models are driven by the same climate conditions and the same natural litterfall and just differ in the representation of SOC and litter dynamics. With our focus on cropland SOC dynamics, we compare only cells with more than $1000\,\text{ha}$ of cropland (capturing 99.9% of global cropland area). Spatial correlations of PNV SOC stock values are high (global $R2 = 0.81$), especially for dry

climate zones (Fig. A5). For temperate and tropical moist areas estimates of this study tend to be a bit lower compared to LPJmL4 results.

### 3.4.5 Actual SOC stock comparison with SoilGrids 2.0

SoilGrids 2.0 (Poggio et al., 2021) is a digital soil mapping approach that uses over 240 000 soil profile observations to produce high resolution soil maps including SOC stocks and estimates of their uncertainties. To evaluate the performance of our model at the global scale, we correlate SoilGrids 2.0 SOC stock values, which were aggregated to 0.5 degree resolution, to our estimates for the year 2010 in Fig. 8. To focus our comparison on cropland areas, we mask out grid cells with less than $1000\,\mathrm{ha}$ of cropland. Spatial correlation is moderate for tropical climate zones, whereas it is low for temperate moist areas. In tropical dry and temperate dry areas, we simulate also very low SOC values (below $10\,\mathrm{t\,C\,ha^{-1}}$), which is not found in SoilGrids 2.0 whereas our modeled SOC stocks can be substantially higher in temperate moist areas than reported by SoilGrids 2.0. Additionally, we use the uncertainty estimates from SoilGrids 2.0 in Fig. 9 to identify areas, where our modeled SOC stocks that are below the 5th or above the 95th percentile of the SoilGrids 2.0 data. For the vast majority of grid cells our model results are between the 5th and 95th percentile of SoilGrids 2.0 estimates. We underestimate SOC stocks especially in dry areas (e.g. close to the Sahara). Overestimated stocks are often situated in mountainous regions.

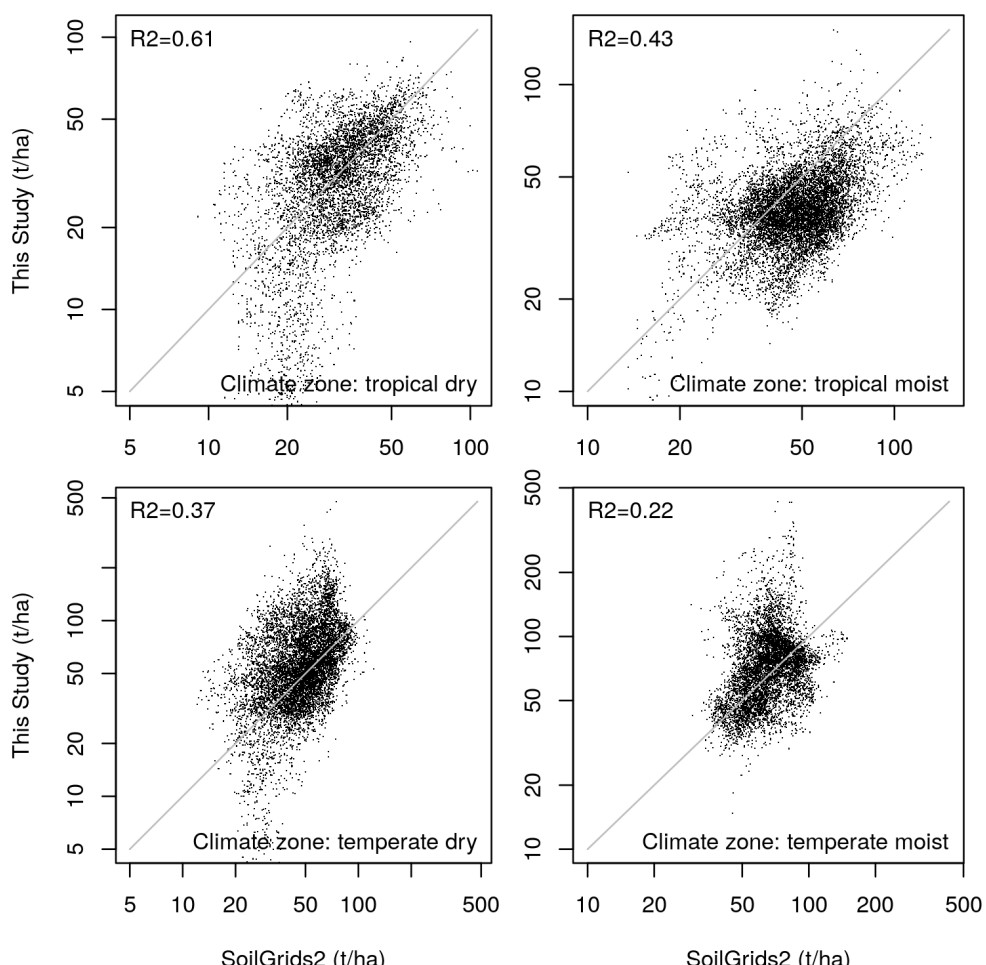

**Figure 8.** Correlation between modeled SOC stocks of this study and projected values from SoilGrids 2.0.

## 4 Discussion

We have (1) developed a reduced-complexity model and (2) compiled a spatially explicit time series data set of agricultural

management data in order to (3) analyze the role of agricultural management in historical cropland SOC dynamics. Our study shows that information on agricultural management alters estimates of the SOC debt and slows down loss of SOC compared to the often used constant management assumptions.

It is important to evaluate the validity of our results as modeling management effects on SOC dynamics at the global scale comes with large uncertainties. The model includes a large number of parameters, and for most of these the uncertainty

distributions have not been quantified so far. Moreover, we think that beyond parameter uncertainty, the structural uncertainty

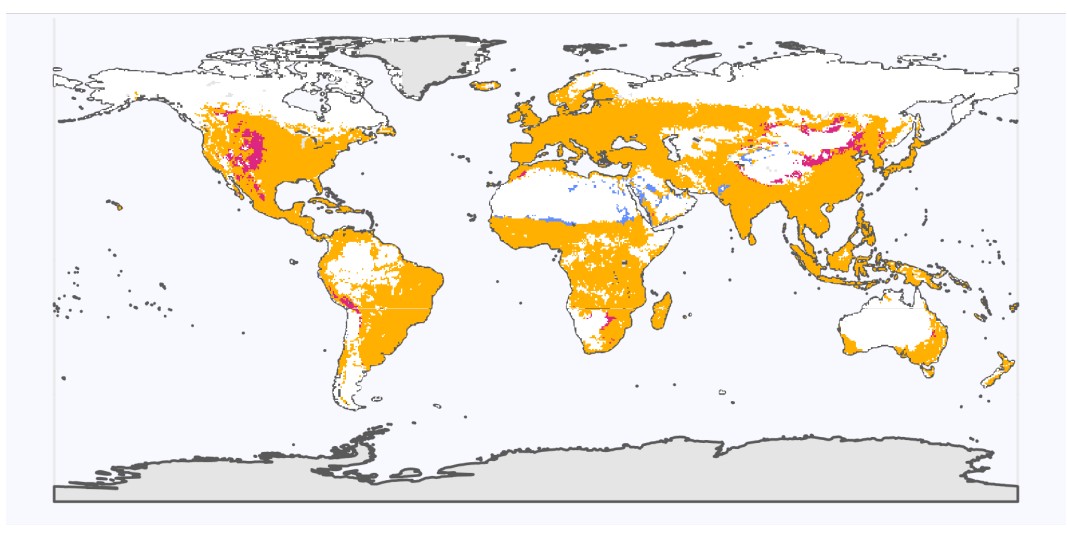

Modeled SOC stock in relation to uncertainty range of SoilGrids 2.0

lower than Q0.05   inside [Q0.05, Q0.95]   higher than Q0.95   No Cropland

**Figure 9.** Global map on SOC results compared to uncertainty estimates from SoilGrids 2.0.

from the model design is high. The management data itself is prone to uncertainties as well, as most of it is only indirectly calculated from reported data.

In the following, we give a qualitative assessment of the uncertainties and limitations of this study as well as discuss our three study objectives and results against existing literature.

## 4.1 Comprehensive historical agricultural management data set

Our spatially explicit time series data set of agricultural management is based on country-specific FAO production and cropland statistics (FAOSTAT, 2016) as well as 0.5 degree land-use data from LUH2 (Hurtt et al., 2020). Starting from these two sources, we derive a harmonized and consistent data set for the major C flows within the cropping system (4) using a mass balance approach from the IPCC guidelines Vol. 4 (Eggleston et al., 2006; Calvo Buendia et al., 2019) and other auxiliary data sets (e.g. Porwollik et al., 2019).

For some of the aspects covered in our data set, for example livestock distribution (Robinson et al., 2014) or manure production and application (Zhang et al., 2017), well-compiled data sets in high resolution exist that capture real world conditions much better than our estimates. However, they often come with the caveats of either being static in time, demanding large sets of auxiliary data or being inconsistent with each other.

For most of the parameters used in our management estimates no uncertainty estimates exist. This is why, in our view, a large part of the uncertainty with respect to the impacts of SOC management — next to the parametric and structural uncertainty of the soil model — is included in the management data itself. This is especially the case for the residue and manure production and application numbers, as these are only indirectly derived from crop and livestock production, feed and area data (FAOSTAT, 2016; Weindl et al., 2017). The uncertainty of recycling shares adds on top of the uncertain total numbers of manure and residue

biomass. Previous modeling studies of SOC carbon on cropland often only used stylized scenarios of management practices (Pugh et al., 2015; Lutz et al., 2019), rather than trying to estimate real management.

    While our data set, by including crop residues and manure, likely the largest carbon inputs to soils, it does not account for a list of minor carbon inputs from cover crops, agroforestry, green manure, weed biomass as well as application of human excreta, sewage sludge, processing wastes, forestry residues or biochar. Including these sources would correct our estimates

upwards and bring our estimates closer to the IPCC stock change factors (see Sect. 3.4.1). Unfortunately, data on the quantity of these inputs is very scarce and does not exist with global coverage.

    SOC inputs from above-ground residues had the strongest management effect on SOC debt dynamics on cropland (see Fig. 5). As pointed out by Keel et al. (2017) and Smith et al. (2020), carbon input calculations are highly sensitive to the choice of allometric functions determining below- and above-ground residue estimates from harvested quantities (see A1 for coefficients

used in this study). Keel et al. (2017) question whether below-ground residues might increase with a fixed root:shoot ratio rather than being independent of productivity gains. Moreover, the study pointed out that plant breeding shifts allometries, which might not be reflected in outdated data sources. While our study considers a dynamic harvest index with rising yields for several crops, we may still overestimate residue biomass, in particular for below-ground biomass.

### 4.2 Reduced complexity SOC model

Our reduced-complexity SOC model is based on a Tier 2 modeling approach. This reduces the computational and data demand of the model in comparison to DGVMs, while still allowing for the explicit representation of agricultural management practices. Along with the effects of various C inputs, the impacts of water supply from rainfall and irrigation as well as tillage systems can also be accounted for in the computation of SOC decay rates. As such, the model can reflect the spatial and temporal heterogeneity in both management and biophysical conditions.

The substantial impact of changing management practices through time is indicated by the development of our estimated stock change factors (see Fig. 6) as well as by the time trend of the SOC debt (see Fig. 2(a)). Residue management has changed over the last decades, especially with the phasing out of residue burning practices in several regions and increased general productivity, showing a clear impact on SOC dynamics — underlining the importance to account for these effects in soil carbon modeling.

The Tier 2 approach (Ogle et al., 2019b) used here is explicitly designed for agricultural soils, whereas we apply it also to soils under PNV. This is necessary in order to represent SOC losses under land-use change in a dynamic way, as this is an important driver of SOC dynamics. The comparison of simulated PNV data with LPJmL4 shows the model's capability in reproducing PNV SOC stocks (Fig. A5). Concurrently, the point-data comparison (see Fig. A4) shows low correlation for PNV,

however also for cropland sites. This might also point to the fact that SOC stocks can vary at field and local scale considerably and thus a very high number of point data is needed to derive statistical properties that could improve the point-to-grid-cell comparison (see Rammig et al., 2018).f

Using litterfall estimates from LPJmL4, we have been able to estimate the total SOC stocks of the world, which is dominated by SOC under natural vegetation. However, as the world's SOC stock is highly uncertain, which is seen in the wide range of global SOC stock estimates for the first 30 cm of the soil profile (Batjes, 2016; Hengl et al., 2017; FAO, 2018; Schaphoff et al., 2018a; Poggio et al., 2021; Sanderman et al., 2017) in Fig. 7, the only conclusion we can draw from this is that our result is within a plausible range.

Additionally, we find our $0.2\,\mathrm{GtC\,yr^{-1}}$ SOC stock change within the period between 1975 and 2010 for the first 30 cm of the soil profile at the upper end of estimates comparing it to estimates of Ito et al. (2020) of $0.18 \pm 0.41\,\mathrm{GtC\,yr^{-1}}$ within the period between 1850 and 2014 for whole soil profile. This might be not surprising as the $CO_2$ effect is most likely stronger and land-use change effects weaker within our later and shorter modeling period compared to a mean value of the period between 1850 and 2014. The large uncertainty within estimates of SOC stock and its changes (Ito et al., 2020) again points to the large structural uncertainty within SOC modeling.

To avoid a strong impact of natural land representation and its uncertainties on our results, we focus on SOC changes on cropland. Pristine natural vegetated areas (like permafrost and rain forests) without human land management drop out in our calculation of SOC debt and stock change factors. Natural SOC estimates only influence results when natural land is converted to cropland. Moreover, the temporal dynamic of the SOC debt and stock change factors is not (or only to a small degree) altered by climatic or atmospheric effects on SOC stocks, as they cancel out by taking the difference (for the SOC debt) and ratio (for the stock change factors) of cropland SOC and SOC under hypothetical PNV conditions.

The initialization of SOC stocks, however, is important for the size of the SOC debt and its change over time, since the presence and size of legacy fluxes affect these values strongly (see Fig. A2). According to our sensitivity analysis the SOC debt varies between $33.3\,\mathrm{GtC}$ and $50.7\,\mathrm{GtC}$ depending on the initialization choice, with our best guess at $39.6\,\mathrm{GtC}$. Concurrently, our results indicate that the impact of the dynamic agricultural management is robust to the initialization of SOC stocks.

Comparing the geographic SOC patterns to Soil Grids 2.0 (Poggio et al., 2021) (see Fig. 9), we find that our model estimates values of SOC stocks greater than the estimated confidence intervals in Soil Grid 2.0 for some mountainous regions across the globe. This could indicate that we are not capable in capturing specific processes that would reduce the vegetation's productivity (such as erosion on steep slopes or shallow soils (Borrelli et al., 2017)). A large swathe of eastern North America was heavily affected by the dust bowl event, with wind erosion removing large parts of the topsoils, a process not considered in our model. Similarly, we likely overestimate SOC stocks for the loess soils in northern China and the Altiplano in Latin America; in both cases erosion is a likely reason. In contrast, we estimate lower SOC stocks at the edges of the Sahara, where uncertain local water availability and artifical irrigation may dominate spatial SOC patterns.

In our model, erosion might however only affect the spatial pattern but not the aggregate SOC pool. As pointed out by Doetterl et al. (2016), the final fate of leached or eroded carbon is uncertain and might even offset LUC emissions (Wang et al., 2017). Concurrently, other studies have claimed erosion moves SOC into aquatic reservoirs (Zhang et al., 2020) and

thus changing total global terrestrial SOC stock. Whereas for soil quality analysis SOC displacement might play an important role, in this budget approach focused especially on the SOC debt, displaced but not emitted SOC can be treated as SOC that remains on the cropland. Erosion and degradation impacts on yields and therefore on soil C inputs are captured by our method as we base them on FAO statistics of actual production. Yet the distribution of production below the country level - which we allocate proportional to LPJmL production potentials that do not reflect erosion feedback to yields - will overestimate yields and therefore biomass inputs to eroded soils.

In comparison with default stock change factors of the IPCC guidelines, our model estimates a stronger decline of SOC stocks (Fig. 6) for almost all regions. Tropical soils might suffer from low C input rates due to large yield gaps (Global Yield Gap and Water Productivity Atlas. Available URL: www.yieldgap.org (accessed on: 03/01/2022)) and high shares of residue removal and burning in lower-income countries (Smil, 1999a; Williams et al., 1997; Jain et al., 2014). Yet, even when comparing our estimates to the low-input stock change factors of the IPCC, our SOC loss is roughly twice as large as the revised 2019 IPCC default values (2019b), while it shows good agreement with the older default values from IPCC (2006). However, the revised estimates of the IPCC included much more and more recent data points, calling for a closer look on causes for the large deviations between our results and the refined Tier 1 factors. On the one hand, our approach does not account for unharvested carbon inputs from weeds or biomass cover on short-term fallows. Shifting agriculture with fallow periods might be prominent in tropical regions. While long-term fallow land (>4 years) is excluded from FAOSTAT as cropland, short-term fallow is not. Thus, our carbon inputs for these areas might be underestimated, leading to too low stock change factors. On the other hand, older studies by Don et al. (2011) estimated SOC losses for tropical soils of around 25% on average corresponding to a stock change factor of 0.75, but also reported a wide range of measured SOC changes from -80% to +58%. Fujisaki et al. (2015) however found much lower loss rates of around 9%, attributing the difference to the different time period lengths since the conversion to cropland. As our results do not specifically account for cropland age and most of the cropland is older than 20 years (as assumed for the default IPCC Tier 1 stock change factors) our stock change factors have to be lower by definition following the steady-state assumption that cropland will continue to approach a new equilibrium. For the same reason, our estimates for temperate regions might be lower than both IPCC (2006) and IPCC (2019b) default values. With the production-increasing impact of irrigation and fertilization on carbon-poor dryland soils, SOC under cropland can also be higher than under PNV with stock change factors above 1 (see Fig. 2(d)), but these areas are much smaller than where the stock change factors are well below unity.

Generally, limiting the analysis to the first 30 cm of the soil profile follows the IPCC guidelines (Eggleston et al., 2006; Calvo Buendia et al., 2019) and assumes that most of the SOC dynamics happen in the topsoil. In this regard several aspects are strongly simplified within our approach. Firstly, distribution of carbon inputs into different soil layers are neglected and all carbon inputs are allocated to the topsoil. This particularly overestimates SOC stocks in the first 30 cm of soil below deeper rooting vegetation, which is certainly the case for most of the woody natural vegetated areas. Second, changes to the subsoil are neglected, which is most important for tillage effects. Other management practices might be more equally effecting top- and subsoil as they do not directly the relocates carbon vertically. As Powlson et al. (2014) have shown, the subsoil can be make a large difference in evaluating total SOC losses or gains for no-tillage systems. No-tillage effects may seem larger than

they actually are if only topsoil is considered. SOC transfers to deeper soil layers under tillage might enhance subsoil SOC compared to no-till practices. However, the effect of no-till to the subsoil as well as its overall importance as a SCS measure is still debated (Ogle et al., 2019a). Finally, organic soils (like peat- and wetlands) and drained cropland areas are not explicitly considered and emissions from these cropland areas are thus likely substantially underestimated.

## 4.3 SOC debt and SOC drivers

The analysis of SOC stock gains and losses is complex and has several dimensions as climatic and anthropogenic effects overlap. There is broad consensus that land conversion to cropland has caused substantial C emissions over the historical period (e.g. Friedlingstein et al., 2020). There is uncertainty with respect to the overall size of these emissions from different methods and reference points and with respect to the contribution of cropland and agricultural management to these emissions. In order to mitigate greenhouse gas emissions, it is essential to stop the decline of SOC stocks or even transform cropland management to sequester atmospheric C in cropland soils (Minasny et al., 2017). Defining the SOC debt of 1975 as the baseline, and measuring land-use emissions on cropland as the difference between a potential natural state and the state under human interventions (see Pugh et al., 2015), we find that global cropland has acted as a emissions source since 1975. Comparing our SOC loss rate (the change of SOC debt) of $0.14\,\mathrm{GtC\,yr^{-1}}$ to estimates on land-use change induced emissions of $2.0 \pm 1.0\,\mathrm{GtC\,yr^{-1}}$ (sum of 'Bookkeeping LULCC emissions' and 'Loss of additional sink capacity' for the years 2009–2018 in Gasser et al. (2020)), we find SOC emissions of the first 30 cm of the soil profile to be a minor contributor to overall land-use change induced emissions. Annual C loss rates of 0.2 per 1000 C still have the opposite trend as the promoted 4 per 1000 C sequestration rate target (Minasny et al., 2017). Dedicated efforts to increase cropland SOC are thus necessary, as management improvements at historical rates are not enough to counteract ongoing SOC degradation on cropland. Yet our study also shows the substantial impact of changing management on the development of SOC debt (Fig. 3 and Fig. 5).

According to Sanderman et al. (2017), the SOC debt since the beginning of human cropping activities has been at around 37 GtC for the first 30 cm of the soil with half of it attributed to SOC depletion on grasslands. Our estimate of 39.6 GtC in 2010 for cropland debt is thus twice as high as their estimate. However, there are large uncertainties in modeling SOC at the global scale, and Sanderman et al. (2017) pointed out that their results might be conservatively low compared to experimental results.

Furthermore, Sanderman et al. (2017) modeled historical trends based on agricultural land expansion without considering SOC variations due to time-variant agricultural management. Pugh et al. (2015) considered management effects like tillage and incorporation of residues in stylized and static scenarios only, so that they could not account for historical management effects on SOC dynamics. Their study moreover concludes that yield gains (by 18% in their simulations) do not lead to a substantial decline in SOC debt (less than 1% SOC increase). Historical yield increases, however, are estimated to be well above 50% (Pellegrini and Fernández, 2018; Ray et al., 2012; Rudel et al., 2009) and often lead to an increase in below- and above-ground residue biomass inputs to the soil. While we find substantially larger SOC increase in response to productivity gains than the 1% reported by Pugh et al. (2015), this is not sufficient to compensate SOC losses from global cropland expansion of around 11% between 1974 and 2010.

The effects of agricultural productivity on cropland SOC dynamics, including historical yield trends and associated increases in residue inputs, can be directly accounted for in our modeling approach. In contrast, process-based studies (Pugh et al., 2015; Herzfeld et al., 2021) often lack data on relevant management aspects that drive production increases. Herzfeld et al. (2021) also consider historical management trends for fertilizer and manure inputs as well as on residue removal rates and tillage systems, but cannot reproduce the substantial increase in agricultural productivity over the last decades. Still, they find that compared to no-tillage systems, residue management has much larger potential to affect the strength of their projected future global cropland SOC decline. This is consistent with our finding that increasing SOC inputs from above-ground residues had the strongest effect on the slowing-down of the SOC debt increase (Fig. 5). In line with this, Dangal et al. (2022) finds that no-tillage has only minor impacts on SOC dynamics across parts of the US.

Elliott et al. (2018) show that yield trends in the USA can be reproduced by models, but require information on inputs that are not available at the global scale, such as annual data on sowing dates, planting densities, and genetic traits such as kernel number and radiation use efficiency. As such, it will remain challenging for process-based DGVMs to capture the trend of agricultural productivity on cropland SOC dynamics.

Our study emphasizes again that the expansion of cropland is still a major source of $CO_2$ emissions — not only through the removal of vegetation, but also by a slow depletion of C stocks in soils. Our estimates indicate a SOC debt of 39.6 GtC in 2010, and every additional deforestated hectare adds to this debt. Avoided deforestation and other environmental regulation leads to intensification on existing cropland (Humpenöder et al., 2018) and our results show that such intensification could lead to increased cropland SOC, if residues are returned to the soil — amplifying the C sequestration potential of avoided deforestation.

There is also ample potential for further improved SOC management. As shown in Fig. 4, the annual SOC respiration (1.3 GtC per year) is slightly above one quarter of total annual net C uptake by crops (4.6 GtC per year). C compounds have to be respired by soil organisms to maintain basic soil functions and regulate the nutrient cycle, which often leaves limited options to decrease C losses via SOC respiration (Janzen, 2006). However, similar C losses occur at the end of the food supply chain (1.2 GtC per year), at the soil surface (1.5 GtC per year), and smaller but still considerable during residue burning (0.2 GtC per year) and within animal waste management systems (0.2 GtC per year). Improved management could include, firstly, a circular flow from the food supply chain back to soils. Waste composting or excreta recycling could represent a major additional C input to cropland soils (Brenzinger et al., 2018). Secondly, soil carbon sequestration techniques (Smith, 2016), deep ploughing (Alcántara et al., 2016) or the transformation of C inputs to more recalcitrant biochar (Woolf et al., 2010) may transfer larger parts of the biomass at the litter soil barrier into permanent soil pools. Thirdly, reducing the share of residue burning and improved manure recycling could further increase C inputs. Finally, other carbon-accumulating practices, such as the cultivation of cover crops (Poeplau and Don, 2015; Porwollik et al., 2022) and agroforestry (Lorenz and Lal, 2014) could increase total C sequestration on cropland.

## 5    Conclusions

We have compiled a spatially explicit and time-variant data set on agricultural management aspects relevant for cropland SOC dynamics. We have also developed a reduced-complexity SOC model that is able to be applied in optimization-based IAM frameworks, for which detailed process-based models are computationally too expensive. Making use of these data and model, we are able to estimate spatially explicit SOC stocks, SOC debts, and stock change factors considering agricultural management. It is — to our knowledge — the first study that analyzes the role of time-variant and spatially explicit historical

agricultural management for global SOC dynamics.

Our results demonstrate that historical changes in agricultural management have shaped the SOC debt on cropland. It is thus necessary to explicitly consider agricultural management in a dynamic manner in global carbon assessments and models, especially when exploring climate mitigation pathways with so-called land-based solutions (e.g. Popp et al., 2016). That also implies that we need better monitoring of agricultural practices to create this data, but also better accessibility of existing data.

Our open-source model (Karstens and Dietrich, 2020), published data-set (Karstens, 2020a) and the flexible data processing with the MADRaT package (Dietrich et al., 2020) constitute a starting point for building comprehensive data sets on agricultural management aspects.

With the reduced-complexity SOC model we are able to account for agricultural management effects on cropland SOC dynamics within optimization-based IAM frameworks. Reduced input data requirements such as accounting for changes in

productivity rather than reproducing the processes that lead to such changes in productivity (Elliott et al., 2018) will help to explore the role of agricultural management in sustainable development pathway analyses (Sörgel et al., 2021). However, we clearly see that increases in agricultural productivity are not sufficient to create positive net SOC sequestration in cropland soils. More management options that explicitly target the sequestration of C in cropland soils need to be considered. Our open-source model can be expanded to account for additional management options for carbon farming, such as cover crops, agroforestry,

or biochar applications.

*Code and data availability.* We compile calculations as open-source R packages available at github.com/pik-piam/mrcommons (Bodirsky et al., 2020a) for the management related functions, github.com/pik-piam/mrsoil (Karstens and Dietrich, 2020) for soil dynamic related functions and github.com/pik-piam/mrvalidation (Bodirsky et al., 2020b) for validation data. All libraries are based on the MADRaT package at github.com/pik-piam/madrat (Dietrich et al., 2020), a framework which aims to improve reproducibility and transparency in data processing.
Model results including C input data are accessable under https://doi.org/10.5281/zenodo.4320663 (Karstens, 2020a). Software code for paper and result prepartion can be found under www.github.com/k4rst3ns/historicalsocmanegement.

## Appendix A:  Figures and tables in appendices

## A1    Methods

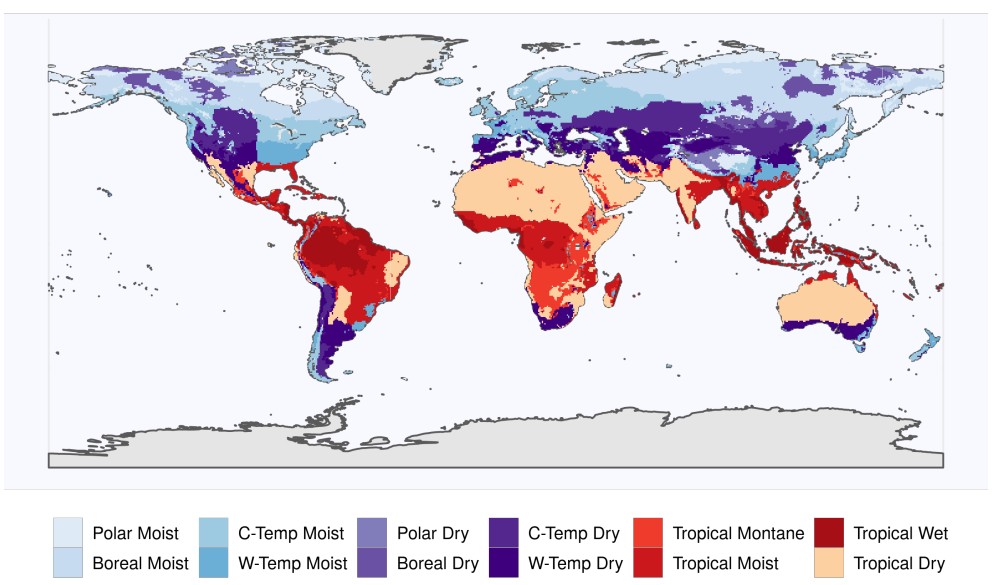

Polar Moist      C-Temp Moist      Polar Dry      C-Temp Dry      Tropical Montane      Tropical Wet

Boreal Moist      W-Temp Moist      Boreal Dry      W-Temp Dry      Tropical Moist      Tropical Dry

**Figure A1.** Climate zone map adpated from IPCC: The climate zone classification is based on the classification scheme of the IPCC guidelines (Eggleston et al., 2006) and has been reimplented by Carre et al. (2010), which is the source of this data. Note that the reduced set, used for the comparison of stock change factors is included in the color code with temperate moist in light blue, temperate dry in dark violett, tropical moist in red and tropical dry in orange.

**Table A1.** Parameterization of harvested organs and their corresponding residues parts as well as allometric coefficients: This table is mainly based on Bodirsky et al. (2012) together with simple carbon to dry matter assumptions. Allometric coefficients are used as descriped in Eggleston et al. (2006) with $HI^{prod}$ being $slope_{(T)}$, $HI^{area}$ $intercept_{(T)}$ and RS $R_{BG-BIO}$.

| Crop code | Crop Type | Harvested Organs | | | Above-ground Residues | | | Below-ground Residues | | Allometric coefficients | | |
|---|---|---|---|---|---|---|---|---|---|---|---|---|
| | | nr/dm | wm/dm | c/dm | nr/dm | wm/dm | c/dm | nr/dm | c/dm | $HI^{area}$ | $HI^{prod}$ | RS |
| tece | Temperate cereals | 0.0217 | 1.14 | 0.42 | 0.0074 | 1.11 | 0.42 | 0.0098 | 0.38 | 0.58 | 1.36 | 0.24 |
| maiz | Maize | 0.016 | 1.14 | 0.42 | 0.0088 | 1.18 | 0.42 | 0.007 | 0.38 | 0.61 | 1.03 | 0.22 |
| trce | Tropical cereals | 0.0163 | 1.14 | 0.42 | 0.007 | 1.18 | 0.42 | 0.006 | 0.38 | 0.79 | 1.06 | 0.22 |
| rice_pro | Rice | 0.0128 | 1.15 | 0.42 | 0.007 | 1.11 | 0.42 | 0.009 | 0.38 | 2.46 | 0.95 | 0.16 |
| soybean | Soybean | 0.0629 | 1.13 | 0.42 | 0.008 | 1.11 | 0.42 | 0.008 | 0.38 | 1.35 | 0.93 | 0.19 |
| rapeseed | Other oil crops (incl rapeseed) | 0.0334 | 1.08 | 0.42 | 0.0081 | 1.11 | 0.42 | 0.0081 | 0.38 | 0 | 1.86 | 0.22 |
| groundnut | Groundnuts | 0.0299 | 1.06 | 0.42 | 0.0224 | 1.11 | 0.42 | 0.008 | 0.38 | 1.54 | 1.07 | 0.19 |
| sunflower | Sunflower | 0.0216 | 1.08 | 0.42 | 0.008 | 1.11 | 0.42 | 0.008 | 0.38 | 0 | 1.86 | 0.22 |
| oilpalm | Oilpalms | 0.0027 | 1.01 | 0.49 | 0.0052 | 1.11 | 0.48 | 0.0053 | 0.47 | 0 | 1.86 | 0.24 |
| puls_pro | Pulses | 0.0421 | 1.1 | 0.42 | 0.0105 | 1.16 | 0.42 | 0.008 | 0.38 | 0.79 | 0.89 | 0.19 |
| potato | Potatoes | 0.0144 | 4.55 | 0.42 | 0.0133 | 6.67 | 0.42 | 0.014 | 0.38 | 1.06 | 0.1 | 0.2 |
| cassav_sp | Tropical roots | 0.0053 | 2.95 | 0.42 | 0.0101 | 6.67 | 0.42 | 0.014 | 0.38 | 0 | 0.85 | 0.2 |
| sugr_cane | Sugar beet | 0.0024 | 3.7 | 0.42 | 0.008 | 3.82 | 0.42 | 0.008 | 0.38 | 0 | 0.67 | 0.07 |
| sugr_beet | Sugar beet | 0.0056 | 4.17 | 0.42 | 0.0176 | 5 | 0.42 | 0.014 | 0.38 | 0 | 0.54 | 0.2 |
| others | Fruits, Vegetables, Nuts | 0.0267 | 5.49 | 0.42 | 0.0081 | 1.88 | 0.42 | 0.007 | 0.38 | 0 | 0.39 | 0.22 |
| foddr | Forage | 0.0201 | 4.29 | 0.42 | 0.0192 | 4.1 | 0.42 | 0.0141 | 0.38 | 0 | 0.28 | 0.45 |
| cottn_pro | Cotton seed | 0.0365 | 1.09 | 0.42 | 0.0093 | 1.18 | 0.42 | 0.007 | 0.38 | 0 | 1.48 | 0.13 |

nr/dm – nitrogen to dry matter ratio

wm/dm – wet matter to dry matter ratio

c/dm – carbon to dry matter ratio

$HI^{area}$ – harvest index per area

$HI^{prod}$ – harvest index per production

RS – root:shoot ratio

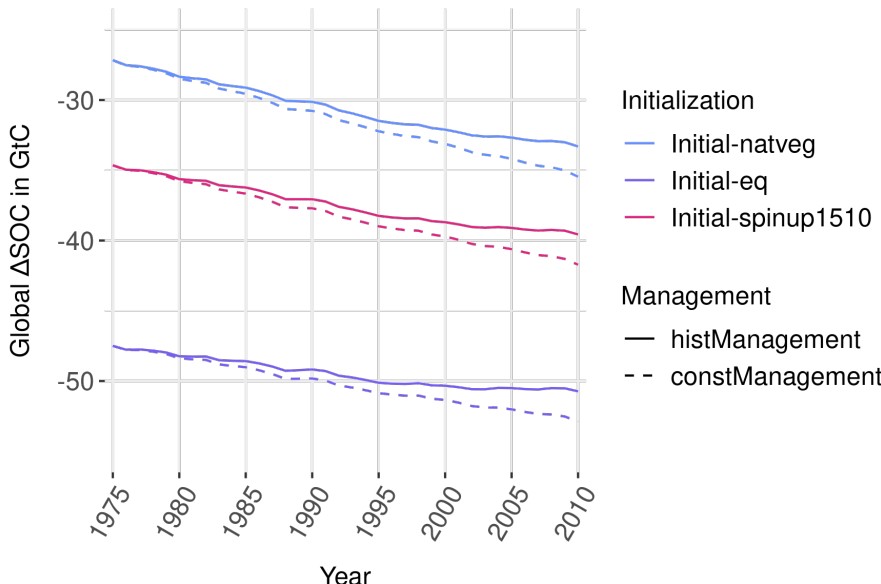

**Figure A2.** Global $\Delta SOC$ for different SOC initialization choices in the start year 1901: Starting in 1901 with steady-state SOC under vegetation for all land-use types without any human cropping activities (Initial-natveg) lead to a smaller $\Delta SOC$ in 1975 and a steeper increase till 2010, as compared to initialzing with steady-state SOC stocks under historic land-use (Initial-spinup1510). On the other hand, assuming all SOC to be in land-use specific steady-state already in 1901 (Initial-eq) leads to the opposite effect of an already higher $\Delta SOC$ in 1975 and a less steeper increase till 2010. However, the difference between the dynamic historical management assumption (histManagement) compared to constant management assumption from 1975 on (constManagement) are of a similar size for all initialization choices.

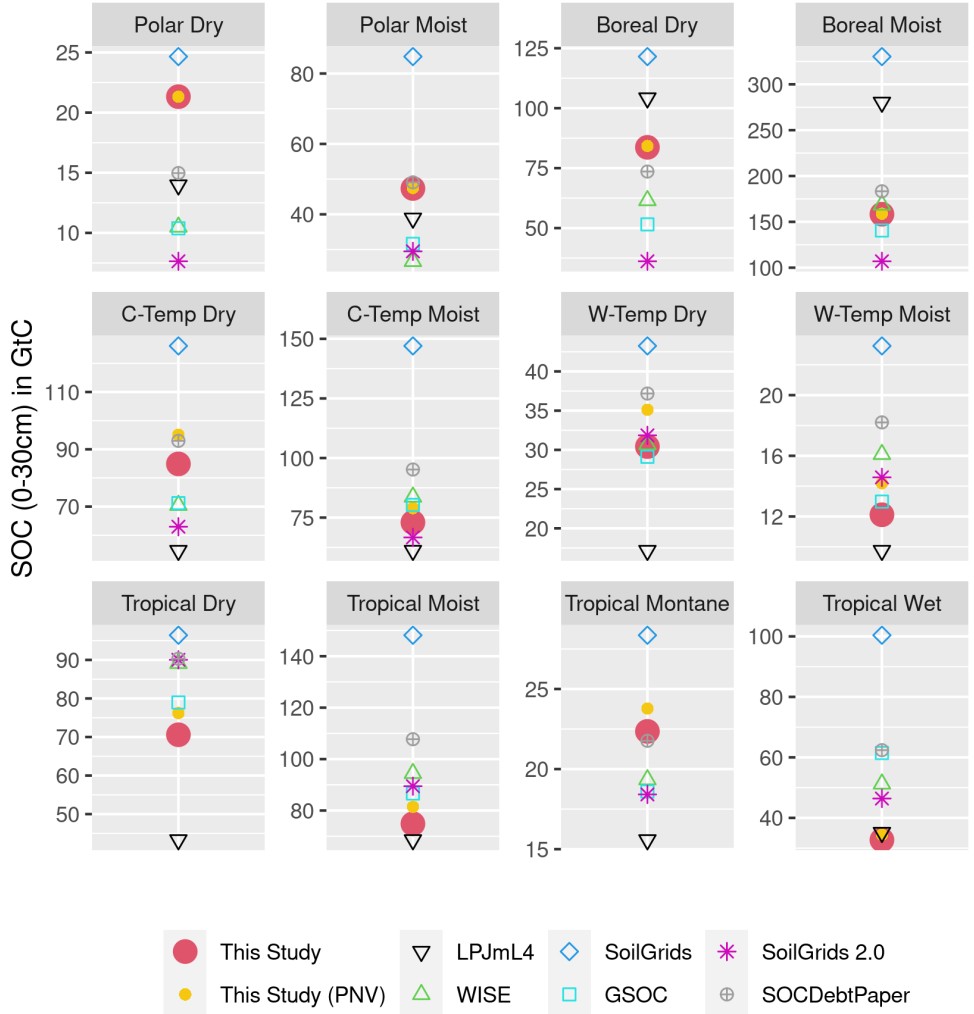

**Figure A3.** Modelled as well as data based estimation for climate zone specific SOC stock in GtC for the first 30 cm of soil aggregated over all land area: SoilGrids, GSOC and WISE do not consider changes over time and rely on soil profile data gather over a long period of time, which makes it hard to pinpoint a specific year to these SOC estimations. In this context they will be compared to modelled data (LPJmL4, this study) for the year 2010. PNV denotes the potential natural vegetation state without considering human cropping activities, calculated as reference stock within our model. We use the climate zone specification of the IPCC (Eggleston et al., 2006).

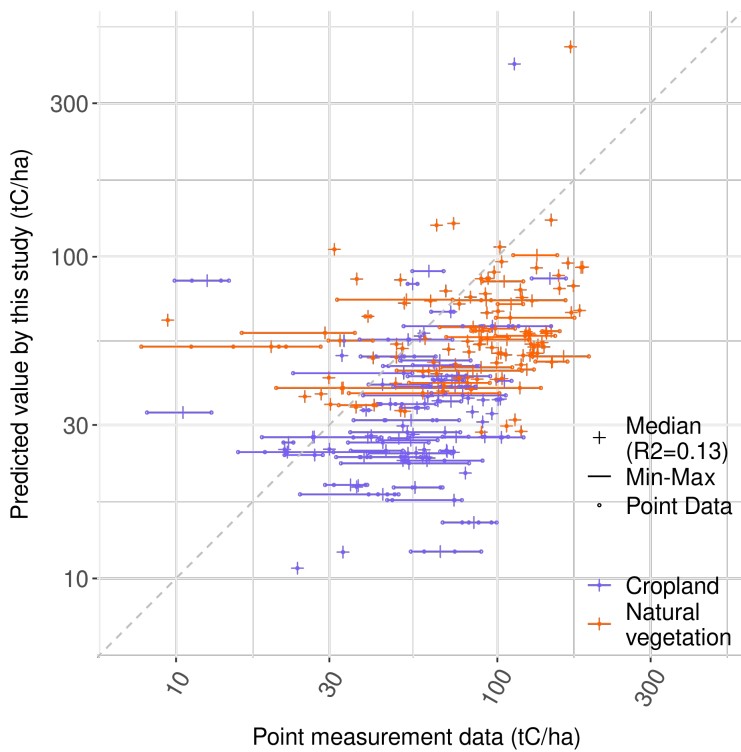

**Figure A4.** Correlation between modeled and measured $SOC$ stocks. Given the wide span between minimun and maximum measured $SOC$ stocks within in a given cell, we correlated median values with our modeled results. Both cropland ($R2 = 0.09$) and areas with natural vegetation ($R2 = 0.08$) tend to be lower in our results than in the point measurements.

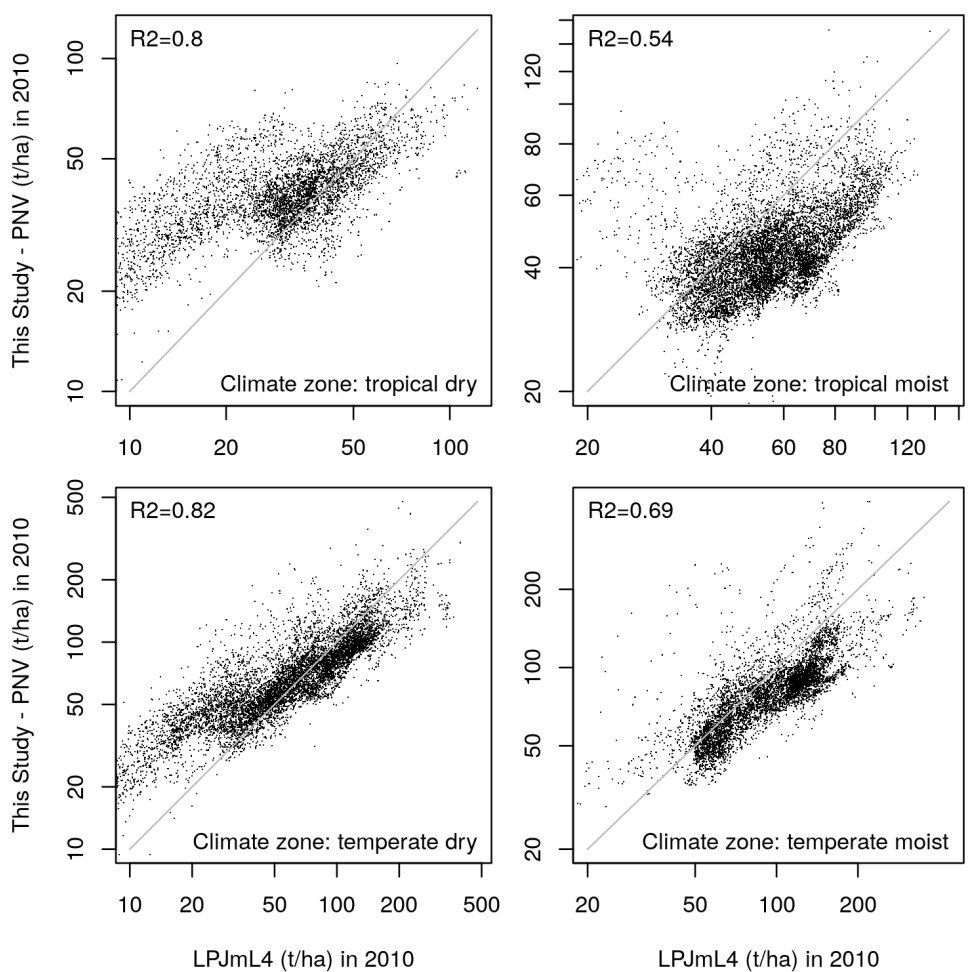

**Figure A5.** Correlation between modeled SOC stocks of LPJmL4 and this study for an hypothetical potential natural state (PNV) for the year 2010. The grey lines indicate the 1:1 line.

*Author contributions.* KK, BLB and AP designed the study and the model idea. KK wrote the code build on work of BLB, IW. JPD revised and improved the model code. CM, JH and SR provided the LPJmL simulation data. KK wrote the paper with important contributions of BLB and CM. MK, JS, SR and IW provided extensive feedback to outline of the study. All authors discussed the results and commented on the manuscript.

*Competing interests.* The authors declare no competing interests.

*Acknowledgements.* Thanks to Vera Porwollik for contributing the time resolved tillage data set based on her previous work. Additional thanks to the rticles contributors for providing a R Markdown template. The authors thank for the data provided by FAOSTAT and LUH2v2. The work of KK was funded by the DFG Priority Program "Climate Engineering: Risks, Challenges, Opportunities?" (SPP 1689) and specifically the CEMICS2 project (grant no. ED78/3-2). The research leading to these results has received funding for BLB from the European Union's Horizon 2020 research and innovation program under grant agreement no. 776479 (COACCH) and no. 821010 (CASCADES). The 675 work of SR, JS and IW was also supported by CLIMASTEPPE (01DJ8012), EXIMO (01LP1903D) and FOCUS (031B0787B) funded by the German Federal Ministry of Education and Research (BMBF). The input of PS, MK and MD contributes to the Soils-R-GGREAT project (NE/P019455/1) and CIRCASA (EU H2020; grant agreement no. 774378).

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
