# Peer review of "Management induced changes of soil organic carbon on global croplands"

_Biogeosciences, 2020_

## Referee Comment (RC1) · Jonathan Sanderman (Referee) · 10 Jan 2021

This MS presents the findings of a spatially explicit implementation of a new IPCC Tier II approach to soil carbon stock changes. The authors use this model to calculate how much SOC has been lost due to agriculture but then go on to run the model annually for the period 1975 – 2010 to produce a dynamic picture of SOC recovery over this modern era of farming. The major takeaway message is that while agriculture has incurred a large SOC debt, recent agronomic improvements have led to 4 Pg C of SOC sequestration over this period of time. This detailed picture of SOC in croplands over the past several decades is of incredible importance to policy makers and as such I believe that paper can be an important contribution to the literature; however, I do have several major concerns that may or may not be addressable with revisions.

[Figure]

Methodological concerns.

I had to read the methods section twice and spend an hour with Calvo Buendia et al. 2019 to fully understand what the authors have done. I'm still not 100% confident that I fully understand the methodology. I suggest adding an illustrative example or two graphically demonstrating how the process works. Perhaps starting with a simple case of one lu transition and then showing a more complex case of multiple lu transitions within a pixel.

I have not been convinced that this sort of "dynamic" implementation of a steady state modeling approach is appropriate. I understand that the method was developed by the IPCC as a way to add more nuance into the Tier I emission factor approach but I don't think the method was intended to be applied annually. Why not go all the way to Tier III approach using the process-based dynamics that are embedded in this simplified model? It appears you have all the data assembled to do this. My main concern with applying a steady-state model to annual changes is we know that the recent past trajectory of SOC (particularly in the slow and passive pools) will greatly influence the short-term model response to improved management – i.e. the model will take years to decades before SOC stocks start to rebuild if the trajectory was negative prior to the change – but this will be completely missed with the steady state application (stocks will start increasing immediately upon change).

What time frame are the monthly climate data averaged over to get the rate modifiers for a steady state solution? Given the passive pool has an intrinsic decay rate equivalent to >100 year turnover time, it seems that you need to have a 100+ year average climate to come up with the proper rate modifiers.

Transfer between lu types is not clear. I do not understand how a "respective share of the SOC is reallocated." My concern is that the per area SOC stock for long-term cultivated land will be much different than the per area SOC stock for newly converted cropland, so I don't see how you can suddenly bin these separate areas into one model

component. Perhaps my request for a visual guide will help me (and other readers) understand that I have misunderstood this part of the methods.

Why was 1901 chosen to spin up to steady state? We know that this was a time of rapid agricultural expansion in several major regions of the world and thus a time of rapid soil carbon loss.

LULUC data – are these data all provided as area within each grid cell (or percent of a grid)? I think so, but please indicate. A supplemental table with the cross-walk between the LUH2 and the 17 crop groups used in this study should be included.

Please include units for eq 9 – I don't follow the AGR calculation – it sounds like you are adding biomass and area together. Additionally, HI is usually defined as CP divided by total aboveground biomass, so CP x HI is a meaningless number.

Lack of validation.

There appears to be no attempt to validate the model or the input data used to drive the model. In general, there is a lack of quantitative evaluation throughout. There are just two qualitative quality assessments – a table comparing calculated stock change factors to Tier I estimates and a discussion on how the map looks similar to other SOC maps.

The model itself was developed recently as part of the 2019 Refinements to IPCC Guidelines and those updated guidelines discuss how the model was calibrated to a set of long-term trial sites but do not report any model performance metrics. As pointed out by the authors some areas of central EU and the UK more than double SOC under current agriculture than under native vegetation. This is certainly indicative that there should be some checks against real data (see detailed elaboration on this point further down in this review). It could be argued that point-based validation for a model run at 0.5 degree resolution is meaningless but it would be an interesting exercise to see how the model reproduces trends with known SOC histories.

Issues with residue C return.

Given that the major takeaway from this paper is that the SOC is being sequestered due to improved yield leading to increased residue return, is there any empirical evidence that C inputs to the mineral soil have nearly doubled (Fig 3)? I think the method for calculating residue return to the soil is potentially flawed leading to this large apparent increase. The authors have assumed that both harvest index (HI) and root-to-shoot (RS) ratios have been constant through time. However, yield improvements over the last century, and in particular the last 50 years, are a result of improvements in genetics and nutrition. Breeding has resulted in the ability to plant most crops at much higher densities and selection towards more photosynthate being allocated to harvestable organs. Both of these improvements have altered HI and RS ratios. Additionally, there are strong interactions between N fertilization rate and root density. There is a huge literature on crop breeding that support the non-stationarity of these important parameters.

Other specific comments.

Units – Gt and Mt are not SI units, please use Pg and Tg

L13 (and elsewhere) – "we also find that SOC is very sensitive. . ." – this is in reference to an unvalidated model result. I'd suggest rewording these sorts of phrases to, "Our model results suggest that SOC is very sensitive. . ."

L279 – "we provide the first world map" – no, you did not. All of the global maps that have been developed using a statistical environmental-covariate modeling approach (i.e. soilgrids and similar) implicitly include all historic land management.

It is great that all the data are provided but I found the Karstens 2020a repository to be confusing. Can you have a description for each file in the repository? The naming convention is not clear. I did not want to download 9 Gb of data to figure it out.

Fig 1 – perhaps it is just the spatial scale of these small maps (and I haven't down-

**BGD**

Interactive
comment
[Figure]

loaded the results to explore in more detail) but it looks like there is zero intact forest in the Congo Basin and very little intact forest in the Amazon. Also, I would have liked to see a map showing the trend in SOC spatially – how are the 4 Pg C that has been sequestered been spread across the globe? Is it all in Central EU and UK?

Fig 2 – this is a really interesting way of summarizing the model results.

L313-314 – This sentence ("global SOC would still increase") is confusing as the graph shows a loss in the constResidue scenario.

Fig 4 – the finding presented here is very counterintuitive to me. Why is the SOC debt halved when the model is initialized with natural vegetation? Shouldn't the 1975 SOC debt be much greater if the 1901 starting point was natural vegetation instead of actual land use? Perhaps I am just misunderstanding this sensitivity analysis.

Discussion section – in general, there is very little discussion of how these results fit into the large literature on SOC. There are many places were a reference or two would great increase the credibility of the statements that are being made.

Section 4.2 – I think this section should come right before the conclusions especially as you refer to analysis that is only presented for the first time in section 4.3

L358-360 – the finding that northern temperate zones (particularly in EU and UK) now have SOC levels up to twice that of native state yet tropical soils have lost 40-70% of their SOC is problematic and, as the authors point out in relation to the EU example, likely points to issues with getting C input to soil correct. The EU has the perfect data set to test this model finding – the EU JRC LUCAS survey was a balanced sampling design between forested and agricultural land uses. In the tropics, it has been fairly well documented that already infertile tropical soils do not lose nearly as much SOC as their fertile temperate zone soil counterparts. While there are issues and large scale generalizations in the IPCC Tier I default factors, they do represent the consensus literature on the topic. The updated meta-analysis between the 2006 and 2019 IPCC

guidelines when this emission factor for the tropics changed dramatically (see Table 4 in this MS) points to this new knowledge.

L396 – how is this validation? It is just a comparison.

L400-406 – there is a large literature that can be drawn upon to support some of the claims made in this section.

Section 4.4 – I do not think this is a valid comparison because SoilGrids explicitly tried to capture high carbon density soils well while your model explicitly excludes organic soils. I suggest applying an agriculture mask to all of these datasets and then redo the analysis. Additionally, ISRIC released an update to SoilGrids >6 months ago that focuses primarily on mineral soil carbon stocks. This update is probably a better comparison.

L425-428 – this seems out of place.

L453 – comparison to 4p1000 is not really fair because your model is really just the business-as-usual scenario with SOC gains simply because yields are improving globally. 4p1000 is about intentional management shifts to increase SOC.

---

## Author Comment (AC1) · 26 Jan 2021

Dear Dr. Jonathan Sanderman,

thank you very much for you valuable feedback. We already started to improve model and paper draft based on your comments. On your methodological concerns about our "dynamic" implementation of a steady-state modeling approach, we want to provide an immediate clarification to not risk confusion by the other reviewers.

In your review, you stated "My main concern with applying a steady-state model to annual changes is we know that the recent past trajectory of SOC (particularly in the slow and passive pools) will greatly influence the short-term model response to improved management – i.e. the model will take years to decades before SOC stocks start to

rebuild if the trajectory was negative prior to the change – but this will be completely missed with the steady state application (stocks will start increasing immediately upon change)."

We would like to clarify that the Tier 2 modeling approach within the refinement of the IPCC Guidelines of 2019 is a first order kinetic approach. It was just named by the IPCC authors as "steady-state method", since it is based on steady-state calculation as an intermediate step. Yet, it does not assume SOC stocks in immediate steady state, and does account - depending on the pool type - for longer than annual transition phase to new steady states. We share your confusion about the naming within the guidelines, and will clarify the naming within the revisions.

We will provide a detailed point-by-point response to all your comments soon.

---

## Referee Comment (RC2) · Anonymous Referee #2 · 16 Mar 2021

The authors have conducted a study evaluating the influence of management on soil organic carbon in global croplands. This is an important topic for consideration of greenhouse gas mitigation with natural solutions for climate change policy and programs such as the 4 per mille initiative. As the authors mention, there are few studies that have evaluated cropland management effects on soil organic carbon, and possibly none that have addressed the influence at a global scale. The result that increased residue return to soils is the leading driver of carbon changes over the past few decades in croplands is an important finding. As the authors note, the 4 Gt C increase in carbon is less than the goals of the 4 per mille initiative, which some have argued is not realistic. I have a few concerns about the study after review of the IPCC documentation on the method that the authors selected for this analysis. I would suggest that the authors

make revisions before the manuscript is accepted.

Specific comments:

1) The Tier 2 method is in a croplands chapter of the IPCC report. The documentation in the report states that the model would need to be parameterized for other land uses. Did the authors parameterize the model for other land uses that would be considered natural vegetation? If not, the estimation of soil organic carbon for natural vegetation may not be valid. The authors seem to suggest that this is a possibility in Section 4.4 when stating the soil organic carbon and debt from land use change have to be interpreted with caution. If the model has not been parameterized for natural vegetation, I would suggest that the authors focus on cropland model results, and remove the carbon debt results. The results for the cropland alone are important, and deserve publication even if the natural vegetation estimates are not valid with this model.

2) Is it possible to estimate uncertainty with this method? IPCC methods often have large uncertainty but does this method have less uncertainty because it is a Tier 2 method. If it is not possible to estimate uncertainty could the authors speculate on the level of uncertainty in the predictions. Knowing something about uncertainty would be helpful in comparisons to the modeled results from other studies that are shown in the manuscript.

3) The authors state that a sensitivity analysis presented in Figure 4 shows that management impact is robust to the initialization of the soil organic carbon stocks at the beginning of the spin-up phase. But, the stocks and change in stocks almost halves the values if the initialization is done with natural vegetation. The initialization does make a difference, and needs further explanation.

4) Good to see that the authors have made a comparison to another approach to confirm the Tier 2 results. The Tier 1 method provided by the IPCC has been used for this purpose. In section 2.2, the authors present a method estimating stock change factors instead of soil organic carbon changes. But, the results in Table 4 for the stock change

factors are not convincing that the methods are consistent, and the text seems unclear with discussion about larger differences with the IPCC2019 values, which were updated by the authors and should be more accurate – I would think. Why not estimate the change in soil organic carbon for a direct comparison with the Tier 2 method instead of the stock change factors? Also, the placement of these results after the discussion seems odd, and conventionally would be presented in the results section before discussion.

5) Figure 3 shows results from making certain practices constant from 1975 to 2010. The authors state around line 315 that the effect of no-till has been strong since 1990, but the effect seems minor and may not differ statistically from the histManagement with uncertainty. The conclusion about the importance of residue seems most important here.

6) The authors suggest that there needs to be a circular flow with food supply chain back to soils. They assumed that none of the waste from supply chains are returned to soils (near line 300) but this seems incorrect. Municipal waste and materials are amended to soils in many regions of the world although maybe there are no data on these amendments? If this is the issue, the authors could mention that they are making a conservative assumption due to lack of data.

7) The authors evaluate the sensitivity of the Tier 2 model for tree litter with methods in Section 2.4.3. The Tier 2 model divides litter into metabolic and structural components, and the authors have averaged lignin to nitrogen across tree components as input to the model. But forest also include deadwood and should be separated from other forest litter to model decomposition. Did the authors add a deadwood pool? I question if this model is appropriate for forest if deadwood is not modeled separately.

8) Recommend that the authors provide more explanation for Equation 9, which determines the residue amount of C, and is a key driver of the carbon change. Harvest index is the proportion of plant biomass that is harvested, but the authors are multiplying the

harvested crop product by the harvest index. But the conventional approach is 'harvested crop production divided by the harvest index' to determine the total biomass and then subtract the harvested amount to estimate the residue. The authors are accounting for double harvesting and fallow in this calculation, which I agree is important, but some further explanation is needed about the calculation to understand how residue carbon is estimate from crop production, harvest index and area.

9) For the Tier 1 method, IPCC divides the reference carbon stocks by climate and soil types. Did the authors also divide the grid cells by climate and soil because only climate is mentioned in the text? And, I found a diagram in Figure 5.1 in the IPCC report that divides low, medium and high input categories. Did the authors use this diagram to classify the input? It is not clear if the authors use the diagram or developed their own. If they developed their own, is it consistent with the IPCC factors?

10) What is 'resp' is 'area reduction resp' on line 110? This sentence should be revised to improve readability. I also found other sentences that were difficult to read or missing words in some cases, but did not make a list during my review. Suggest a careful review before final publication.

11) I found the Tier 2 method in Chapter 5 of Volume 4 of the 2019 IPCC report, and would suggest that the authors cite this chapter rather than the entire 2019 IPCC report, which has 5 volumes. This would make it easier for others interested in the study to find the method in the IPCC report.

---

## Author Comment (AC2) · 8 Jun 2021

**Preface: The authors have conducted a study evaluating the influence of management on soil organic carbon in global croplands. This is an important topic for consideration of greenhouse gas mitigation with natural solutions for climate change policy and pro-grams such as the 4 per mille initiative. As the authors mention, there are few studies that have evaluated cropland management effects on soil organic carbon, and possibly none that have addressed the influence at a global scale. The result that increased residue return to soils is the leading driver of carbon changes over the past few decades in croplands is an important finding. As the authors note, the 4 Gt C increase in carbon is less than the goals**

**of the 4 per mille initiative, which some have argued is not realistic. I have a few concerns about the study after review of the IPCC documentation on the method that the authors selected for this analysis. I would suggest that the authors make revisions before the manuscript is accepted.**

Answer to preface.: Dear reviewer, thank you for the thorough and helpful review of our manuscript. While checking and revising our processing in response to your and the other reviewer's feedback, we discovered a bug in the soil model, leading to an overestimation of the transfer of carbon from active to the slow pool exclusively for cropland. Additionally, we found unreasonable high forage crop production values (specifically for pumpkins used as fodder) in our input data, which were taken from FAO statistics. This made the overall intensification trend in agriculture lead to increasing carbon stocks in cropland. After correcting the bugs, this is no longer the case. Whereas this implies major revisions of our discussion and interpretation of results, we argue that the essence of the paper remains unchanged, albeit modified. We suggest that our key findings are a) we introduce a soil carbon model that can account for changes in agricultural management and can be applied within integrated assessment frameworks for the first time and b) we show that it is critical to account for management dynamics in SOC assessments. We provide an assessment of how results changed after correcting the bugs as a supplement to this author's comment. Here, we respond point-by-point to the reviewers' comments. Where appropriate, this will also address the implications of the bug fix. We look forward to your response.

**1. The Tier 2 method is in a croplands chapter of the IPCC report. The documentation in the report states that the model would need to be parameterized for other land uses.Did the authors parameterize the model for other land uses that would be considered natural vegetation? If not, the estimation of soil organic carbon for natural vegetation may not be valid. The authors seem to suggest that this is a possibility in Section 4.4 when stating the soil organic carbon and debt from land use change have to be interpreted with caution. If the model has**

**not been parameterized for natural vegetation, I would suggest that the authors focus on cropland model results, and remove the carbon debt results. The results for the cropland alone are important, and deserve publication even if the natural vegetation estimates are not valid with this model.**

Answer to 1.: This is an important point. While our analysis focuses on croplands, estimating the natural soil carbon stocks is necessary to account for the C entering the cropland budget via land conversion. Apart from this process, we only use natural vegetation SOC to make our results comparable to other global estimates for validation purposes. To improve the parametrization of natural soil carbon, we will include the following model updates: * We will improve the litterfall parameterization in natural vegetation. * We will compare our results on soil stock under natural vegetation with the results of a model parametrized for natural vegetation (LPJmL).

**2. Is it possible to estimate uncertainty with this method? IPCC methods often have large uncertainty but does this method have less uncertainty because it is a Tier 2method. If it is not possible to estimate uncertainty could the authors speculate on the level of uncertainty in the predictions. Knowing something about uncertainty would be helpful in comparisons to the modeled results from other studies that are shown in the manuscript.**

Answer to 2.: The quantitative assessment of the uncertainty of our projections unfortunately exceeds the scope of this article and would likely require a study in itself. The model includes a high number of parameters, and for most of these the uncertainty distributions have not been quantified so far. Moreover, we think that beyond parameter uncertainty, the structural uncertainty from the model design is also very high.

To address the uncertainty within this manuscript, we will therefore discuss the uncertainty qualitatively and add a short statement here: Most of the uncertainty in our view is included in the management data itself, and especially in the residue and manure numbers, as they are only indirectly calculated from production, feed and area

data. The uncertainty of recycling shares adds on top of the uncertain total numbers of manure and residue biomass. As this is one of the first attempts to compile a comprehensive, global data set on these management data, it is complex to evaluate it against literature. Furthermore, it is still to be evaluated, if our data set covers most of the carbon inputs to the soil, as it not includes cover crops, green manure and weed biomass. Additionally, litter estimations are based on the set of global parameters, that contain large uncertainty and at the same time have a great impact on the overall size of SOC stocks. Both effects will be discussed in more detail.

**3. The authors state that a sensitivity analysis presented in Figure 4 shows that management impact is robust to the initialization of the soil organic carbon stocks at the beginning of the spin-up phase. But, the stocks and change in stocks almost halves the values if the initialization is done with natural vegetation. The initialization does make a difference, and needs further explanation.**

Answer to 3.: It is correct that the SOC stocks are highly dependent on the legacy of management. However, Figure 4 shows that the SOC gap - the difference between a baseline scenario and a counterfactual scenario with constant management - was not strongly dependent on the initialization. We therefore conclude that the initialization does not affect our central finding.

Still, in order to improve our estimates also for the absolute SOC stock, we now extended the length of the spin-up phase, starting in 1510 (default spin-up start for introducing land use in simulations with LPJmL, see e.g. Schaphoff et al. 2018a/b, von Bloh et al. 2018). We will analyze and discuss our results with respect to the initialization in more detail.

(References: von Bloh, W. et al. 2018: Implementing the nitrogen cycle into the dynamic global vegetation, hydrology, and crop growth model LPJmL (version 5.0). Geoscientific Model Development 11, 2789–2812. Schaphoff, S. et al. 2018a: LPJmL4 – a dynamic global vegetation model with managed land – Part 1: Model description.

Geoscientific Model Development 11, 1343–1375. Schaphoff, S. et al. 2018b: LPJmL4 – a dynamic global vegetation model with managed land – Part 2: Model evaluation. Geoscientific Model Development 11, 1377–1403.)

**4. Good to see that the authors have made a comparison to another approach to confirm the Tier 2 results. The Tier 1 method provided by the IPCC has been used for this purpose. In section 2.2, the authors present a method estimating stock change factors instead of soil organic carbon changes. But, the results in Table 4 for the stock change factors are not convincing that the methods are consistent, and the text seems unclear with discussion about larger differences with the IPCC 2019 values, which were updated by the authors and should be more accurate – I would think. Why not estimate the change in soil organic carbon for a direct comparison with the Tier 2 method instead of the stock change factors? Also, the placement of these results after the discussion seems odd, and conventionally would be presented in the results section before discussion.**

Answer to 4.: Thank you for the recommendation. We will place the comparison within the result section. For a Tier 1 approach the change in soil organic carbon must be calculated based on a reference stock. The default stocks given by the IPCC have a very low spatial resolution (42 coarse climate zones and soil type specific values), which lead to additional uncertainty when directly comparing changes in soil organic carbon. We will add more detail to the discussion of the strong deviation between the 2006 and 2019 default stock change factors.

**5. Figure 3 shows results from making certain practices constant from 1975 to 2010. The authors state around line 315 that the effect of no-till has been strong since 1990, but the effect seems minor and may not differ statistically from the histManagement with uncertainty. The conclusion about the importance of residue seems most important here.**

Answer to 5.: Due to the overall changes in the results, we will rewrite major part of

the result and discussion. We will take more thoroughly into account the effects of uncertainty.

**6. The authors suggest that there needs to be a circular flow with food supply chain back to soils. They assumed that none of the waste from supply chains are returned to soils (near line 300) but this seems incorrect. Municipal waste and materials are amended to soils in many regions of the world although maybe there are no data on these amendments? If this is the issue, the authors could mention that they are making a conservative assumption due to lack of data.**

Answer to 6.: Indeed, further soil inputs include the application of human excreta and sewage sludge, as well as the application of processing wastes, forestry residues or biochar. Unfortunately, data on the quantity of these inputs is very scarce and often does not exist with global coverage. In the draft of the International Nitrogen Assessment (will be provided confidentially to the reviewer), the available literature estimates for these flows (in regard to nitrogen) were reviewed. So far no reliable estimates exist, but the existing estimates indicate that these inputs must be by an order of magnitude lower than those of crop residues and manure.

**7. The authors evaluate the sensitivity of the Tier 2 model for tree litter with methods in Section 2.4.3. The Tier 2 model divides litter into metabolic and structural components, and the authors have averaged lignin to nitrogen across tree components as input to the model. But forest also include deadwood and should be separated from other forest litter to model decomposition. Did the authors add a deadwood pool? I question if this model is appropriate for forest if deadwood is not modeled separately.**

Answer to 7.: We follow this suggestion and will add more detail on the litterfall of the natural vegetation. Using additional information from LPJmL, we will split up the litterfall into a wood- and soft-tissue fraction and add different parameterizations for these. We will not be able to add a deadwood pool, since that would require additional

parameterization of turnover dynamics for this new pool. Deadwood pools are also not treated explicitly in many DGVMs but are considered part of litter pools, which distinguish woody from non-wood litter pools. The separation of litterfall into wood and soft tissue fluxes will thus add similar stock detail as in DGVMs.

**8. Recommend that the authors provide more explanation for Equation 9, which determines the residue amount of C, and is a key driver of the carbon change. Harvest index is the proportion of plant biomass that is harvested, but the authors are multiplying the harvested crop product by the harvest index. But the conventional approach is 'harvested crop production divided by the harvest index' to determine the total biomass and then subtract the harvested amount to estimate the residue. The authors are accounting for double harvesting and fallow in this calculation, which I agree is important, but some further explanation is needed about the calculation to understand how residue carbon is estimate from crop production, harvest index and area.**

Answet to 8.: Our harvest index for the yield (t/ha) is calculated based on a linear function with positive intercept (ha) and a slope dependent on the yield (t/ha). This allometric function accounts for the fact that higher-yielding crops often have a lower harvest index than low-yielding crops. In our revised manuscript, we will rewrite the equations to make the functional form more visible. Instead of $AGR_{i,t,cg} = CP_{i,t,cg} \cdot HI_{prod,cg} + CA_{i,t,cg} \cdot HI_{area}$ we now write $AGR_{i,t,cg} = CA_{i,t,cg} \cdot (Y_{i,t,cg} \cdot HI_{prod,cg} + HI_{area})$ We will also add more explanations to the text.

**9. For the Tier 1 method, IPCC divides the reference carbon stocks by climate and soil types. Did the authors also divide the grid cells by climate and soil because only climate is mentioned in the text? And, I found a diagram in Figure 5.1 in the IPCC report that divides low, medium and high input categories. Did the authors use this diagram to classify the input? It is not clear if the authors use the diagram or developed their own. If they developed their own, is it consistent with the IPCC factors?**

Answer to 9.: We do not calculate carbon stocks and stock changes based on the Tier 1 method. Instead we calculate stock change factors from our Tier 2 approach to compare only these factors to the Tier 1 factors as given within the guidelines. We will point this out more clearly in the method and discussion part. In addition, we note here: * To our knowledge, the IPCC method splits up soil types into mineral and organic. This analysis focuses exclusively on mineral soils, which will be pointed out more clearly throughout the paper. * To aggregate our Tier 2 results we assigned only one climate type for each grid cell (the most dominant one). * The Tier 2 method does not take into account different soil classes directly, but rather takes the sand fraction as a proxy for soil properties (like water holding capacities). * For the Tier 1 method no detailed analysis has been conducted. All factors shown in the manuscript are default factors without any spatial disaggregation of effects. We did not classify the inputs into Tier 1 categories (as there was no need to do so for Tier 2 and no data available to do so at the global scale).

**10. What is 'resp' is 'area reduction resp' on line 110? This sentence should be revised to improve readability. I also found other sentences that were difficult to read or missing words in some cases, but did not make a list during my review. Suggest a careful review before final publication.**

Answer to 10.: We will do so.

**11. I found the Tier 2 method in Chapter 5 of Volume 4 of the 2019 IPCC report, and would suggest that the authors cite this chapter rather than the entire 2019 IPCC report,which has 5 volumes. This would make it easier for others interested in the study to find the method in the IPCC report.**

Answer to 11.: We will change the citation from full report to the Chapter 5 where needed.

Please also note the supplement to this comment:

https://bg.copernicus.org/preprints/bg-2020-468/bg-2020-468-AC2-supplement.pdf

---

## Author Comment (AC3) · 8 Jun 2021

**Preface: This MS presents the findings of a spatially explicit implementation of a new IPCC Tier II approach to soil carbon stock changes. The authors use this model to calculate how much SOC has been lost due to agriculture but then go on to run the model annually for the period 1975 – 2010 to produce a dynamic picture of SOC recovery over this modern era of farming. The major takeaway message is that while agriculture has incurred a large SOC debt, recent agronomic improvements have led to 4 Pg C of SOC sequestration over this period of time. This detailed picture of SOC in croplands over the past several decades is of incredible importance to policy makers and as such I believe that paper can**

**be an important contribution to the literature; however, I do have several major
concerns that may or may not be addressable with revisions.**

Answer to preface: Dear Dr. Sanderman, thank you for the thorough and helpful review of our manuscript. While checking and revising our data processing in response to
your and the other reviewer's feedback, we discovered a bug in the soil model, leading
to an overestimation of the transfer of carbon from the active to the slow pool exclusively for cropland. Additionally, we found unreasonable high forage crop production
values (specifically for pumpkins used as fodder) in our input data, which are taken
from FAO statistics. This made the overall intensification trend in agriculture lead to
increasing carbon stocks in cropland. After correcting the bugs, this is no longer the
case. Whereas this implies major revisions of our discussion and interpretation of results, we argue that the essence of the paper remains unchanged, albeit modified.
We suggest that our key findings are a) we introduce a soil carbon model that can
account for changes in agricultural management and can be applied within integrated
assessment frameworks for the first time and b) we show that it is critical to account
for management dynamics in SOC assessments. We provide an assessment of how
results changed after correcting the bugs as a supplement to this author's comment.
Here, we respond point-by-point to the reviewers' comments. Where appropriate, this
will also address the implications of the bug fix. We look forward to your response.

**1. I had to read the methods section twice and spend an hour with Calvo Buendia et al., 2019 to fully understand what the authors have done. I'm still not 100%
confident that I fully understand the methodology. I suggest adding an illustrative example or two graphically demonstrating how the process works. Perhaps
starting with a simple case of one lu transition and then showing a more complex
case of multiple lu transitions within a pixel.**

Answer to 1.: We will add more detail and graphical representation for demonstrating
especially the land-transition accounting.

**2. I have not been convinced that this sort of "dynamic" implementation of a steady state modeling approach is appropriate. I understand that the method was developed by the IPCC as a way to add more nuance into the Tier I emission factor approach but I don't think the method was intended to be applied annually. Why not go all the way to Tier III approach using the process-based dynamics that are embedded in this simplified model? It appears you have all the data assembled to do this. My main concern with applying a steady-state model to annual changes is we know that the recent past trajectory of SOC (particularly in the slow and passive pools) will greatly influence the short-term model response to improved management – i.e. the model will take years to decades before SOC stocks start to rebuild if the trajectory was negative prior to the change – but this will be completely missed with the steady state application (stocks will start increasing immediately upon change).**

Answer to 2.: This has already been answered in the earlier reply ('Reply on RC1', Kristine Karstens, 26 Jan 2021).

**3. What time frame are the monthly climate data averaged over to get the rate modifiers for a steady state solution? Given the passive pool has an intrinsic decay rate equivalent to >100 year turnover time, it seems that you need to have a 100+ year average climate to come up with the proper rate modifiers.**

Answer to 3.: As we do not use a steady state model, we believe the comment might be based on a misunderstanding and no averaging of climate data is needed.

**4. Transfer between lu types is not clear. I do not understand how a "respective share of the SOC is reallocated." My concern is that the per area SOC stock for long-term cultivated land will be much different than the per area SOC stock for newly converted cropland, so I don't see how you can suddenly bin these separate areas into one model component. Perhaps my request for a visual guide will help me (and other readers)understand that I have misunderstood this part**

**of the methods.**

Answer to 4.: See above: We will add more detail and graphical representation here. This will illustrate the linearity of the problem; accounting separately for freshly converted cropland and old cropland is - in our equation system - essentially the same as taking the area-weighted mean.

**5. Why was 1901 chosen to spin up to steady state? We know that this was a time of rapid agricultural expansion in several major regions of the world and thus a time of rapid soil carbon loss.**

Answer to 5.: We extended the spin-up phase for a much longer period, starting in 1510 (the default spin-up start in LPJmL) to capture the land-use effects. Our initial choice of starting in 1901 was driven by the availability of climate data. To extend our spin-up into the past, we will recycle climate data from 1901 to 1930, as this is done in DGVM simulations as well (Schaphoff et al. 2018a/b, von Bloh et al. 2018). The management data will be held constant on the first known level.

(References: von Bloh, W. et al. 2018: Implementing the nitrogen cycle into the dynamic global vegetation, hydrology, and crop growth model LPJmL (version 5.0). Geoscientific Model Development 11, 2789–2812. Schaphoff, S. et al. 2018a: LPJmL4 – a dynamic global vegetation model with managed land – Part 1: Model description. Geoscientific Model Development 11, 1343–1375. Schaphoff, S. et al. 2018b: LPJmL4 – a dynamic global vegetation model with managed land – Part 2: Model evaluation. Geoscientific Model Development 11, 1377–1403.)

**6. LULUC data – are these data all provided as area within each grid cell (or percent of a grid)? I think so, but please indicate. A supplemental table with the cross-walk between the LUH2 and the 17 crop groups used in this study should be included.**

Answer to 6.: Yes, all area data are in hectares and meant to represent the area within

each grid cell. We will indicate that more clearly. Additionally, we will add a table to the supplemental material.

**7. Please include units for eq 9 – I don't follow the AGR calculation – it sounds like you are adding biomass and area together. Additionally, HI is usually defined as CP divided by total above ground biomass, so CP x HI is a meaningless number.**

Answer to 7.: Our harvest index for the yield (t/ha) is calculated based on a linear function with positive intercept (ha) and a slope dependent on the yield (t/ha). This allometric function accounts for the fact that higher-yielding crops often have a lower harvest index than low-yielding crops. In our revised manuscript, we will rewrite the equations to make the functional form more visible. Instead of $AGR_{i,t,cg} = CP_{i,t,cg} \cdot HI_{prod,cg} + CA_{i,t,cg} \cdot HI_{area}$ we now write $AGR_{i,t,cg} = CA_{i,t,cg} \cdot (Y_{i,t,cg} \cdot HI_{prod,cg} + HI_{area})$ We will also add more explanations to the text.

**8. Lack of validation. There appears to be no attempt to validate the model or the input data used to drive the model. In general, there is a lack of quantitative evaluation throughout. There are just two qualitative quality assessments – a table comparing calculated stock change factors to Tier I estimates and a discussion on how the map looks similar to other SOC maps.**

Answer to 8.: To address the raised issue, we will add to our analysis: * A grid level comparison of SOC stock results to SoilGrids 2.0 and LPJmL to improve spatial evaluation of our results. * More literature comparison of our derived agricultural land emissions as they are next to the SOC stocks the most central outcome of our analysis. * An extended qualitative discussion of uncertainty and where it might be the largest.

**9. The model itself was developed recently as part of the 2019 Refinements to IPCC Guidelines and those updated guidelines discuss how the model was calibrated to a set of long-term trial sites but do not report any model performance metrics. As pointed out by the authors some areas of central EU and the UK more**

than double SOC undercurrent agriculture than under native vegetation. **This is certainly indicative that there should be some checks against real data (see detailed elaboration on this point further down in this review). It could be argued that point-based validation for a model run at0.5 degree resolution is meaningless but it would be an interesting exercise to see how the model reproduces trends with known SOC histories.**

Answer to 9.: Our updated model does not show the superising behaviour of strongly increasing SOC stocks in UK and central EU anymore due to the bugfix within the model and the corrected fodder production data. We will add a more detailed spatial comparison to SoilGrids 2.0 (SOC stock estimates based on point data measurements) and other model based SOC estimates (LPJmL) to account for the missing spatial dimension of model output evaluation.

**10. Issues with residue C return. Given that the major takeaway from this paper is that the SOC is being sequestered due to improved yield leading to increased residue return, is there any empirical evidence that C inputs to the mineral soil have nearly doubled (Fig 3)? I think the method for calculating residue return to the soil is potentially flawed leading to this large apparent increase. The authors have assumed that both harvest index (HI) and root-to-shoot(RS) ratios have been constant through time. However, yield improvements over the last century, and in particular the last 50 years, are a result of improvements in genetics and nutrition. Breeding has resulted in the ability to plant most crops at much higher densities and selection towards more photosynthate being allocated to harvestable organs. Both of these improvements have altered HI and RS ratios. Additionally, there are strong interactions between N fertilization rate and root density. There is a huge literature on crop breeding that support the nonstationarity of these important parameters.**

Answer to 10.: This is an excellent point that we took up in the discussion. In particular, we would like to highlight the following: The IPCC methodology, which we use here,

tries to capture the effect of a shifting harvest index by making the harvest index a linear function of yield with a positive intercept. The parametrization of this dynamic harvest index was not possible for all crops due to a lack of literature estimates, but the most important crops like cereals or soybeans are captured.

**11. Units – Gt and Mt are not SI units, please use Pg and Tg**

Answer to 11.: We will change all units to SI and add non-SI units in brackets were needed to bridge to IAM modelling community.

**12. L13 (and elsewhere) – "we also find that SOC is very sensitive..." – this is in reference to an unvalidated model result. I'd suggest rewording these sorts of phrases to, "Our model results suggest that SOC is very sensitive...**

Answer to 12.: We will do a careful revision of naming and framing of the evaluation related statements.

**13. L279 – "we provide the first world map" – no, you did not. All of the global maps that have been developed using a statistical environmental-covariate modeling approach(i.e. soilgrids and similar) implicitly include all historic land management.**

Answer to 13.: This is correct. We will rewrite the paragraph.

**14. It is great that all the data are provided but I found the Karstens 2020a repository to be confusing. Can you have a description for each file in the repository? The naming convention is not clear. I did not want to download 9 Gb of data to figure it out.**

Answer to 14.: We will follow this suggestion and will add more detail to the README file including a description for each file.

**15. Fig 1 – perhaps it is just the spatial scale of these small maps (and I haven't downloaded the results to explore in more detail) but it looks like there is zero**

**intact forest in the Congo Basin and very little intact forest in the Amazon. Also, I would have liked to see a map showing the trend in SOC spatially – how are the 4 Pg C that has been sequestered been spread across the globe? Is it all in Central EU and UK?**

Answer to 15.: Fig 1.(a) is showing cropland SOC for every grid cell that contains cropland, without giving an information on the extent of cropland. We will mask out cells with very low cropland area shares, as they might give to the impression of greater cropland extent within large parts of intact forested area. We will also provide a total SOC map in the appendix.

**16. Fig 2 – this is a really interesting way of summarizing the model results**

Answer to 16.: Thank you!

**17. Fig 4 – the finding presented here is very counterintuitive to me. Why is the SOC debt halved when the model is initialized with natural vegetation? Shouldn't the 1975 SOC debt be much greater if the 1901 starting point was natural vegetation instead of actual land use? Perhaps I am just misunderstanding this sensitivity analysis.**

Answer to 17.: The initialization analysis was meant to help the reader understand the potential maximal error of underestimating on-going emissions in croplands, that where converted to croplands before 1901. We have now extended the spin-up phase from 1510 to now. We will move the initialization analysis to the appendix and improve the caption of the figure to make its interpretation more clear.

**18. Discussion section – in general, there is very little discussion of how these results fit into the large literature on SOC. There are many places were a reference or two would great increase the credibility of the statements that are being made.**

Answer to 18.: We will include more connections to existing literature here. Section 4.2

– I think this section should come right before the conclusions especially as you refer to analysis that is only presented for the first time in section 4.3 Section 4.3 will be moved to the end of the result as evaulation results subsection.

**19. L358-360 – the finding that northern temperate zones (particularly in EU and UK) now have SOC levels up to twice that of native state yet tropical soils have lost 40-70% of their SOC is problematic and, as the authors point out in relation to the EU example,likely points to issues with getting C input to soil correct. The EU has the perfect data set to test this model finding – the EU JRC LUCAS survey was a balanced sampling design between forested and agricultural land uses. In the tropics, it has been fairly well documented that already infertile tropical soils do not lose nearly as much SOC as their fertile temperate zone soil counterparts. While there are issues and large scale generalizations in the IPCC Tier I default factors, they do represent the consensus literature on the topic. The updated meta-analysis between the 2006 and 2019 IPCCC guidelines when this emission factor for the tropics changed dramatically (see Table 4in this MS) points to this new knowledge.**

Answer to 19.: After our bug fix, SOC stocks for the EU no longer gain SOC compared to natural vegetation. In general, all carbon stocks are much smaller and show much higher losses compared to the Tier 1 approach for both temperate and tropical soils. This may be indicative of gaps in the accounting of carbon inputs to the soil and will be discussed in more detail within the discussion. The comparison to point data is however challenging, also as point measurements do not well capture the landscape average and will likely show a very high variance. On the one hand the LUCAS database is given as soil carbon density and would need consistent bulk density data to be comparable to our results. A full comparison to point measurements may also exceed the scope of this first article. We hope that our additional comparison to SoilGrids 2.0 might help fill the evaluation gap here as SoilGrids 2.0 is based on point measurements.

**20. L396 – how is this validation? It is just a comparison.**

Answer to 20.: The naming and framing of the evaluation of our results will be improved.

**21. L400-406 – there is a large literature that can be drawn upon to support some of the claims made in this section.**

Answer to 21.: We will add more detail and references here.

**22. Section 4.4 – I do not think this is a valid comparison because SoilGrids explicitly tried to capture high carbon density soils well while your model explicitly excludes organic soils. I suggest applying an agriculture mask to all of these data sets and then redo the analysis. Additionally, ISRIC released an update to SoilGrids >6 months ago that focuses primarily on mineral soil carbon stocks. This update is probably a better comparison.**

Answer to 22.: We included the recommended SoilGrids 2.0 data for a more spatially detailed evaluation of our results especially for cropland soils.

**23. L425-428 – this seems out of place.**

Answer to 23.: We will include a discussion of the sensitivity in our extended discussion on evaluation and uncertainty of our results.

**24. L453 – comparison to 4p1000 is not really fair because your model is really just the business-as-usual scenario with SOC gains simply because yields are improving glob-ally. 4p1000 is about intentional management shifts to increase SOC.**

Answer to 24.: We agree with the reviewer, that the comparison is a bit misleading. We still think that it is fruitful to compare the observed rates with ambitious targets and will reframe the discussion on this point.

Please also note the supplement to this comment:
https://bg.copernicus.org/preprints/bg-2020-468/bg-2020-468-AC3-supplement.pdf

**Supplement:**

**Update after major correction to *Management induced changes of soil organic carbon on global croplands**

Kristine Karstens

June 8, 2021

In the following we will provide an update on some of our main results in comparison to the version of our initial draft to help the reader understand the implication of our changes. The following major corrections and improvements have been made, that let to substantial changes (next to smaller improvements):

- We discovered a bug in the very core of the soil model, leading to an overestimation of the transfer of carbon from active to the slow pool *just* for cropland. In regions with high carbon inputs (e.g. UK or Central EU) this let to unreasonable high SOC stocks.
- Additionally, we found unreasonable high forage crop production values for pumpkins used as fodder in our input data, we used from FAO. This led to an additional decrease of residue inputs to the soil for several countries (including, Australia, Belgium and Germany). The values were excluded and replaced by zeros.
- We moreover improved the spin-up of our model by accounting for land-use change since 1510.

**Summary of main changes**

The SOC debt is not decreasing anymore, but still continuously increases. Unreasonable high numbers for SOC stocks and stock change factors compared to natural vegetation stocks have vanished. The impact of management is lower but still considerably. Stock change factors for cropland SOC are now globally lower than default factor from the IPCC for all four climate zones.

In the following, we provide the comparison of updated (first) and original (second) figures for three of our main results:

- SOC distribution and depletion (Fig. 1 in the manuscript)
- Agricultural management effects on SOC debt (Fig. 3 in the manuscript)
- Modeled management effect in comparison with default IPCC Tier 1 factors (Tab. 4 in the manuscript)

**SOC distribution and depletion**

[Figure]

Figure 1: Updated figure! (a): Distribution of total global SOC stocks for the first 30 cm on cropland: Carbon stocks are large in high yielding areas. (b)+(c): Absolute (b) and relative (c) SOC stocks changes compared to a potential natural state identify different hot spots of SOC dynamics. Whereas absolute losses $\Delta SOC$ are often highest in temperate dry regions, relative losses $F^{\mathrm{SCF}}$ are often larger in tropical moist areas. (d): SOC debt is the difference between SOC under historic land use and potential natural vegetation. Within the period 1975 – 2010 the SOC debt is continuing to increase.

[Figure]

Figure 2: Original figure of the manuscript

With the figures above we provide a world map of SOC stocks for the first 30 cm on croplands considering historic management data at the global scale for the year 2010. Values ranging between well over $100 \text{tha}^{-1}$ in northern temperate croplands to less than $5 \text{tha}^{-1}$ for arid and semiarid croplands. The correction of to high SOC stock values is visible for all four figure parts, as it corrects the unreasonable high values in UK and Central Europe (see old figure).

**Agricultural management effects on SOC debt**

[Figure]

Figure 3: Updated figure! (a) Global $\Delta SOC$ in GtC for different management scenarios: The stylized scenarios deviate from historic agricultural management by holding effects of carbon inflows from residues or manure constant, or neglecting adoption of no-tillage practices. ConstManagement combines all three modifications. Note that $\Delta SOC$ is defined as the difference of SOC under land-use compared to a natural vegetation state. Figure (b) shows the carbon inflows from crop residue and manure.

[Figure]

Figure 4: Original figure of the manuscript

Most notably difference is the trend of the historical reference line as well as the split between stylized constManagement and the baseline histManagement. Before we calculated a split of 8GtC, that now decreased to around 2GtC, which is still a third of the SOC loss of the period from $1975 - 2010$.

**Modeled management effect in comparison with default IPCC Tier 1 factors**

Stock change factors $F^{mathrmSCF}$ in comparison to IPCC Tier 1 default factors: Our updated results are much smaller and still considerably lower compared to the default values of the IPCC for all four climate zones.

|   | Source | Input | Year | tropical moist | tropical dry | temperate dry | temperate moist |
|---|--------|-------|------|----------------|--------------|---------------|-----------------|
| 1 | IPCC2006 | low | invariant | 0.44 | 0.55–0.61 | 0.74 | 0.66 |
| 2 | IPCC2006 | medium | invariant | 0.48 | 0.58–0.64 | 0.80 | 0.69 |
| 3 | IPCC2006 | high | invariant | 0.53 | 0.60–0.67 | 0.83 | 0.77 |
| 4 | IPCC2019 | low | invariant | 0.76 | 0.87 | 0.70–0.71 | 0.66–0.67 |
| 5 | IPCC2019 | medium | invariant | 0.83 | 0.92 | 0.76–0.77 | 0.69–0.70 |
| 6 | IPCC2019 | high | invariant | 0.92 | 0.96 | 0.79–0.80 | 0.77–0.78 |
| 7 | This Study | hist | 1990 | 0.49 | 0.54 | 0.67 | 0.63 |
| 8 | This Study | hist | 2010 | 0.51 | 0.56 | 0.66 | 0.63 |

Table 1: Updated table.

|   | Source | Input | Year | tropical moist | tropical dry | temperate dry | temperate moist |
|---|--------|-------|------|----------------|--------------|---------------|-----------------|
| 1 | IPCC2006 | low | invariant | 0.44 | 0.55–0.61 | 0.74 | 0.66 |
| 2 | IPCC2006 | medium | invariant | 0.48 | 0.58–0.64 | 0.80 | 0.69 |
| 3 | IPCC2006 | high | invariant | 0.53 | 0.60–0.67 | 0.83 | 0.77 |
| 4 | IPCC2019 | low | invariant | 0.76 | 0.87 | 0.70–0.71 | 0.66–0.67 |
| 5 | IPCC2019 | medium | invariant | 0.83 | 0.92 | 0.76–0.77 | 0.69–0.70 |
| 6 | IPCC2019 | high | invariant | 0.92 | 0.96 | 0.79–0.80 | 0.77–0.78 |
| 7 | This Study | hist | 1990 | 0.57 | 0.61 | 0.78 | 0.76 |
| 8 | This Study | hist | 2010 | 0.64 | 0.68 | 0.83 | 0.83 |

Table 2: Original table of the manuscript.

---

## Author Response (AR1)

**Point by point replies**

**Point by point replies - Jonathan Sanderman**

**Preface: This MS presents the findings of a spatially explicit implementation of a new IPCC Tier II approach to soil carbon stock changes. The authors use this model to calculate how much SOC has been lost due to agriculture but then go on to run the model annually for the period 1975 – 2010 to produce a dynamic picture of SOC recovery over this modern era of farming. The major takeaway message is that while agriculture has incurred a large SOC debt, recent agronomic improvements have led to 4 Pg C of SOC sequestration over this period of time. This detailed picture of SOC in croplands over the past several decades is of incredible importance to policy makers and as such I believe that paper can be an important contribution to the literature; however, I do have several major concerns that may or may not be addressable with revisions.**

Answer to preface:

Dear Dr. Sanderman, thank you for the thorough and helpful review of our manuscript. While checking and revising our data processing in response to your and the other reviewer's feedback, we discovered a bug in the soil model, leading to an overestimation of the C transfer to the soil exclusively for cropland. Additionally, we found unreasonable high forage crop production values in our input data, which are taken from FAO statistics. This made the overall intensification trend in agriculture lead to increasing carbon stocks in cropland. After correcting the bugs, this is no longer the case. Following upon your and your co-reviewers comments we improved additionally the initialization of our SOC estimates by extending the spin-up to a much longer period (1510) and refined the natural litter representation substantially.

Whereas this implies major revisions of our results, discussion and interpretation, we argue that the essence of the paper remains intact, albeit modified. We suggest that our key findings are a) we introduce a soil carbon model that can account for changes in agricultural management and can be applied within integrated assessment frameworks for the first time and b) we show that it is critical to account for management dynamics in SOC assessments. We provide an assessment of how results changed after correcting the bugs as a supplement to our last author's comment.

Here, we respond point-by-point to the reviewers' comments. Please note, that due to the amount of changes introduced within this update, we will refer to the section within the new manuscript rather than include the entire changed paragraph within this replies. We look forward to your response.

**1. I had to read the methods section twice and spend an hour with Calvo Buendia et al., 2019 to fully understand what the authors have done. I'm still not 100% confident that I fully understand the methodology. I suggest adding an illustrative example or two graphically demonstrating how the process works. Perhaps starting with a simple case of one lu transition and then showing a more complex case of multiple lu transitions within a pixel.**

Answer to 1.:

We added a graphical representation (see Fig. 1) for demonstrating the land-transition accounting to the method part (see Sect. 2.1.2 on "SOC transfer between land-use types").

**2. I have not been convinced that this sort of "dynamic" implementation of a state modeling approach is appropriate. I understand that the method was developed by the IPCC as a way to add more nuance into the Tier I emission factor approach but I don't think the method was intended to be applied annually. Why not go all the way to Tier III approach using the**

process-based dynamics that are embedded in this simplified model? It appears you have all the data assembled to do this. My main concern with applying a steady-state model to annual changes is we know that the recent past trajectory of SOC (particularly in the slow and passive pools) will greatly influence the short-term model response to improved management – i.e. the model will take years to decades before SOC stocks start to rebuild if the trajectory was negative prior to the change – but this will be completely missed with the steady state application (stocks will start increasing immediately upon change).

Answer to 2.: We would like to clarify that the Tier 2 modeling approach within the refinement of the IPCC Guidelines of 2019 is a first order kinetic approach. It was just named by the IPCC authors as "steady-state method", since it is based on steady-state calculation as an intermediate step. Yet, it does not assume SOC stocks in immediate steady state, and does account - depending on the pool type - for longer than annual transition phase to new steady states. We share your confusion about the naming within the guidelines, and changed the naming in our analysis from "steady-state method" to "Tier 2 modeling approach".

We also highlight in the introduction and conclusions that our Tier-II approach with its reduced complexity has advantages to a more complex process-based Tier-III approach when integrating it into computational intensive Integrated Assessment Models.

**3. What time frame are the monthly climate data averaged over to get the rate modifiers for a steady state solution? Given the passive pool has an intrinsic decay rate equivalent to >100 year turnover time, it seems that you need to have a 100+ year average climate to come up with the proper rate modifiers.**

Answer to 3.: As we do not use a steady state model, we believe the comment might be based on a misunderstanding and no averaging of climate data is needed.

**4. Transfer between lu types is not clear. I do not understand how a "respective share of the SOC is reallocated." My concern is that the per area SOC stock for long-term cultivated land will be much different than the per area SOC stock for newly converted cropland, so I don't see how you can suddenly bin these separate areas into one model component. Perhaps my request for a visual guide will help me (and other readers)understand that I have misunderstood this part of the methods.**

Answer to 4.: See answer to 1. The added scheme of land-use transition representation (see Fig.1) illustrates the linearity of the problem; accounting separately for newly converted cropland and existing cropland is - in our equation system - mathematically the same as taking the area-weighted mean.

**5. Why was 1901 chosen to spin up to steady state? We know that this was a time of rapid agricultural expansion in several major regions of the world and thus a time of rapid soil carbon loss.**

Answer to 5.: We extended the spin-up phase for a much longer period, starting in 1510 (the default spin-up start in LPJmL) to capture the land-use effects. Our initial choice of starting in 1901 was driven by the availability of climate data. To extend our spin-up into the past, we will recycle climate data from 1901 to 1930, as this is done in DGVM simulations as well (Schaphoff et al. 2018a/b, von Bloh et al. 2018). We rewrote the Sect. 2.1.4 on "Initialization of SOC pools" to explain this in more detail.

(References: von Bloh, W. et al. 2018: Implementing the nitrogen cycle into the dynamic global vegetation, hydrology, and crop growth model LPJmL (version 5.0). Geoscientific Model Development 11, 2789–2812. Schaphoff, S. et al. 2018a: LPJmL4 – a dynamic global vegetation model with managed land – Part 1: Model description. Geoscientific Model Development 11, 1343–1375. Schaphoff, S. et al. 2018b: LPJmL4 – a dynamic global vegetation model with managed land – Part 2: Model evaluation. Geoscientific Model Development 11, 1377–1403.)

**6. LULUC data – are these data all provided as area within each grid cell (or percent of a grid)? I think so, but please indicate. A supplemental table with the cross-walk between the LUH2 and the 17 crop groups used in this study should be included.**

Answer to 6.: Yes, all area data are in million hectares and meant to represent the area within each grid cell. We indicated that more clearly in Sect. 2.3.1 by adding "(given as total land area in million ha)" to our explanation on "physical area" (LN 200). Additionally, we added a mapping table to the data archive at zenodo translating LUH2 croptypes and our crop types into FAO crop categories (see Table "FAO2LUH2MAG_croptypes.csv").

**7. Please include units for eq 9 – I don't follow the AGR calculation – it sounds like you are adding biomass and area together. Additionally, HI is usually defined as CP divided by total above ground biomass, so CP x HI is a meaningless number.**

Crop residue production (ton dry matter residues) are calculated based on a linear function with positive intercept and a yield-dependent slope. This allometric function accounts for the fact that higher-yielding crops often have a lower harvest index than low-yielding crops. In our revised manuscript, we rewrote the Eq. 9 to make the functional form more visible to

$$
\begin{aligned}
AGR_{i,t,cg} &= CA_{i,t,cg} \cdot \left( Y_{i,t,cg} \cdot r_{cg}^{\mathrm{ag,prod}} + MCF_{i,t} \cdot r_{cg}^{\mathrm{ag,area}} \right) \qquad \text{and} \\
BGR_{i,t,cg} &= \left( CA_{i,t,cg} \cdot Y_{i,t,cg} + AGR_{i,t,cg} \right) \cdot r_{cg}^{\mathrm{bg}} \quad .
\end{aligned}
\tag{1}
$$

Where one can also simplify $CA \cdot Y$ to a production value, as we did in the last version of the manuscript; keeping area and yield explicit in the new version however makes the dynamic harvest index more explicit.

We also added more detail to Sect. 2.3.2 on "Crop and crop residues production" and units for the three crop-group $cg$ specific ratios: * above-ground residues to harvested biomass $r_{cg}^{\mathrm{ag,prod}}$ in $(tDM\,ha^{-1})(tDM\,ha^{-1})^{-1}$ * above-ground residues to harvested area $r_{cg}^{\mathrm{ag,area}}$ in $tDM\,ha^{-1}$ * below-ground residues to above-ground biomass $r_{cg}^{\mathrm{bg}}$ in $tDM\,tDM^{-1}$.

**7. Lack of validation. There appears to be no attempt to validate the model or the input data used to drive the model. In general, there is a lack of quantitative evaluation throughout. There are just two qualitative quality assessments – a table comparing calculated stock change factors to Tier I estimates and a discussion on how the map looks similar to other SOC maps.**

Answer to 7.:

To address the raised issue, we added to our analysis a whole section on "Model evaluation" (see Sect. 3.4) featuring (next to the assessments done within in the first draft): * a grid level comparison of SOC stock results to SoilGrids 2.0 and LPJmL4 to improve spatial evaluation of our results, * a comparison to uncertainty estimates of SoilGrids 2.0, * a comparison to point-based SOC data.

**8. The model itself was developed recently as part of the 2019 Refinements to IPCC Guidelines and those updated guidelines discuss how the model was calibrated to a set of long-term trial sites but do not report any model performance metrics. As pointed out by the authors some areas of central EU and the UK more than double SOC undercurrent agriculture than under native vegetation. This is certainly indicative that there should be some checks against real data (see detailed elaboration on this point further down in this review). It could be argued that point-based validation for a model run at0.5 degree resolution is meaningless but it would be an interesting exercise to see how the model reproduces trends with known SOC histories.**

Answer to 8.: Our updated model does not show the surprising behavior of strongly increasing SOC stocks in UK and central EU anymore due to the correction within the model and the corrected fodder production data. Additionally, we added point-based evaluation of the results (see answer to 7).

**9. Issues with residue C return. Given that the major takeaway from this paper is that the SOC is being sequestered due to improved yield leading to increased residue return, is there any empirical evidence that C inputs to the mineral soil have nearly doubled (Fig 3)? I think the method for calculating residue return to the soil is potentially flawed leading to this large apparent increase. The authors have assumed that both harvest index (HI) and root-to-shoot(RS) ratios have been constant through time. However, yield improvements over the**

last century, and in particular the last 50 years, are a result of improvements in genetics and nutrition. Breeding has resulted in the ability to plant most crops at much higher densities and selection towards more photosynthate being allocated to harvestable organs. Both of these improvements have altered HI and RS ratios. Additionally, there are strong interactions between N fertilization rate and root density. There is a huge literature on crop breeding that support the nonstationarity of these important parameters.

Answer to 9.: We took up this point in the discussion. In particular, we would like to highlight the following: The IPCC methodology, which we use here, tries to capture the effect of a shifting harvest index by making the harvest index a linear function of yield with a positive intercept (see reply to point 7). The parametrization of this dynamic harvest index was not possible for all crops due to a lack of literature estimates, but the most important crops like cereals or soybeans are captured. We rewrote Sect. 2.3.2 "Crop and crop residues production" to be more clear on this aspect and reformulated Eq. 9. See also answer to 7.

**10. Units – Gt and Mt are not SI units, please use Pg and Tg**

Answer to 10.: To be consistent within the paper, we sticked to tons (t) and hectare (ha) as our two base units as agricultural production is often measured in t per ha, land-use areas in ha and SOC estimates on global scale (e.g. SoilGrids 2.0) sometimes in t per ha as well.

**11. L13 (and elsewhere) – "we also find that SOC is very sensitive. . . " – this is in reference to an unvalidated model result. I'd suggest rewording these sorts of phrases to, "Our model results suggest that SOC is very sensitive. . .**

Answer to 11.: We carefully revised our manuscript on naming and framing of the evaluation related statements.

**12. L279 – "we provide the first world map" – no, you did not. All of the global maps that have been developed using a statistical environmental-covariate modeling approach(i.e. soilgrids and similar) implicitly include all historic land management.**

Answer to 12.: This is correct. We will rewrite the paragraph.

**13. It is great that all the data are provided but I found the Karstens 2020a repository to be confusing. Can you have a description for each file in the repository? The naming convention is not clear. I did not want to download 9 Gb of data to figure it out.**

Answer to 13.: We added more detail to the README file including a description for each file. We also changed the output format to the more commonly used NetCDF format.

**14. Fig 1 – perhaps it is just the spatial scale of these small maps (and I haven't downloaded the results to explore in more detail) but it looks like there is zero intact forest in the Congo Basin and very little intact forest in the Amazon. Also, I would have likedto see a map showing the trend in SOC spatially – how are the 4 Pg C that has been sequestered been spread across the globe? Is it all in Central EU and UK?**

Answer to 14.: Fig. 2 (a) showed cropland SOC for every grid cell that contains cropland, without giving an information on the extent of cropland. We will mask out cells with very low cropland area (less than 1000 ha on a 0.5 degree grid), as they might give to the impression of greater cropland extent within large parts of intact forested area. We now also provide a total SOC debt and SOC debt trend map (see Fig. 3).

**15. Fig 2 – this is a really interesting way of summarizing the model results**

Answer to 15.: Thank you!

**16. Fig 4 – the finding presented here is very counterintuitive to me. Why is the SOC debt halved when the model is initialized with natural vegetation? Shouldn't the 1975 SOC debt be much greater if the 1901 starting point was natural vegetation instead of actual land use? Perhaps I am just misunderstanding this sensitivity analysis.**

Answer to 16.: The initialization analysis was meant to help the reader understand the potential maximal error of underestimating on-going emissions in cropland, that where converted to cropland before 1901. We have now extended the spin-up phase from 1510 to now. We therefore omitted the initialization analysis in the revised manuscript.

**17. Discussion section – in general, there is very little discussion of how these results fit into the large literature on SOC. There are many places were a reference or two would great increase the credibility of the statements that are being made.**

Answer to 17.: We rewrote the whole discussion section and included more connections to existing literature here.

**18. L358-360 – the finding that northern temperate zones (particularly in EU and UK) now have SOC levels up to twice that of native state yet tropical soils have lost 40-70% of their SOC is problematic and, as the authors point out in relation to the EU example,likely points to issues with getting C input to soil correct. The EU has the perfect data set to test this model finding – the EU JRC LUCAS survey was a balanced sampling design between forested and agricultural land uses. In the tropics, it has been fairly well documented that already infertile tropical soils do not lose nearly as much SOC as their fertile temperate zone soil counterparts. While there are issues and large scale generalizations in the IPCC Tier I default factors, they do represent the consensus literature on the topic. The updated meta-analysis between the 2006 and 2019 IPCCC guidelines when this emission factor for the tropics changed dramatically (see Table 4in this MS) points to this new knowledge.}**

Answer to 18.: After our bugfix, SOC stocks for the EU no longer gain SOC compared to natural vegetation. In general, all carbon stocks are much smaller and show much higher losses compared to the Tier 1 approach for both temperate and tropical soils. This may be indicative of gaps in the accounting of carbon inputs to the soil and will be discussed in more detail within the discussion. The comparison to point data is however challenging, also as point measurements do not well capture the landscape average and will likely show a very high variance. On the one hand the LUCAS database is given as soil carbon density and would need consistent bulk density data to be comparable to our results. However, we included point measurement comparison using the data provided in by Sanderman et al., 2017 for their model evaluation. Additionally, we hope that our additional comparison to SoilGrids 2.0 might help fill the evaluation gap here as SoilGrids 2.0 is based on point measurements.

**19. L396 – how is this validation? It is just a comparison.**

Answer to 19.: The naming and framing of the evaluation of our results was improved. We included a whole result section on "Model evaluation" (see Sect. 3.4).

**20. L400-406 – there is a large literature that can be drawn upon to support some of the claims made in this section.**

Answer to 20.: We rewrote the paragraph on the discussion on the stock change factors compared to the IPCC, as also some of the results have changed.

It now reads:

LN 481-495: "In comparison with default stock change factors of the IPCC guidelines, our model estimates a stronger decline of SOC stocks (Table 3) for almost all regions. Tropical soils might suffer from low C input rates due to large yield gaps (Global Yield Gap and Water Productivity Atlas. Available URL: www.yieldgap.org (accessed on: 03/01/2022)) and high shares of residue removal and burning in lower-income countries (Smil, 1999a; Williams et al., 1997; Jain et al., 2014). Yet, even when comparing our etimates to the low-input stock change factors of the IPCC, our SOC loss is roughly twice as large as the revised 2019 IPCC default values, while it shows very good agreement with the older default values from IPCC (2006). Don et al. (2011) estimated SOC losses for tropical soils of around 25% on average corresponding to a stock change factor of 0.75, but also reported a wide range of measured SOC changes from -80% to +58%. Fujisaki et al. (2015) however found much lower loss rates of around 9%, attributing the difference to the different time period length since the conversion to cropland. As our results do not specifically account for cropland age

and most of the cropland is older than 20 years (as assumed for the default IPCC Tier 1 stock change factors) our stock change factors have to be lower by definition following the steady-state assumption that cropland will continue to approach a new equilibrium. For the same reason, our estimates for temperate regions might be lower than both IPCC (2006) and IPCC (2019) default values. With the production-increasing impact of irrigation and fertilization on carbon-poor dryland soils, SOC under cropland can also be higher than under PNV with stock change factors above 1 (see Fig. 2(d)), but these areas are much smaller than where the stock change factors are well below unity."

**21. Section 4.4 – I do not think this is a valid comparison because SoilGrids explicitly tried to capture high carbon density soils well while your model explicitly excludes organic soils. I suggest applying an agriculture mask to all of these data sets and then redo the analysis. Additionally, ISRIC released an update to SoilGrids >6 months ago that focuses primarily on mineral soil carbon stocks. This update is probably a better comparison.**

Answer to 21.: We included the recommended SoilGrids 2.0 data for a more spatially detailed evaluation of our results especially for cropland soils. We aso applied a cropland mask to most f our results inclusing only grid cells with more than 1000 ha of cropland.

**22. L425-428 – this seems out of place.**

Answer to 22.: We removed the sensitivity analysis on plant parameterization, because we greatly improved litter parameterization of the natural vegetation.

**23. L453 – comparison to 4p1000 is not really fair because your model is really just the business-as-usual scenario with SOC gains simply because yields are improving glob-ally. 4p1000 is about intentional management shifts to increase SOC.**

Answer to 23.: We agree with the reviewer that the comparison is a bit misleading. We still think that it is fruitful to compare the observed rates with ambitious targets on the global scale and reframed it to: LN 514-517:"Annual C loss rates of 0.2 per 1000 C still have the opposite trend as the promoted 4 per 1000 C sequestration rate target (Minasny et al., 2017). Dedicated efforts to increase cropland SOC are thus necessary, as management improvements at historical rates are not enough to counteract ongoing SOC degradation on cropland."

**Point by point replies - second reviewer**

**0. The authors have conducted a study evaluating the influence of management on soil organic carbon in global croplands. This is an important topic for consideration of greenhouse gas mitigation with natural solutions for climate change policy and pro-grams such as the 4 per mille initiative. As the authors mention, there are few studies that have evaluated cropland management effects on soil organic carbon, and possibly none that have addressed the influence at a global scale. The result that increased residue return to soils is the leading driver of carbon changes over the past few decades in croplands is an important finding. As the authors note, the 4 Gt C increase in carbon is less than the goals of the 4 per mille initiative, which some have argued is not realistic. I have a few concerns about the study after review of the IPCC documentation on the method that the authors selected for this analysis. I would suggest that the authors make revisions before the manuscript is accepted.**

Answer to 0.: Dear reviewer, thank you for the thorough and helpful review of our manuscript. While checking and revising our data processing in response to your and the other reviewer's feedback, we discovered a bug in the soil model, leading to an overestimation of the C transfer to the soil exclusively for cropland. Additionally, we found unreasonable high forage crop production values in our input data, which are taken from FAO statistics. This made the overall intensification trend in agriculture lead to increasing carbon stocks in cropland. After correcting the bugs, this is no longer the case. Following upon your and your co-reviewers comments we improved additionally the initialization of our SOC estimates by extending the spin-up to a much longer period (1510) and refined the natural litter representation substantially.

Whereas this implies major revisions of our results, discussion and interpretation, we argue that the essence of the paper remains intact, albeit modified. We suggest that our key findings are a) we introduce a soil carbon model that can account for changes in agricultural management and can be applied within integrated assessment frameworks for the first time and b) we show that it is critical to account for management dynamics in SOC assessments. We provide an assessment of how results changed after correcting the bugs as a supplement to our last author's comment.

Here, we respond point-by-point to the reviewers' comments. Please note, that due to the amount of changes introduced within this update, we will refer to the section within the new manuscript rather than include the entire changed paragraph within this replies. We look forward to your response.

**1. The Tier 2 method is in a croplands chapter of the IPCC report. The documentation in the report states that the model would need to be parameterized for other land uses.Did the authors parameterize the model for other land uses that would be considered natural vegetation? If not, the estimation of soil organic carbon for natural vegetation may not be valid. The authors seem to suggest that this is a possibility in Section 4.4 when stating the soil organic carbon and debt from land use change have to be interpreted with caution. If the model has not been parameterized for natural vegetation, I would suggest that the authors focus on cropland model results, and remove the carbon debt results. The results for the cropland alone are important, and deserve publication even if the natural vegetation estimates are not valid with this model.**

Answer to 1.: This is an important point. No we did not reparameterized the model for other land-use types. While our analysis focuses on cropland, estimating the natural soil carbon stocks is necessary to account for the C entering the cropland budget via land conversion. To improve and evaluate the parametrization of natural soil carbon, we will include the following model updates: * We improved the litterfall parameterization in natural vegetation, accounting for plant-functional type and plant-organ specific parameterization of nitrogen and lignin contents. * We compared our results on soil stock under natural vegetation with the results of a model parametrized for natural vegetation (LPJmL4). * We compared our results for natural vegetation with a dataset of point measurements

**2. Is it possible to estimate uncertainty with this method? IPCC methods often have large uncertainty but does this method have less uncertainty because it is a Tier 2method. If it is not possible to estimate uncertainty could the authors speculate on the level of uncertainty in the predictions. Knowing something about uncertainty would be helpful in comparisons to the modeled results from other studies that are shown in the manuscript.**

Answer to 2.: The quantitative assessment of the uncertainty of our projections unfortunately exceeds the scope of this article and would likely require a study in itself. The model includes a high number of parameters, and for most of these the uncertainty distributions have not been quantified so far. Moreover, we think that beyond parameter uncertainty, the structural uncertainty from the model design is also very high.

A good way to account for uncertainty, also accounting for structural uncertainty, is the comparison of different modeling approaches. Our manuscript now includes our own model, a comparison to the IPCC stock change factors, a comparison of natural vegetation estimates to the DGVM LPJmL, a comparison to a machine-learning approach based on observation data (SoilGrids 2.0), and to point measurement data. The high ranges that emerge from this comparison reveal that it is likely too early to meaningfully quantify the total uncertainty connected to global SOC estimates.

We moreover discuss sources of uncertainty within the discussion section e.g. * LN 406-410: "It is important to evaluate the validity of our results as modeling management effects on SOC dynamics at the global scale comes with large uncertainties. The model includes a large number of parameters, and for most of these the uncertainty distributions have not been quantified so far. Moreover, we think that beyond parameter uncertainty, the structural uncertainty from the model design is high. The management data itself is prone to uncertainties as well, as most of it is only indirectly calculated from reported data." * LN 423-427: "For most of the parameters used in our management estimates no uncertainty estimates exist. This is why, in our view, most of the uncertainty with respect to the impacts of SOC management is included in the management

data itself, and especially in the residue and manure production and application numbers, as these are only indirectly derived from crop and livestock production, feed and area data (FAOSTAT, 2016; Weindl et al., 2017). The uncertainty of recycling shares adds on top of the uncertain total numbers of manure and residue biomass."

**3. The authors state that a sensitivity analysis presented in Figure 4 shows that management impact is robust to the initialization of the soil organic carbon stocks at the beginning of the spin-up phase. But, the stocks and change in stocks almost halves the values if the initialization is done with natural vegetation. The initialization does make a difference, and needs further explanation.**

Answer to 3.: It is correct that the SOC stocks are highly dependent on the legacy of management. However, Figure 4 shows that the SOC gap - the difference between a baseline scenario and a counterfactual scenario with constant management - was not strongly dependent on the initialization. We therefore concluded that the initialization does not affect our central finding.

Still, in order to improve our estimates also for the absolute SOC stock, we now extended the length of the spin-up phase, starting in 1510 (default spin-up start for introducing land use in simulations with LPJmL, see e.g. Schaphoff et al. 2018a/b, von Bloh et al. 2018). We describe this initialization in section 2.1.4, and updated the results section accordingly. This new feature makes our sensitivity analysis and the connected discussion obsolete and shortened the manuscript.

(von Bloh, W. et al. 2018: Implementing the nitrogen cycle into the dynamic global vegetation, hydrology, and crop growth model LPJmL (version 5.0). Geoscientific Model Development 11, 2789–2812. Schaphoff, S. et al. 2018a: LPJmL4 – a dynamic global vegetation model with managed land – Part 1: Model description. Geoscientific Model Development 11, 1343–1375. Schaphoff, S. et al. 2018b: LPJmL4 – a dynamic global vegetation model with managed land – Part 2: Model evaluation. Geoscientific Model Development 11, 1377–1403.)

**4. Good to see that the authors have made a comparison to another approach to confirm the Tier 2 results. The Tier 1 method provided by the IPCC has been used for this purpose. In section 2.2, the authors present a method estimating stock change factors instead of soil organic carbon changes. But, the results in Table 4 for the stock change factors are not convincing that the methods are consistent, and the text seems unclear with discussion about larger differences with the IPCC 2019 values, which were updated by the authors and should be more accurate – I would think. Why not estimate the change in soil organic carbon for a direct comparison with the Tier 2 method instead of the stock change factors? Also, the placement of these results after the discussion seems odd, and conventionally would be presented in the results section before discussion.**

Answer to 4.: Thank you for the recommendation. We placed the comparison within the result section. For a Tier 1 approach the change in soil organic carbon must be calculated based on a reference stock. The default stocks given by the IPCC have a very low spatial resolution (42 coarse climate zones and soil type specific values), which lead to additional uncertainty when directly comparing changes in soil organic carbon. As our aim is to isolate the management impact on SOC stocks, the stock change factors seem to us more informative than the absolute levels (which mix the uncertainties of the SOC stocks and the SOC change by management). We added more detail to the discussion of the strong deviation between the 2006 and 2019 default stock change factors.

It now reads:

LN 481-495: "In comparison with default stock change factors of the IPCC guidelines, our model estimates a stronger decline of SOC stocks (Table 3) for almost all regions. Tropical soils might suffer from low C input rates due to large yield gaps (Global Yield Gap and Water Productivity Atlas. Available URL: www.yieldgap.org (accessed on: 03/01/2022)) and high shares of residue removal and burning in lower-income countries (Smil, 1999a; Williams et al., 1997; Jain et al., 2014). Yet, even when comparing our etimates to the low-input stock change factors of the IPCC, our SOC loss is roughly twice as large as the revised 2019 IPCC default values, while it shows very good agreement with the older default values from IPCC (2006). Don et

al. (2011) estimated SOC losses for tropical soils of around 25% on average corresponding to a stock change factor of 0.75, but also reported a wide range of measured SOC changes from -80% to +58%. Fujisaki et al. (2015) however found much lower loss rates of around 9%, attributing the difference to the different time period length since the conversion to cropland. As our results do not specifically account for cropland age and most of the cropland is older than 20 years (as assumed for the default IPCC Tier 1 stock change factors) our stock change factors have to be lower by definition following the steady-state assumption that cropland will continue to approach a new equilibrium. For the same reason, our estimates for temperate regions might be lower than both IPCC (2006) and IPCC (2019) default values. With the production-increasing impact of irrigation and fertilization on carbon-poor dryland soils, SOC under cropland can also be higher than under PNV with stock change factors above 1 (see Fig. 2(d)), but these areas are much smaller than where the stock change factors are well below unity."

**5. Figure 3 shows results from making certain practices constant from 1975 to 2010.The authors state around line 315 that the effect of no-till has been strong since 1990,but the effect seems minor and may not differ statistically from the histManagement with uncertainty. The conclusion about the importance of residue seems most important here.**

Answer to 5.: Indeed, the effect of residues management seems most important. We now highlight this at several places in the discussion and removed the statement about tillage, e.g.

- LN 349-350: "Both the constManure and constTillage scenarios show only small deviations from the historical agricultural management values with $0.15\,\mathrm{GtC\,yr^{-1}}$. The effect of no-tillage only becomes discernible from 2000 onwards."
- LN 448-452: "The substantial impact of changing management practices through time is indicated by the development of our estimated stock change factors (see Table 3)) as well as by the time trend of the SOC debt (see Fig. 2(a)). Residue management has changed over the last decades, especially with the phasing out of residue burning practices in several regions and increased general productivity, showing a clear impact on SOC dynamics — underlining the importance to account for these effects in soil carbon modeling."
- LN 533-538: "Herzfeld et al. (2021) also consider historical management trends for fertilizer and manure inputs as well as on residue removal rates and tillage systems, but cannot reproduce the substantial increase in agricultural productivity over the last decades. Still, they find that compared to no-tillage systems, residue management has much larger potential to affect the strength of their projected future global cropland SOC decline. This is consistent with our finding that increasing SOC inputs from above-ground residues had the strongest effect on the slowing-down of the SOC debt increase (Fig. 5)."

**6. The authors suggest that there needs to be a circular flow with food supply chain back to soils. They assumed that none of the waste from supply chains are returned to soils (near line 300) but this seems incorrect. Municipal waste and materials are amended to soils in many regions of the world although maybe there are no data on these amendments? If this is the issue, the authors could mention that they are making a conservative assumption due to lack of data.**

Answer to 6.: Indeed. We now write:

LN 430-434: "While our data set, by including crop residues and manure, likely the largest carbon inputs to soils, it does not account for a list of minor carbon inputs from cover crops, agroforestry, green manure, weed biomass as well as application of human excreta, sewage sludge, processing wastes, forestry residues or biochar. Including these sources would correct our estimates upwards and bring our estimates closer to the IPCC stock change factors (see Sect. 3.4.1). Unfortunately, data on the quantity of these inputs is very scarce and does not exist with global coverage."

**7. The authors evaluate the sensitivity of the Tier 2 model for tree litter with methods in Section 2.4.3. The Tier 2 model divides litter into metabolic and structural components,and the authors have averaged lignin to nitrogen across tree components as input to the model. But forest also include deadwood and should be separated from other forest litter to model decomposition. Did the authors add a deadwood pool? I question if this model is appropriate**

**for forest if deadwood is not modeled separately.**

Answer to 7.: We follow this suggestion and added more detail on the litterfall of the natural vegetation. Using additional information from LPJmL4, we use plant-functional type specific parameterization for the natural litterfall and split up the litterfall into a wood, leaf and root fraction adding different parameterizations for these (see Sect. 2.1.1 and Table 1). We are not be able to add a deadwood pool, since that would require additional parameterization of turnover dynamics for this new pool. Deadwood pools are also not treated explicitly in many DGVMs but are considered part of litter pools, which distinguish woody from non-wood litter pools. The separation of litterfall into wood and soft tissue fluxes will thus add similar stock detail as in DGVMs.

**8. Recommend that the authors provide more explanation for Equation 9, which determines the residue amount of C, and is a key driver of the carbon change. Harvest index is the proportion of plant biomass that is harvested, but the authors are multiplying the harvested crop product by the harvest index. But the conventional approach is 'harvested crop production divided by the harvest index' to determine the total biomass and then subtract the harvested amount to estimate the residue. The authors are accounting for double harvesting and fallow in this calculation, which I agree is important, but some further explanation is needed about the calculation to understand how residue carbon is estimate from crop production, harvest index and area.**

Answet to 8.:

Crop residue production (ton dry matter residues) are calculated based on a linear function with positive intercept and a yield-dependent slope. This allometric function accounts for the fact that higher-yielding crops often have a lower harvest index than low-yielding crops. In our revised manuscript, we rewrote the Eq. 9 to make the functional form more visible to

$$
\begin{aligned}
AGR_{i,t,cg} &= CA_{i,t,cg} \cdot \left( Y_{i,t,cg} \cdot r_{cg}^{\mathrm{ag,prod}} + MCF_{i,t} \cdot r_{cg}^{\mathrm{ag,area}} \right) \qquad \text{and} \\
BGR_{i,t,cg} &= (CA_{i,t,cg} \cdot Y_{i,t,cg} + AGR_{i,t,cg}) \cdot r_{cg}^{\mathrm{bg}} \quad .
\end{aligned}
\tag{2}
$$

Where one can also simplify $CA \cdot Y$ to a production value, as we did in the last version of the manuscript; keeping area and yield explicit in the new version however makes the dynamic harvest index more explicit.

We also added more detail to Sect. 2.3.2 on "Crop and crop residues production" and units for the three crop-group $cg$ specific ratios: * above-ground residues to harvested biomass $r_{cg}^{\mathrm{ag,prod}}$ in $(tDM\,ha^{-1})(tDM\,ha^{-1})^{-1}$ * above-ground residues to harvested area $r_{cg}^{\mathrm{ag,area}}$ in $tDM\,ha^{-1}$ * below-ground residues to above-ground biomass $r_{cg}^{\mathrm{bg}}$ in $tDM\,tDM^{-1}$.

**9. For the Tier 1 method, IPCC divides the reference carbon stocks by climate and soil types. Did the authors also divide the grid cells by climate and soil because only climate is mentioned in the text? And, I found a diagram in Figure 5.1 in the IPCC report that divides low, medium and high input categories. Did the authors use this diagram to classify the input? It is not clear if the authors use the diagram or developed their own. If they developed their own, is it consistent with the IPCC factors?**

Answer to 9.: We do not calculate SOC stocks and SOC changes based on the Tier 1 method. Instead we calculate stylized stock change factors based on the Tier 1 method, and compare them to our Tier 2 approach. Soil types therefore do not influence these factors, as the management is multiplied on top of the stocks. Our approach therefore stays fully consistent with the IPCC factors.

**10. What is 'resp' is 'area reduction resp' on line 110? This sentence should be revised to improve readability. I also found other sentences that were difficult to read or missing words in some cases, but did not make a list during my review. Suggest a careful review before final publication.**

Answer to 10.: We changed in LN 137-138:

*"with $A_{lu}$ being the land-use type specific areas, $AR_{lu}$ and $AE_{lu}$ the area reduction resp. area expansion of the two land-use types"*

to

*"with $A_{lu}$ being the land-use type specific areas, $AR_{lu}$ the area reduction and $AE_{lu}$ the area expansion of the two land-use types."*

**11. I found the Tier 2 method in Chapter 5 of Volume 4 of the 2019 IPCC report, and would suggest that the authors cite this chapter rather than the entire 2019 IPCC report,which has 5 volumes. This would make it easier for others interested in the study to find the method in the IPCC report.**

Answer to 11.: We took up the reviewers comment and changed from citing the whole IPCC guidelines Vol. 4 (Eggleston et al., 2006) and their refinements (Calvo Buendia et al., 2019) to referencing the Cropland chapters (Lasco et al., 2006; Ogle et al., 2019). Note that not all references changed, since part of the methodology is stated in other chapter (e.g. on "Generic Methodologies Applicable to Multiple Land-use Categories" - Chapter 2).

---

## Author Response (AR2)

**Point by point replies II**

**Point by point replies - Jonathan Sanderman (2. round)**

**Preface: The authors have thoughtfully addressed all reviewer comments and have done a thorough job in revising and improving the manuscript. With fresh eyes, I found the manuscript to be extremely clear and insightful, and I believe it will have a big impact on our field. I have only a few comments and suggestions for further discussion of your findings:**

Answer to preface:

Dear Dr. Sanderman,

thank you for reviewing our paper again and adding very helpful points. We believe especially the hint to swidden agricultural practices is very important, even so we could not integrate this into the study, but are happy to take it up for future research. We here respond point-by-point to the your comments. Please note that we only included the updated (and not the outdated) version of the respective statements and paragraphs. We refer to the marked-up manuscript version for showing the changes in more detail.

**1. L365 - The IPCC2006 values for the tropics were highly criticized as being way too high. The 2019 refinements incorporated >10x as much data. The fact that your results are showing large losses in tropics is problematic and worth some discussion (see additional comment below). Note IPCC factors are derived under the assumption that there is a linear change between steady states over 20 years. Are Fscf factors consistent with this assumption? If not, this might not be a fair comparison.**

Answer to 1: We added the following clarification in the method and results part:

LN 188 (in the method part): "Note that IPCC factors are derived under the assumption that there is a linear change between steady states over 20 years."

LN 394 (in the results part): "Note that IPCC Tier 1 factors are derived under the assumption that there is a linear change between steady states over 20 years, whereas our aggregated factors just reflect the relative change compared to a given potential natural vegetation reference stock without specifically tracking the age of the cropland."

LN 398 (in the results part): "[For the tropical regions the IPCC factors increased notably from the guidelines in 2006 (Lasco et al., 2006) to the update in 2019 (Ogle et al., 2019)] due to the inclusion of more data points."

We reframed the discussion (also including the point on swidden agriculture practices) - see answer to point 8.

**2. L378 – This section can go for more than one sentence. Can you describe the results? Goodness of fit for natural veg and cropland (add R2 values to Fig 7)? Individually the results look fairly poor.**

Answer to 2.:

We shifted the figure into the appendix and added additional information within the manuscript. It now reads:

LN 414: "In Fig. A3 we correlate our SOC results for natural vegetation and cropland in 2010 with literature values from point measurements (for data base see appendix of Sanderman et al., 2017). The goodness of the fit is very low with an $R^2$ of 0.13. Individually, the correlations are even lower with a $R^2$ of 0.09 for cropland

and 0.08 for areas of natural vegetation. This points to the fact that differences between land-use type SOC stocks could be better matched than the spatial pattern of the rather small point measurement data base. Due to the low number of small-scale measurements, statistical properties of the point data variability are not derived and thus, could not be used to improve the point-to-grid-cell comparison (see Rammig et al., 2018)."

**3. Fig 6 - There is a big difference between soilgrids and soilgrids2.0 - in v2.0, they explicitly excluded litter (i.e. organic) horizons on top of dominantly mineral soils. I think this makes your cropland results more comparable to v2.0 but for global stocks, the original soilgrids product may be more reasonable particularly for northern soils where thick litter layers dominate SOC stocks.**

Answer to 3.:

We add clarification on the difference between the two SoilGrids products and its comparability. We rewrote the paragraph to:

LN 408: "SoilGrids (Hengl et al., 2017) especially stands out with its high estimate, since they include the litter horizon on top of the soil that might dominate especially polar and boreal soils. SoilGrids 2.0 (Poggio et al., 2021) however, excludes litter C and thus marks the lower end particularly for northern regions. For the same reason it is also more comparable to our results, which do not account for litter C as well."

**4. L383 - It seems hardly surprising to find good correlation between model results since both models have similar structure albeit with different levels of complexity**

Answer to 4.:

We agree with the reviewer that this comparison is not very central to evaluate model performance and therefore moved the figure to the appendix. Additionally, we added the following sentence to the results section to clarify the added value of this comparison from our perspective:

LN 422: "As the Tier 2 modeling approach (Ogle et al., 2019) is not specifically parameterized for natural vegetation, it is important to evaluate its suitability to produce reasonable results in that domain as well at least comparable to other modeling approaches."

**5. L407 – Thinking about structural uncertainly, how would you reconcile your findings with that of the CMIP6-LUMP (Ito et al 2020 ERL - https://iopscience.iop.org/article/10.1088/1748-9326/abc912) which found strong increases in SOC? I think there are strong CO2 feedbacks in those models.**

Answer to 5.: The SOC debt — as the difference of a natural state to a actual state — only indirectly include changes of SOC with climatic or atmospheric conditions. As our analysis is focused on time trends of the SOC debt and not the SOC stocks itself, most of the effect of CO2 feedbacks cancel out.

However, we added some results on total SOC stock dynamic to the results part and also pick up the above study together with the added numbers within the discussion section. It now reads:

LN 326 (in the results part): "Note that the SOC stock itself — without comparing it to a PNV state — increases during the period between 1975 and 2010 from $705\,\mathrm{GtC}$ to $712\,\mathrm{GtC}$, which corresponds to an overall SOC stock increase of $0.2\,\mathrm{GtC\,yr^{-1}}$."

LN 507 (in the discussion part): "Additionally, we find our $0.2\,\mathrm{GtC\,yr^{-1}}$ SOC stock change within the period between 1975 and 2010 for the first 30 cm of the soil profile at the upper end of estimates comparing it to estimates of Ito et al. (2020) of $0.18 \pm 0.41\,\mathrm{GtC\,yr^{-1}}$ within the period between 1850 and 2014 for the whole soil profile. This might be not surprising as the $CO_2$ effect is most likely stronger and land-use change effects weaker within our later and shorter modeling period compared to a mean value in the period between 1850 and 2014. The large uncertainty within estimates of SOC stock and its changes (Ito et al., 2020) again points to the large structural uncertainty within SOC modeling."

LN 516 (in the discussion part): "Moreover, the temporal dynamic of the SOC debt and stock change factors is not (or only to a small degree) altered by climatic or atmospheric effects on SOC stocks, as they cancel out

by taking the difference (for the SOC debt) and ratio (for the stock change factors) of cropland SOC and SOC under hypothetical PNV conditions."

**6. L425 - I disagree that input uncertainty is the largest source of error. Structural uncertainty is likely much larger − look at inter-model comparison studies such as CMIP5 (Todd-Brown publications) and others that actually include novel model structures (i.e. microbially-explicit model classes). . . structural uncertainty leads to differences in +/- 100s of GtC. Wieder et al (https://doi.org/10.1111/gcb.13979) also demonstrated this with a testbed of constant input data.**

Answer to 6.:

We agree with the reviewer that the effect of structural uncertainty were underestimated. It seems hard to make a judgement on the most important uncertainties. We reframed the sentence. It now reads:

LN 465: "This is why, in our view, a large part of the uncertainty with respect to the impacts of SOC management — next to the parametric and structural uncertainty of the soil model — is included in the management data itself."

**7. L455 - again, model to model comparison doesn't mean much if the model structure is similar. The comparison to PNV point data (with the caveat that a large grid cell and a point measurement are hard to reconcile) does not look nearly as good.**

Answer to 7.:

We reframed the sentence slightly and added two sentences on the point-data validation. It now reads:

LN 497: "The comparison of simulated PNV data with LPJmL4 shows the model's capability in reproducing PNV SOC stocks (Fig. A5). Concurrently, the point-data comparison (see Fig. A4) shows low correlation for PNV, however also for cropland sites. This might also point to the fact that SOC stocks can vary at field and local scale considerably and thus a very high number of point data is needed to derive statistical properties that could improve the point-to-grid-cell comparison (see Rammig et al., 2018)."

**8. L473 − This is not quite accurate. Erosion moves a lot of SOC into aquatic reservoirs, not just to somewhere else on the land, and the 'dynamic' replacement (i.e., high net sink strength on eroding surfaces) of that carbon is what is leading to the offset in losses.**

Answer to 8.: We added a references supporting the claim made by the reviewer and weakened the sentence in order to show the still controversial nature of the erosion sink paradox. It now reads:

LN 531: "In our model, erosion might however only affect the spatial pattern but not the aggregate SOC pool. As pointed out by Doetterl et al. (2016), the final fate of leached or eroded carbon is uncertain and might even offset LUC emissions (Wang et al., 2017). Concurrently, other studies have claimed erosion moves SOC into aquatic reservoirs (Zhang et al., 2020) and thus changing total global terrestrial SOC stock."

**9. L487 - Another thought on the particularly high losses in the central african basin region − there is a lot of swidden agriculture practiced here - several years of cropping followed by 10 or more years of fallow to allow soil fertility (and presumably SOC) to rebuild. Is this type of cropping pattern represented in your management data?**

Answer to 9.: Thank you very much for this comment. We think this is a very valid point and added the missing carbon from fallow land (as part of the swidden agriculture practices) to the discussion. We rephrased a large part of the paragraph on stock change factors. It now reads:

LN 543: "Yet, even when comparing our estimates to the low-input stock change factors of the IPCC, our SOC loss is roughly twice as large as the revised 2019 IPCC default values (2019), while it shows good agreement with the older default values from IPCC (2006). However, the revised estimates of the IPCC included much more and more recent data points, calling for a closer look on causes for the large deviations between our results and the refined Tier 1 factors. On the one hand, our approach does not account for unharvested carbon inputs from weeds or biomass cover on short-term fallows. Shifting agriculture with fallow periods might be prominent in tropical regions. While long-term fallow land (>4 years) is excluded

from FAOSTAT as cropland, short-term fallow is not. Thus, our carbon inputs for these areas might be underestimated, leading to too low stock change factors. On the other hand, older studies by Don et al. (2011) estimated SOC losses for tropical soils of around 25% on average corresponding to a stock change factor of 0.75, but also reported a wide range of measured SOC changes from -80% to +58%. Fujisaki et al. (2015) however found much lower loss rates of around 9%, attributing the difference to the different time period lengths since the conversion to cropland. As our results do not specifically account for cropland age and most of the cropland is older than 20 years (as assumed for the default IPCC Tier 1 stock change factors) our stock change factors have to be lower by definition following the steady-state assumption that cropland will continue to approach a new equilibrium."

**10. L536 – It is not surprising that tillage didn't have much of an impact. In line with observations, Century model does not have a strong tillage feedback. Dangal et al 2022 (JAMES - https://agupubs.onlinelibrary.wiley.com/doi/full/10.1029/2021MS002622) showed shifting to no-till had pretty minimal effects of SOC when running DayCent across parts of the US.**

Answer to 10.:

Thanks for that suggestion. We added this citation. It now reads:

LN 609: "In line with this, Dangal et al. (2022) finds that no-tillage has only minor impacts on SOC dynamics across parts of the US."

**Point by point replies - second reviewer**

**0. This paper is a convincing step forward for global cropland modelling of soil carbon, based on land use and agricultural management data that are dynamic in time going back to 1975, and a model framework that is reduced in complexity in order to allow global modelling. The paper is rather long but of great value for the soil carbon and modelling community. The authors clearly state the limitations of the approach and involve several model evaluation steps that are of great help to see the limitations but also the robustness of the results. The paper is written well. At some points it could be improved in order to make it better understandable and easier to read. In particular a clear differentiation between land use change effects and agricultural management effects within croplands would be important. A topic that is not fully explored is the effect of climate change vs. anthropogenic direct impact on global soil carbon dynamics (but this maybe topic for a next paper).**

Answer to 0.: Dear reviewer, thank you for the thorough and helpful review of our manuscript. We agree with the reviewer that more detailed analysis of the magnitude of management effects compared to land-use change effects is needed and added two additional global maps on attribution of SOC debt change (see Answer to 14.). Here, we respond point-by-point to the reviewers' comments. Please note that sometimes due to the amount of changes introduced within this update, we only refer to the section within the new manuscript rather than including the entire changed paragraph within these replies.

Detailed remarks:

**1. l. 1-2: The first sentences of the abstract are too general and has been repeated too often – please remove.**

Answer to 1.:

We shortened the respective sentences to:

LN 1: "Soil organic carbon (SOC), one of the largest terrestrial carbon (C) stocks on Earth, has been depleted by anthropogenic land-cover change and agricultural management."

**2. l. 17: The carbon pool in the lithosphere (including fossil carbon) is much larger than the soil C pool.**

Answer to 2.: We removed the statement so that the sentence now reads:

LN 16: "Soil organic carbon (SOC), the amount of organic carbon stored in the Earth's soil, exceeds the carbon in the atmospheric and vegetation pools multiple times (Batjes, 1996)."

**3. l. 41: N-deposition does not play a role in agricultural systems that are fertilized or in areas with no N-deposition. Also, CO2 fertilisation is of minor importance compared to agricultural management including breeding.**

Answer to 3.:

We agree that these aspects may not be of central importance for SOC dynamics on agricultural land and have removed the claim. Now it reads:

LN 38: "BKMs are designed to estimate LUC related emissions and have largely improved in estimating additional emissions from wood harvest and shifting cultivation. However, state of the art models do not consider impacts of varying agricultural management (Frielingstein et al., 2020; Houghton et al., 2012_carbon_2012; Hansis et al., 2015; Bastos et al., 2021)."

**4. L, 46: What are stylized scenarios? And in l. 54: stylized future management? Please explain or rephrase.**

Answer to 4.:

We added some more detail to what we mean with stylized. It now reads:

LN 42: "Pugh et al. (2015) explicitly consider agricultural management in the form of tillage, irrigation and biomass extraction at harvest, but worked with uniform scenario assumptions on management rather than with historical management data."

LN 47: "In global-scale carbon cycle assessments, management systems are typically represented as spatially explicit patterns that are static in time (e.g. for growing seasons in Portmann et al., 2010; multiple cropping systems in Waha et al., 2020; irrigation systems in Jägermeyr et al., 2015) or as stylized (in the sense of uniform management assumptions) scenarios (e.g. Pugh et al., 2015; Lutz et al., 2019)."

**5. l. 56: Please explain why no net sink is possible? In the next paragraph you write about soil C sequestration. The reader might ask why if it is not possible.**

Answer to 5.:

In the above mentioned sentence, we refer to the results of one specific study (Herzfeld et al. 2021), which reports that none of their studied changes in management (if applied globally) could counteract the legacy flux caused by the initial land conversion. We have revised the sentence as follows to avoid any confusion about the general possibility of net carbon sinks. It now reads:

LN 52: "Within their stylized future management scenarios under future climate change they find that none of the management aspects considered (residue management, no-tillage) can create a positive SOC stock change on current cropland areas that counteracts the still continuing legacy flux from the initial land-use change."

**6. l. 106: How did you treat grasslands that are no pastures (mowed meadows e.g.)?**

Answer to 6.:

We changed pastures to grasslands and added more detailed explanations here. It now reads:

LN 102: "We distinguish two land-use types: cropland and uncropped land under potential natural vegetation as representative for all other land-use types including forestry and grasslands (referred to as natural vegetation in the following). Forage crops are included within cropland, whereas pastures (including mowed meadows (perennials) and rangelands) are assigned to natural vegetation. Carbon flows connected to livestock are only considered in this study when they originate from cropland feed sources, while the manure originating from pasture biomass is disregarded, implicitly assuming that this manure is excreted or applied to pastures."

**7. Fig 1: I am not sure if this figure is required to understand the paper. However, please check the values for SOC stocks at the y-axis. They are too low for 0-30 cm SOC stocks for most regions of the world.**

Answer to 7.: We changed the unit of the values to 0 - 100 t/ha to represent a more realistic SOC stock. Another reviewer found this figure to be of value help to understand, why we do not have to distinguish between newly converted and existing cropland, so that we are convinced that it should remain in the manuscript.

**8. l. 169: The assumption that agricultural management from 1510 to 1965 was the same (the same as in 1965) is an assumption with high uncertainty. You might need to explain how important or not important this assumption is for your results.**

Answer to 8.: Indeed, this is an important point. There are no data and any simple assumption is obviously wrong. We acknowledge this shortcoming in the methods section and discuss the initialization uncertainty (see answer to point 18). We added:

LN 171: "We acknowledge that this introduces a bias as agricultural management has changed prior to 1965, but this approach follows other studies on effects of land-use change and management (e.g. Schaphoff et al., 2018; Herzfeld et al., 2021) and is limited by data availability on harvest statistics (and other management effects)."

**9. Chapter 2.2: This consistency check with the IPCC Tier 1 approach is very useful and good.**

Answer to 9.: Thanks. We improved readability of the results section on the IPCC Tier 1 comparison as well (see answer to 15.).

**10. The start of the results section (l. 304) with two references from the same authors may give the impression that results are published in other papers. Please indicate here that you refer to supplementary studies supporting the actual study with data and code.**

Answer to 10.: We clarified that both citations refer to data and script repositories, so it now reads:

LN 318: "Detailed results for the spatially explicit global SOC budget including intermediate results on input data as well as SOC stock results for all scenario runs can be found in the data repository from Karstens (2020a). In the following, the most important results (see Karstens (2020b) for post-processing scripts) are summarized."

**11 Fig 2b) The choice of the colours is suboptimal since the different green cannot be distinguished by eye. Please use more contrasting and different colours. The Fig 2b-d give the impression that most land on earth is cropland even though only around 10% of land surface is cropland.**

Answer to 11.: We agree with the reviewer and changed the color scale to a more distinguishable palette. We already excluded all cells from the global maps with less than 1000 ha cropland. Maps that additionally to the SOC information (stock, diff or change factor) include information on the cropland extent (e.g. via saturation of the color) are in our opinion more complex to understand. Thus, we decided against this and stated in legend and caption that this maps are exclusively for cropland soils.

**12 l. 339: This sentence is not clear to me. It would be better to provide sums of the global input to cropland soils and not to the agricultural system (with undefined system borders). It would be also useful to include terms such as net primary production (NPP) here (also in Fig 4). Also, the term human appropriated fraction of NPP (HANPP) maybe useful also to compare with in the discussion. Derived from the data you stated that 2463 Mt C entered global croplands each year (fig 4). The fraction of manure (16%) is unexpected high since manure can only be transferred to croplands if it is collected (mostly in stables), which is not common in many parts of the world. How much of this manure is feedstock grazing on croplands?**

Answer to 12.: We agree with the reviewer that "agricultural system" is rather undefined and changed it for all occurrences to "cropland system" as our study focus is on croplands only.

On the second point on HANPP, we like kindly have to decline the reviewers suggestion to include this into this study. As our study focus only on the cropped NPP/HANPP not covering grazed system and the whole forestry sector, a comparison is beyond the scope of this study.

At the last point, we like to emphasis that our results came with large uncertainties especially for the manure recycling shares, as they are derived indirectly from feed consumption. This might be the cause for systematic errors that might led to high shares of manure C input. We point to this already within our discussion part (LN 467). Concurrently, even so advanced animal waste management systems (AWMS) are not very common in most parts of the world, a large share of the livestock production is however concentrated in intensive livestock systems often featuring advanced AWMS. Please also see our answer to your point 6, where we added more details to feed biomass from cropland and pastures and LN 102 of the method section 2.3.3, where we outline our assumption on stubble grazing. Moreover, as mentioned in beginning of our result section (LN 318) intermediate results, like the 18 MtC stubble grazed manure, can be derived from the data upload (hopefully quite easily).

**13. The fraction of above ground biomass C input to the soil (55%) looks to me is quite high, see e.g. Bolinder et al. providing lower fractions. The root fraction should also include rhizodeposits, which might be not considered in your study.**

We agree with the reviewer that other studies have shown lower rates of residue inputs. We discuss this already in LN 478 in the discussion part by referring to Keel et al. (2017), who did a comparison of residue allometric equation (including studies from Bolinder et al.). They report rhizodeposists to be included in the residue estimates and also find that below-ground residue shares from the IPCC methodology are often higher than in other approaches (also included in our discussion LN 482).

**14 l. 306: This chapter gives a general overview on agricultural land use and management effect on SOC. The impact of land use changes from natural vegetation to croplands on SOC is well known and thus the maps shown here are in many regions of the world in line with the global cropland maps. The new aspect of this study is the agricultural management within croplands. It would be interesting to show maps for agricultural management effects, e.g. a standard or a worst-case scenario vs. the real data scenario. This is part of the next subchapter (3.3.). However, there it is not showed spatially explicit (with maps).**

Answer to 14.: We agree with the reviewer and added spatial explicit maps on the split up of land-use change and agricultural management effects (see Fig. 3). We also added the following to the result part Sect. 3.3:

LN 370: "Using the constManagement results that only include land-use change (LUC) related changes of the SOC debt between 1975 and 2010, we are able to subtract the LUC effect from the overall SOC debt change within the histManagement results. The remaining effect can be attributed to the changing agricultural management (MAN) as other drivers such as climatic effects have been already canceled out by taking the difference to a PNV reference state when calculating $\Delta SOC$. The increasing SOC debt on global cropland are primarily caused by LUC (red areas in Fig. 3(b2)). Deteriorated management also contributed to increasing SOC debt in parts of Sub-saharan Africa and Central Asia. In contrast, agricultural management has led to an decrease in SOC debt in the USA, Europe, as well as in parts of China and India (blue areas in Fig. 3(b3)), which is not visible in the total $\Delta SOC$ change as LUC was happening at the same time."

**15. Tab. 3 is not easy to read. For the IPCC values you display low medium and high values. For your modelled data, only an average (I guess medium) value are shown. Would it be possible to also provide low and high values derived from your models for each climate region? It might be also helpful to convert this table into a figure. More important, the effect of agricultural management is not visible in this data set since the IPCC values are developed for land use changes. Again, it would be important to distinguish between land use change and land management effects.**

Answer to 15.:

We agree with the reviewer and replace the table with a graphical representation including a range of values (5th, 50th to 95th percentile for each climate region) instead of a medium value in a table. The Tier 1 approach of the IPCC guidelines also tracks management decisions e.g. on tillage, but more importantly on input regimes. We included these input regime differentiation into the comparison figure as well and reframed the sentence in the result part. It now reads:

LN 398: "Stock change factors for temperate climate zones of this study are lower than the default values of the IPCC. For the tropical regions the IPCC factors increased notably from the guidelines in 2006 (Lasco et al., 2006) to the update in 2019 (Ogle et al., 2019) due to the inclusion of more data points. Our results span over a broad range due to the different ages of the cropland but also due to different agricultural management practices within climate regions."

Please also note that we added more detail to the discussion of the stock change factors from LN 540 on.

**16. Fig. 6 looks a bit strange with SOC on the x-axis instead of the y-axis. I would also like to see total stocks for croplands and for all other land use types separately.**

Answer to 16.:

Thanks for the suggestion. We changed x- any y-axis. We agree with the reviewer that land-use type specific comparison would be very useful. Unfortunately, to our knowledge, non of the validation data sources used for this comparison (maybe with the exception of LPJmL) reports land-use type specific carbon stocks. Thus, we can not present such a comparison within this study.

**17. Chapter 3.4.3: There are no results reported in this chapter for the point data comparison. Moreover, it is questionable if point based data are useful for model validation since SOC stocks can vary at field and local scale considerable and thus a very high number of point data (I would recommend >5000 points globally) of high quality (including bulk density measurements) are required for such an exercise. In the cited database of Sandermann et al. includes less than 300 sites. I think the comparison with the SoilGrid2.0 data is sufficient.**

Answer to 17.:

We shifted the figure into the appendix and added additional information within the manuscript. We did not dropped the section as a whole since it was requested by another reviewer. It now reads:

LN 414: "In Fig. A3 we correlate our SOC results for natural vegetation and cropland in 2010 with literature values from point measurements (for data base see appendix of Sanderman et al., 2017). The goodness of the fit is very low with an $R^2$ of 0.13. Individually the correlations are even lower with a $R^2$ of 0.09 for cropland and 0.08 for areas of natural vegetation. This point to the fact that differences between land-use type SOC stocks could be better matched than the spatial pattern of the rather small point measurement data base. Due to the low number of small-scale measurements, statistical properties of the point data variability are not derived and thus, could not be used to improve the point-to-grid-cell comparison (see Rammig et al., 2018)."

**18. l. 399: There might be a direct link between underestimated SOC and overestimated SOC stock increases and vice versa since models are sensitive with their modeled SOC trend to the initial SOC stock. Thus, it needs to be carefully checked and discussed whether underestimation or overestimation of SOC stocks are the reason for the predicted SOC trends. For example, the strong increase in SOC in arid regions might also be a results of an underestimation of initial SOC stocks.**

Answer to 18.:

We agree with the reviewer that SOC initialization is crucial to SOC stocks, SOC debt and its changes over time, since the size of the legacy fluxes will be affecting these values strongly. We add our sensitivity analysis on the initialization choice (see Sect. 3.3), which was included in the first version of our manuscript, back including an updated scenario definition (see Sect. 2.4) and a figure in the Appendix (see Fig. A2). It gives the reader an impression of the uncertainty towards the initialization choice and shows that the impact of the dynamic agricultural management is robust to the initialization of SOC stocks. We also included the following paragraph within the discussion section:

LN 519: "The initialization of SOC stocks, however, is important for the size of the SOC debt and its change over time, since the presence and size of legacy fluxes affect these values strongly (see Fig. A2). According to our sensitivity analysis the SOC debt varies between 33.3 GtC and 50.7 GtC depending on the initialization choice, with our best guess at 39.6 GtC. Concurrently, our results indicate that the impact of the dynamic agricultural management is robust to the initialization of SOC stocks."

**19. l. 500: Its not only tillage that can affect subsoil SOC but all land use and land management options can affect SOC in the subsoil below about 30 cm depth. Powslon et al 2014 found not significant tillage effect below 25 cm depth. This might be due to the low sample size of 43 sites. However, there is no evidence that tillage effects subsoil SOC stronger than other agricultural management or land use change.**

Answer to 19.:

The think the reviewer is right in pointing out that subsoil effects not only from tillage are neglected. However, other management practices considered in this study might be more equally effecting top- and subsoil, whereas tillage practices by definition are the only management option that relocates carbon vertically (as explained in the paragraph following the reviewers comment). We rewrote the paragraph also adding a reference that the effect to the subsoil and the overall C sequestration effect is still debated. It now reads:

LN 565: "Second, changes to the subsoil are neglected, which is most important for tillage effects. Other management practices might be more equally effecting top- and subsoil as they do not directly the relocates carbon vertically. As Powlson et al. (2014) have shown, the subsoil can be make a large difference in evaluating total SOC losses or gains for no-tillage systems. No-tillage effects may seem larger than they actually are if only topsoil is considered. SOC transfers to deeper soil layers under tillage might enhance subsoil SOC compared to no-till practices. However, the effect of no-till to the subsoil as well as its overall importance as a SCS measure is still debated (Ogle et al., 2019)."

**20. l. 528: Please provide more detailed data either here or in the results section how increasing yields affected SOC stocks in the past.**

We reframed the paragraph to more clearly on the direction and cause of the yield effect on SOC stocks. It now reads:

LN 596: "Their study moreover concludes that yield gains (by 18% in their simulations) do not lead to a substantial decline in SOC debt (less than 1% SOC increase). Historical yield increases, however, are estimated to be well above 50% (Pellegrini and Fernández, 2018; Ray et al., 2012; Rudel et al., 2009) and often lead to an increase in below- and above-ground residue biomass inputs to the soil. While we find substantially larger SOC increase in response to productivity gains than the 1% reported by Pugh et al. (2015), this is not sufficient to compensate SOC losses from global cropland expansion of around 11% between 1974 and 2010."

**21. l. 544. It would be more helpful to relate and compare the estimated SOC loss with deforestation in Gt/a to the estimated total land use change induced emissions of about 2.5 Gt/a.**

Answer to 21.: Unfortunately our model does not vegetation biomass, and thus we are only able to capture soil deforestation effects. However, comparing SOC related emissions to over all Land use change induced emissions seems to be a valuable addition. We added to the discussion:

LN 581: "Comparing our SOC loss rate (the change of SOC debt) of $0.14\,\mathrm{GtC\,yr^{-1}}$ to estimates on land-use change induced emissions of $2.0 \pm 1.0\,\mathrm{GtC\,yr^{-1}}$ (sum of 'Bookkeeping LULCC emissions' and 'Loss of additional sink capacity' for the years 2009–2018 in Gasser et al. (2020)), we find SOC emissions of the first 30 cm of the soil profile to be a minor contributor to overall land-use change induced emissions."

**22. l. 549: This sentence ("one fifth of total annual C sequestration by crops is lost through soils (0.8 GtC per year)") is not clear. Please be very careful with the term "C sequestration" throughout the manuscript. C sequestration is the removal and long-term storage of CO2 from the atmosphere. I guess you are referring to the plant photosynthesis flux. The figure 4**

**is rather complex and may need further description in order to make it understandable also here in your discussion.**

Answer to 22.:

Thank you for the suggestion to improve readability and clarity of that paragraph. We believe major confusion came from the un-updated numbers that were not in line with the figure presented in that version of the manuscript. We updated the numbers and reframed the sentence. It now reads:

LN 621: "As shown in Fig. 4, the annual SOC respiration (1.3 GtC per year) is slightly above one quarter of total annual net C uptake by crops (4.6 GtC per year). C compounds have to be respired by soil organisms to maintain basic soil functions and regulate the nutrient cycle, which often leaves limited options to decrease C losses via SOC respiration (Janzen, 2006). However, similar C losses occur at the end of the food supply chain (1.2 GtC per year), at the soil surface (1.5 GtC per year), and smaller but still considerable during residue burning (0.2 GtC per year) and within animal waste management systems (0.2 GtC per year)."

We also revised all occurrences of "sequestration" and changed to "enhancement", "accumulation" or other terms were it was needed.